# Revisiting Heterophily For Graph Neural Networks

**Sitao Luan**[1,2], **Chenqing Hua**[1,2], **Qincheng Lu**[1], **Jiaqi Zhu**[1], **Mingde Zhao**[1,2], **Shuyuan Zhang**[1,2], **Xiao-Wen Chang**[1], **Doina Precup**[1,2,3]

{sitao.luan@mail, chenqing.hua@mail, qincheng.lu@mail, jiaqi.zhu@mail, mingde.zhao@mail, shuyuan.zhang@mail, chang@cs, dprecup@cs}.mcgill.ca

[1]McGill University; [2]Mila; [3]DeepMind

## Abstract

Graph Neural Networks (GNNs) extend basic Neural Networks (NNs) by using graph structures based on the relational inductive bias (homophily assumption). While GNNs have been commonly believed to outperform NNs in real-world tasks, recent work has identified a non-trivial set of datasets where their performance compared to NNs is not satisfactory. Heterophily has been considered as the main cause of this empirical observation and numerous works have been put forward to address it. In this paper, we first revisit the widely used homophily metrics and point out that their consideration of only graph-label consistency is a shortcoming. Then, we study heterophily from the perspective of post-aggregation node similarity and define new homophily metrics, which are verified to be advantageous compared to existing ones. Based on this investigation, we prove that some harmful cases of heterophily can be effectively addressed by local diversification operation. Then, we propose the Adaptive Channel Mixing (ACM), a framework to adaptively exploit aggregation, diversification and identity channels node-wisely to extract richer localized information for diverse node heterophily situations. ACM is more powerful than the commonly used uni-channel framework for node classification tasks on heterophilic graphs and is easy to be implemented in baseline GNN layers. When evaluated on 10 benchmark node classification tasks, ACM-augmented baselines consistently achieve significant performance gain, exceeding state-of-the-art GNNs on most tasks without incurring significant computational burden. Code: https://github.com/SitaoLuan/ACM-GNN

## 1 Introduction

Deep Neural Networks (NNs) [22] have revolutionized many machine learning areas, including image recognition [21], speech recognition [13] and natural language processing [2], due to their effectiveness in learning latent representations from Euclidean data. Recent research has shifted focus on non-Euclidean data [6], *e.g.,* relational data or graphs. Combining graph signal processing and convolutional neural networks [23], numerous Graph Neural Network (GNN) architectures have been proposed [39, 10, 15, 41, 19, 30], which empirically outperform traditional NNs on graph-based machine learning tasks such as node classification, graph classification, link prediction and graph generation, *etc.*GNNs are built on the homophily assumption [35]: connected nodes tend to share similar attributes with each other [14], which offers additional information besides node features. This relational inductive bias [3] is believed to be a key factor leading to GNNs' superior performance over NNs' in many tasks.

However, growing empirical evidence suggests that GNNs are not always advantageous compared to traditional NNs. In some cases, even simple Multi-Layer Perceptrons (MLPs) can outperform GNNs by a large margin on relational data [46, 29, 32, 8]. An important reason for this is believed to be the heterophily problem: the homophily assumption does not always hold, so connected nodes may in fact have different attributes. Heterophily has received lots of attention recently and an increasing number of models have been put forward to address this problem [46, 29, 32, 8, 45, 44, 33, 16, 24]. In

this paper, we first show that by only considering graph-label consistency, existing homophily metrics are not able to describe the effect of some cases of heterophily on aggregation-based GNNs. We propose a post-aggregation node similarity matrix, and based on it, we derive new homophily metrics, whose advantages are illustrated on synthetic graphs (Sec. 3). Then, we prove that diversification operation can help to address some harmful cases of heterophily (Sec. 4). Based on this, we propose the Adaptive Channel Mixing (ACM) GNN framework which augments uni-channel baseline GNNs, allowing them to exploit aggregation, diversification and identity channels adaptively, node-wisely and locally in each layer. ACM significantly boosts the performance of 3 uni-channel baseline GNNs by 2.04% ~ 27.5% for node classification tasks on 7 widely used benchmark heterophilic graphs, exceeding SOTA models (Sec. 6) on all of them. For 3 homophilic graphs, ACM-augmented GNNs can perform at least as well as the uni-channel baselines and are competitive compared with SOTA.

**Contributions** 1. To our knowledge, we are the first to analyze heterophily from post-aggregation node similarity perspective. 2. The proposed ACM framework is highly different from adaptive filterbank with multiple channels and existing GNNs for heterophily: 1) the traditional adaptive filterbank channels [40] uses a scalar weight for each filter and this weight is shared by all nodes. In contrast, ACM provides a mechanism so that different nodes can learn different weights to utilize information from different channels to account for diverse local heterophily; 2) Unlike existing methods that leverage the high-order filters and global property of high-frequency signals [46, 29, 8, 16] which require more computational resources, ACM successfully addresses heterophily by considering only the **nodewise local information adaptively**. 3. Unlike existing methods that try to facilitate learning filters with high expressive power [46, 45, 8, 16], ACM aims that, when given a filter with certain expressive power, we can extract richer information from additional channels in a certain way to address heterophily. This makes ACM more flexible and easier to be implemented.

## 2 Preliminaries

In this section, we introduce notation and background knowledge. We use **bold** font for vectors (*e.g.*, $\boldsymbol{v}$). Suppose we have an undirected connected graph $\mathcal{G} = (\mathcal{V}, \mathcal{E}, A)$, where $\mathcal{V}$ is the node set with $|\mathcal{V}| = N$; $\mathcal{E}$ is the edge set without self-loops; $A \in \mathbb{R}^{N \times N}$ is the symmetric adjacency matrix with $A_{i,j} = 1$ *if* $e_{ij} \in \mathcal{E}$, otherwise $A_{i,j} = 0$. Let $D$ denote the diagonal degree matrix of $\mathcal{G}$, *i.e.*, $D_{i,i} = d_i = \sum_j A_{i,j}$. Let $\mathcal{N}_i$ denote the neighborhood set of node $i$, *i.e.*, $\mathcal{N}_i = \{j : e_{ij} \in \mathcal{E}\}$. A graph signal is a vector $\boldsymbol{x} \in \mathbb{R}^N$ defined on $\mathcal{V}$, where $\boldsymbol{x}_i$ is associated with node $i$. We also have a feature matrix $X \in \mathbb{R}^{N \times F}$, whose columns are graph signals and whose $i$-th row $X_{i,:}$ is a feature vector of node $i$. We use $Z \in \mathbb{R}^{N \times C}$ to denote the label encoding matrix, whose $i$-th row $Z_{i,:}$ is the one-hot encoding of the label of node $i$.

### 2.1 Graph Laplacian, Affinity Matrix and Variants

The (combinatorial) graph Laplacian is defined as $L = D - A$, which is Symmetric Positive Semi-Definite (SPSD) [9]. Its eigendecomposition is $L = U\Lambda U^T$, where the columns $\boldsymbol{u}_i$ of $U \in \mathbb{R}^{N \times N}$ are orthonormal eigenvectors, namely the *graph Fourier basis*, $\Lambda = \text{diag}(\lambda_1, \ldots, \lambda_N)$ with $\lambda_1 \leq \cdots \leq \lambda_N$. These eigenvalues are also called *frequencies*.

In additional to $L$, some variants are also commonly used, *e.g.*, the symmetric normalized Laplacian $L_{\text{sym}} = D^{-1/2} L D^{-1/2} = I - D^{-1/2} A D^{-1/2}$ and the random walk normalized Laplacian $L_{\text{rw}} = D^{-1} L = I - D^{-1} A$. The graph Laplacian and its variants can be considered as high-pass filters for graph signals. The affinity (transition) matrices can be derived from the Laplacians, *e.g.*, $A_{\text{rw}} = I - L_{\text{rw}} = D^{-1} A$, $A_{\text{sym}} = I - L_{\text{sym}} = D^{-1/2} A D^{-1/2}$ and are considered to be low-pass filters [34]. Their eigenvalues satisfy $\lambda_i(A_{\text{rw}}) = \lambda_i(A_{\text{sym}}) = 1 - \lambda_i(L_{\text{sym}}) = 1 - \lambda_i(L_{\text{rw}}) \in (-1, 1]$. Applying the renormalization trick [19] to affinity and Laplacian matrices respectively leads to $\hat{A}_{\text{sym}} = \tilde{D}^{-1/2} \tilde{A} \tilde{D}^{-1/2}$ and $\hat{L}_{\text{sym}} = I - \hat{A}_{\text{sym}}$, where $\tilde{A} \equiv A + I$ and $\tilde{D} \equiv D + I$. The renormalized affinity matrix essentially adds a self-loop to each node in the graph, and is widely used in Graph Convolutional Network (GCN) [19] as follows:

$$Y = \text{softmax}(\hat{A}_{\text{sym}} \, \text{ReLU}(\hat{A}_{\text{sym}} X W_0) \, W_1) \tag{1}$$

where $W_0 \in \mathbb{R}^{F \times F_1}$ and $W_1 \in \mathbb{R}^{F_1 \times O}$ are learnable parameter matrices. GCNs can be trained by minimizing the following cross entropy loss

$$\mathcal{L} = -\text{trace}(Z^T \log Y) \tag{2}$$

where $\log(\cdot)$ is a component-wise logarithm operation. The random walk renormalized matrix $\hat{A}_{\text{rw}} = \tilde{D}^{-1}\tilde{A}$, which shares the same eigenvalues as $\hat{A}_{\text{sym}}$, can also be applied in GCN. The corresponding Laplacian is defined as $\hat{L}_{\text{rw}} = I - \hat{A}_{\text{rw}}$. The matrix $\hat{A}_{\text{rw}}$ is essentially a random walk matrix and behaves as a mean aggregator that is applied in spatial-based GNNs [15, 14]. To bridge spectral and spatial methods, we use $\hat{A}_{rw}$ in this paper.

## 2.2 Metrics of Homophily

The homophily metrics are defined by considering different relations between node labels and graph structures. There are three commonly used homophily metrics: edge homophily [1, 46], node homophily [36] and class homophily [26] [1], defined as follows:

$$H_{\text{edge}}(\mathcal{G}) = \frac{\left|\{e_{uv} \mid e_{uv} \in \mathcal{E}, Z_{u,:} = Z_{v,:}\}\right|}{|\mathcal{E}|}, H_{\text{node}}(\mathcal{G}) = \frac{1}{|\mathcal{V}|}\sum_{v \in \mathcal{V}} H_{\text{node}}^v = \frac{1}{|\mathcal{V}|}\sum_{v \in \mathcal{V}} \frac{\left|\{u \mid u \in \mathcal{N}_v, Z_{u,:} = Z_{v,:}\}\right|}{d_v},$$

$$H_{\text{class}}(\mathcal{G}) = \frac{1}{C-1}\sum_{k=1}^{C}\left[h_k - \frac{\left|\{v \mid Z_{v,k}=1\}\right|}{N}\right]_+, \quad h_k = \frac{\sum_{v \in \mathcal{V}}\left|\{u \mid Z_{v,k}=1, u \in \mathcal{N}_v, Z_{u,:}=Z_{v,:}\}\right|}{\sum_{v \in \{v \mid Z_{v,k}=1\}} d_v}$$

$$(3)$$

where $H_{\text{node}}^v$ is the local homophily value for node $v$; $[a]_+ = \max(a, 0)$; $h_k$ is the class-wise homophily metric [26]. All metrics are in the range of $[0, 1]$; a value close to $1$ corresponds to strong homophily, while a value close to $0$ indicates strong heterophily. $H_{\text{edge}}(\mathcal{G})$ measures the proportion of edges that connect two nodes in the same class; $H_{\text{node}}(\mathcal{G})$ evaluates the average proportion of edge-label consistency of all nodes; $H_{\text{class}}(\mathcal{G})$ tries to avoid sensitivity to imbalanced classes, which can make $H_{\text{edge}}(\mathcal{G})$ misleadingly large. The above definitions are all based on the **linear feature-independent graph-label consistency**. The inconsistency relation is implied to have a negative effect to the performance of GNNs. With this in mind, in the following section, we give an example to illustrate the shortcomings of the above metrics and propose new feature-independent metrics that are defined from post-aggregation node similarity perspective, which is novel.

# 3 Analysis of Heterophily

## 3.1 Motivation and Aggregation Homophily

Heterophily is widely believed to be harmful for message-passing based GNNs [46, 36, 8] because, intuitively, features of nodes in different classes will be falsely mixed, leading nodes to be indistinguishable [46]. Nevertheless, it is not always the case, *e.g.,* the bipartite graph[2] shown in Figure 1 is highly heterophilic according to the existing homophily metrics in equation 3, but after mean aggregation, the nodes in classes 1 and 2 just exchange colors and are still distinguishable[3]. This example tells us that, besides graph-label consistency, we need to study the relation between nodes after aggregation step.

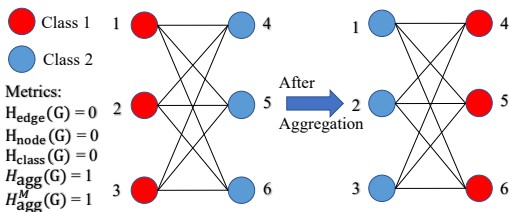

Figure 1: Example of harmless heterophily

To this end, we first define the post-aggregation node similarity matrix as follows:

$$S(\hat{A}, X) \equiv \hat{A}X(\hat{A}X)^T \in \mathbb{R}^{N \times N} \tag{4}$$

where $\hat{A} \in \mathbb{R}^{N \times N}$ denotes a general aggregation operator. $S(\hat{A}, X)$ is essentially the gram matrix that measures the similarity between each pair of aggregated node features.

**Relationship Between $S(\hat{A}, X)$ and Gradient of SGC**    SGC [42] is one of the most simple but representative GNN models and its output can be written as:

$$Y = \text{softmax}(\hat{A}XW) = \text{softmax}(Y') \tag{5}$$

---

[1][26] did not name this homophily metric. We named it *class homophily* based on its definition.

[2][33] use the same example but not to demonstrate the deficiency of homophily metrics.

[3][8] also point out the insufficiency of $H_{\text{node}}$ by examples to show that different graph typologies with the same $H_{\text{node}}(\mathcal{G})$ can carry different label information.

With the loss function in equation 2, after each gradient descent step, we have $\Delta W = \gamma \frac{d\mathcal{L}}{dW}$, where $\gamma$ is the learning rate. The update of $Y'$ is (see Appendix E for derivation):

$$\Delta Y' = \hat{A}X\Delta W = \gamma\hat{A}X\frac{d\mathcal{L}}{dW} \propto \hat{A}X\frac{d\mathcal{L}}{dW} = \hat{A}XX^T\hat{A}^T(Z-Y) = S(\hat{A},X)(Z-Y) \qquad (6)$$

where $Z - Y$ is the prediction error matrix. The update direction of the prediction for node $i$ is essentially a weighted sum of the prediction error, *i.e.*, $\Delta(Y')_{i,:} = \sum_{j\in\mathcal{V}}\left[S(\hat{A},X)\right]_{i,j}(Z-Y)_{j,:}$ and $\left[S(\hat{A},X)\right]_{i,j}$ can be considered as the weights. Intuitively, a high similarity value $\left[S(\hat{A},X)\right]_{i,j}$ means node $i$ tends to be updated to the same class as node $j$. This indicates that $S(\hat{A},X)$ is closely related to a single layer GNN model.

Based on the above definition and observation, we define the aggregation similarity score as follows.

**Definition 1.** *The aggregation similarity score is:*

$$\begin{aligned}&S_{agg}\big(S(\hat{A},X)\big)\\&= \frac{1}{|\mathcal{V}|}\left|\left\{v\,\big|\,\mathrm{Mean}_u\big(\{S(\hat{A},X)_{v,u}|Z_{u,:}=Z_{v,:}\}\big) \geq \mathrm{Mean}_u\big(\{S(\hat{A},X)_{v,u}|Z_{u,:}\neq Z_{v,:}\}\big)\right\}\right|\end{aligned} \qquad (7)$$

*where* $\mathrm{Mean}_u\left(\{\cdot\}\right)$ *takes the average over* $u$ *of a given multiset of values or variables.*

$S_{\mathrm{agg}}(S(\hat{A},X))$ measures the proportion of nodes $v \in \mathcal{V}$ as which the average weights on the set of nodes in the same class (including $v$) is larger than that in other classes. In practice, we observe that in most datasets, we will have $S_{\mathrm{agg}}(S(\hat{A},X)) \geq 0.5$ [4]. To make the metric range in [0,1], like existing metrics, we rescale equation 7 to the following modified aggregation similarity,

$$S_{\mathrm{agg}}^M\big(S(\hat{A},X)\big) = \big[2S_{\mathrm{agg}}\big(S(\hat{A},X)\big)-1\big]_+ \qquad (8)$$

In order to measure the consistency between labels and graph structures without considering node features and to make a fair comparison with the existing homophily metrics in equation 3, we define the graph ($\mathcal{G}$) aggregation ($\hat{A}$) homophily and its modified version [5] as:

$$H_{\mathrm{agg}}(\mathcal{G}) = S_{\mathrm{agg}}\big(S(\hat{A},Z)\big),\ \ H_{\mathrm{agg}}^M(\mathcal{G}) = S_{\mathrm{agg}}^M\big(S(\hat{A},Z)\big) \qquad (9)$$

As the example shown in Figure 1, when $\hat{A} = \hat{A}_{\mathrm{rw}}$, it is easy to see that $H_{\mathrm{agg}}(\mathcal{G}) = H_{\mathrm{agg}}^M(\mathcal{G}) = 1$ and other metrics are 0. Thus, this new metric reflects the fact that nodes in classes 1 and 2 are still highly distinguishable after aggregation, while other metrics mentioned before fail to capture such information and misleadingly give value 0. This shows the advantage of $H_{\mathrm{agg}}(\mathcal{G})$ and $H_{\mathrm{agg}}^M(\mathcal{G})$, which additionally exploit information from aggregation operator $\hat{A}$ and the similarity matrix.

To comprehensively compare $H_{\mathrm{agg}}^M(\mathcal{G})$ with the existing metrics on their ability to elucidate the influence of graph structure on GNN performance, we generate synthetic graphs with different homophily levels and evaluate SGC [42] and GCN [19] on them in the next subsection.

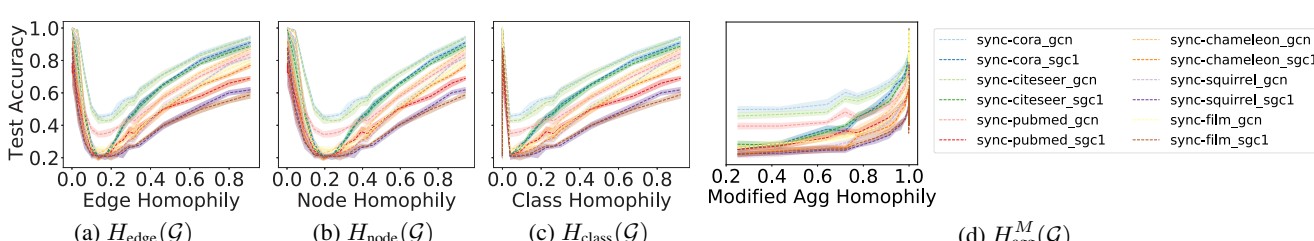

Figure 2: Comparison of baseline performance under different homophily metrics.

---

[4]See Appendix F.1 for an intuitive explanation under certain conditions.
[5]In practice, we will only check $H_{\mathrm{agg}}(\mathcal{G})$ when $H_{\mathrm{agg}}^M(\mathcal{G}) = 0$.

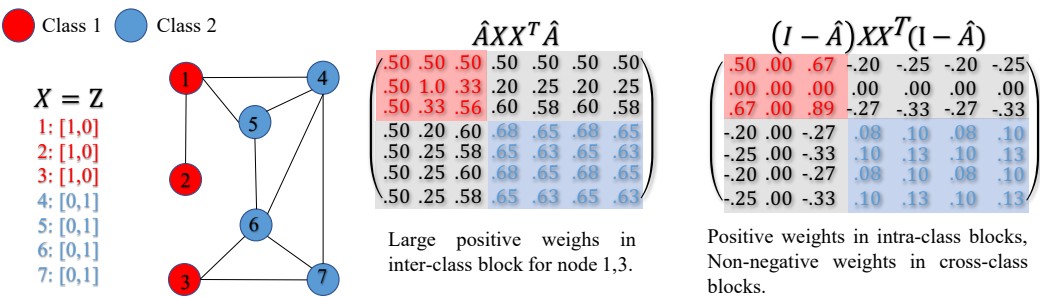

Figure 3: Example of how diversification can address harmful heterophily

## 3.2 Empirical Evaluation and Comparison on Synthetic Graphs

In this subsection, we conduct experiments on synthetic graphs generated with different levels of $H_{\text{edge}}^{M}(\mathcal{G})$ to assess the output of $H_{\text{agg}}^{M}(\mathcal{G})$ in comparison with existing metrics.

**Data Generation & Experimental Setup** We first generated 10 graphs for each of 28 edge homophily levels, from 0.005 to 0.95, for a total of 280 graphs. In every generated graph, we had 5 classes, with 400 nodes in each class. For nodes in each class, we randomly generated 800 intra-class edges and $[\frac{800}{H_{\text{edge}}(\mathcal{G})} - 800]$ inter-class edges. The features of nodes in each class are sampled from node features in the corresponding class of 6 base datasets (*Cora, CiteSeer, PubMed, Chameleon, Squirrel, Film*). Nodes were randomly split into train/validation/test sets, in proportion of 60%/20%/20%. We trained 1-hop SGC (*sgc-1*) [42] and GCN [19] on the synthetic graphs [6]. For each value of $H_{\text{edge}}(\mathcal{G})$, we take the average test accuracy and standard deviation of runs over the 10 generated graphs with that value. For each generated graph, we also calculate $H_{\text{node}}(\mathcal{G})$, $H_{\text{class}}(\mathcal{G})$ and $H_{\text{agg}}^{M}(\mathcal{G})$. Model performance with respect to different homophily values is shown in Figure 2.

**Comparison of Homophily Metrics** The performance of SGC-1 and GCN is expected to be monotonically increasing if the homophily metric is informative. However, Figure 2(a)(b)(c) show that the performance curves under $H_{\text{edge}}(\mathcal{G})$, $H_{\text{node}}(\mathcal{G})$ and $H_{\text{class}}(\mathcal{G})$ are $U$-shaped [7], while Figure 2(d) reveals a nearly monotonic curve with a little numerical perturbation around 1. This indicates that $H_{\text{agg}}^{M}(\mathcal{G})$ provides a better indication of the way in which the graph structure affects the performance of SGC-1 and GCN than existing metrics. (See more discussion on aggregation homophily and theoretical results for regular graphs in Appendix D.)

## 4 Adaptive Channel Mixing (ACM)

In prior work [32, 8, 4], it has been shown that high-frequency graph signals, which can be extracted by a high-pass filter (HP), is empirically useful for addressing heterophily. In this section, based on the similarity matrix in equation 6, we theoretically prove that a diversification operation, *i.e.,* HP filter, can address some cases of harmful heterophily locally. Besides, a node-wise analysis shows that different nodes may need different filters to process their neighborhood information. Based on the above analysis, in Sec. 4.2 we propose Adaptive Channel Mixing (ACM), a 3-channel architecture which can adaptively exploit local and node-wise information from aggregation, diversification and identity channels.

### 4.1 Diversification Helps with Harmful Heterophily

We first consider the example shown in Figure 3. From $S(\hat{A}, X)$, we can see that nodes $\{1, 3\}$ assign relatively large positive weights to nodes in class 2 after aggregation, which will make nodes $\{1, 3\}$ hard to be distinguished from nodes in class 2. However, we can still distinguish nodes $\{1, 3\}$ and $\{4, 5, 6, 7\}$ by considering their neighborhood differences: nodes $\{1, 3\}$ are different from most of their neighbors while nodes $\{4, 5, 6, 7\}$ are similar to most of their neighbors. This indicates that

---

[6]See Appendix C.1 for a description of the hyperparameter searching range and Appendix D for more a detailed description of the data generation process

[7]A similar J-shaped curve for $H_{\text{edge}}(\mathcal{G})$ is found in [46], though using different data generation processes. The authors do not mention the insufficiency of edge homophily.

although some nodes become similar after aggregation, they are still distinguishable through their local surrounding dissimilarities.

This observation leads us to introduce the *diversification operation*, *i.e.,* HP filter $I - \hat{A}$ [11] to extract information regarding neighborhood differences, thereby addressing harmful heterophily. As $S(I - \hat{A}, X)$ in Fig. 3 shows, nodes $\{1, 3\}$ will assign negative weights to nodes $\{4, 5, 6, 7\}$ after the diversification operation, *i.e.,* nodes 1,3 treat nodes 4,5,6,7 as negative samples and will move away from them during backpropagation. This example reveals that there are cases in which the diversification operation is helpful to handle heterophily, while the aggregation operation is not. Based on this observation, we first define the diversification distinguishability of a node and the graph diversification distinguishability value, which measures the proportion of nodes for which the diversification operation is potentially helpful.

**Definition 2** (Diversification Distinguishability (DD) based on $S(I - \hat{A}, X)$). *Given $S(I - \hat{A}, X)$, a node $v$ is diversification distinguishable if the following two conditions are satisfied at the same time,*

$$
\begin{aligned}
&\textbf{1. } \text{Mean}_u \left( \{ S(I - \hat{A}, X)_{v,u} | u \in \mathcal{V} \wedge Z_{u,:} = Z_{v,:} \} \right) \geq 0; \\
&\textbf{2. } \text{Mean}_u \left( \{ S(I - \hat{A}, X)_{v,u} | u \in \mathcal{V} \wedge Z_{u,:} \neq Z_{v,:} \} \right) \leq 0
\end{aligned}
\tag{10}
$$

*Then, graph diversification distinguishability value is defined as*

$$
\text{DD}_{\hat{A}, X}(\mathcal{G}) = \frac{1}{|\mathcal{V}|} \left| \{ v | v \in \mathcal{V} \wedge v \text{ is diversification distinguishable} \} \right|
\tag{11}
$$

We can see that $\text{DD}_{\hat{A}, X}(\mathcal{G}) \in [0, 1]$. Based on Def. 2, the effectiveness of diversification in addressing heterophily can be theoretically proved under certain conditions:

**Theorem 1.** (See Appendix G for proof). For $C = 2$, suppose $X = Z, \hat{A} = \hat{A}_{\text{rw}}$. Then for any $I - \hat{A}_{\text{rw}}$, all nodes are diversification distinguishable and $\text{DD}_{\hat{A}, Z}(\mathcal{G}) = 1$.

With the above results for HP filters, we will now introduce the concept of filterbank which combines both LP (aggregation) and HP (diversification) filters and can potentially handle various local heterophily cases. We then develop ACM framework in the following subsection.

### 4.2 Filterbank and Adaptive Channel Mixing (ACM) Framework

**Filterbank**     For the graph signal $x$ defined on $\mathcal{G}$, a 2-channel linear (analysis) filterbank [11] [8] includes a pair of filters $H_{\text{LP}}, H_{\text{HP}}$, which retain the low-frequency and high-frequency content of $x$, respectively. Most existing GNNs use a uni-channel filtering architecture [19, 41, 15] with either LP or HP channel, which only partially preserves the input information. Unlike the uni-channel architecture, filterbanks with $H_{\text{LP}} + H_{\text{HP}} = I$ do not lose any information from the input signal, which is called the perfect reconstruction property [11]. Generally, the Laplacian matrices ($L_{\text{sym}}, L_{\text{rw}}, \hat{L}_{\text{sym}}, \hat{L}_{\text{rw}}$) can be regarded as HP filters [11] and affinity matrices ($A_{\text{sym}}, A_{\text{rw}}, \hat{A}_{\text{sym}}, \hat{A}_{\text{rw}}$) can be treated as LP filters [34, 14]. Moreover, we extend the concept of filterbank and view MLPs as using the identity (full-pass) filterbank with $H_{\text{LP}} = I$ and $H_{\text{HP}} = 0$, which also satisfies $H_{\text{LP}} + H_{\text{HP}} = I + 0 = I$.

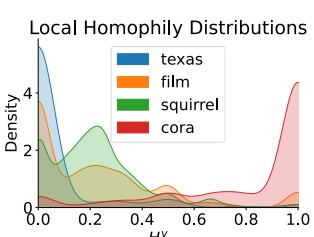

Figure 4: $H_{\text{node}}^v$ distributions

**Node-wise Channel Mixing for Diverse Local Homophily**     The example in Figure 3 also shows that different nodes may need the local information extracted from different channels, *e.g.,* nodes $\{1, 3\}$ demand information from the HP channel while node 2 only needs information from the LP channel. Figure 4 reveals that nodes have diverse distributions of node local homophily $H_{\text{node}}^v$ across different datasets. In order to adaptively leverage the LP, HP and identity channels in GNNs to deal with the diverse local heterophily situations, we will now describe our proposed Adaptive Channel Mixing (ACM) framework.

---

[8] In graph signal processing, an additional synthesis filter [11] is required to form the 2-channel filterbank. But a synthesis filter is not needed in our framework.

**Adaptive Channel Mixing (ACM)**   We will use GCN [9] as an example to introduce the ACM framework in matrix form, but the framework can be combined in a similar manner to many different GNNs. The ACM framework includes the following steps:

**Step 1. Feature Extraction for Each Channel:**

Option 1: $H_L^l = \mathrm{ReLU}\left(H_{\mathrm{LP}}H^{l-1}W_L^{l-1}\right), H_H^l = \mathrm{ReLU}\left(H_{\mathrm{HP}}H^{l-1}W_H^{l-1}\right), H_I^l = \mathrm{ReLU}\left(IH^{l-1}W_I^{l-1}\right);$

Option 2: $H_L^l = H_{\mathrm{LP}}\mathrm{ReLU}\left(H^{l-1}W_L^{l-1}\right), H_H^l = H_{\mathrm{HP}}\mathrm{ReLU}\left(H^{l-1}W_H^{l-1}\right), H_I^l = I\,\mathrm{ReLU}\left(H^{l-1}W_I^{l-1}\right);$

$H^0 = X \in \mathbb{R}^{N \times F_0},\ W_L^{l-1},\ W_H^{l-1},\ W_I^{l-1} \in \mathbb{R}^{F_{l-1} \times F_l},\ l = 1,\ldots,L;$

**Step 2. Row-wise Feature-based Weight Learning:**

$\tilde{\alpha}_L^l = \mathrm{Sigmoid}\left(H_L^l \tilde{W}_L^l\right),\ \tilde{\alpha}_H^l = \mathrm{Sigmoid}\left(H_H^l \tilde{W}_H^l\right),\ \tilde{\alpha}_I^l = \mathrm{Sigmoid}\left(H_I^l \tilde{W}_I^l\right),\ \tilde{W}_L^{l-1}, \tilde{W}_H^{l-1}, \tilde{W}_I^{l-1} \in \mathbb{R}^{F_l \times 1}$

$\left[\alpha_L^l, \alpha_H^l, \alpha_I^l\right] = \mathrm{Softmax}\left(\left(\left[\tilde{\alpha}_L^l, \tilde{\alpha}_H^l, \tilde{\alpha}_I^l\right]/T\right)W_{\mathrm{Mix}}^l\right) \in \mathbb{R}^{N \times 3}, T \in \mathbb{R}\ \text{temperature},\ W_{\mathrm{Mix}}^l \in \mathbb{R}^{3 \times 3};$

**Step 3. Node-wise Adaptive Channel Mixing:**

$H^l = \mathrm{ReLU}\left(\mathrm{diag}(\alpha_L^l)H_L^l + \mathrm{diag}(\alpha_H^l)H_H^l + \mathrm{diag}(\alpha_I^l)H_I^l\right)$

We will refer to the instantiation which uses option 1 in step 1 as ACM and to the one using option 2 as ACMII [10]. In step 1, ACM(II)-GCN implement different feature extractions for 3 channels using a set of filterbanks. Three filtered components, $H_L^l, H_H^l, H_I^l$, are obtained. To adaptively exploit information from each channel, ACM(II)-GCN first extract nonlinear information from the filtered signals, then use $W_{\mathrm{Mix}}^l$ to learn which channel is important for each node, leading to the row-wise weight vectors $\alpha_L^l, \alpha_H^l, \alpha_I^l \in \mathbb{R}^{N \times 1}$ whose $i$-th elements are the weights for node $i$ [11]. These three vectors are then used as weights in defining the updated $H^l$ in step 3.

**Complexity**   The number of learnable parameters in layer $l$ of ACM(II)-GCN is $3F_{l-1}(F_l + 1) + 9$, compared to $F_{l-1}F_l$ in GCN. The computation of steps 1-3 takes $NF_l(8 + 6F_{l-1}) + 2F_l(\mathrm{nnz}(H_{\mathrm{LP}}) + \mathrm{nnz}(H_{\mathrm{HP}})) + 18N$ flops, while the GCN layer takes $2NF_{l-1}F_l + 2F_l(\mathrm{nnz}(H_{\mathrm{LP}}))$ flops, where $\mathrm{nnz}(\cdot)$ is the number of non-zero elements. An ablation study and a detailed comparison on running time are conducted in Sec. 6.1.

**Limitations of Diversification**   Like any other method, there exists some cases of harmful heterophily that diversification operation cannot work well. For example, suppose we have an imbalanced dataset where several small clusters with distinctive labels are densely connected to a large cluster. In this case, the surrounding differences of nodes in small clusters are similar, *i.e.,* the neighborhood differences mainly come from their connections to the same large cluster, and this can lead to the diversification operation failing to discriminate them. See Appendix H for a more detailed discussion.

## 5   Related Work

We now discuss relevant work on addressing heterophily in GNNs. [1] acknowledges the difficulty of learning on graphs with weak homophily and propose MixHop to extract features from multi-hop neighborhoods to get more information. [17] propose measurements based on feature smoothness and label smoothness that are potentially helpful to guide GNNs when dealing with heterophilic graphs. Geom-GCN [36] precomputes unsupervised node embeddings and uses the graph structure defined by geometric relationships in the embedding space to define the bi-level aggregation process to handle heterophily. H$_2$GCN [46] combines 3 key designs to address heterophily: (1) ego- and neighbor-embedding separation; (2) higher-order neighborhoods; (3) combination of intermediate representations. CPGNN [45] models label correlations through a compatibility matrix, which is beneficial for heterophilic graphs, and propagates a prior belief estimation into the GNN by using the compatibility matrix. Non-local GNNs [28] propose a simple and effective non-local aggregation framework with an efficient attention-guided sorting for GNNs. FAGCN [4] learns edge-level aggregation weights as GAT [41] but allows the weights to be negative, which enables the network to capture high-frequency components in the graph signals. GPRGNN [8] uses learnable weights that can be both positive and negative for feature propagation. This allows GPRGNN to adapt to heterophilic graphs and to handle both high- and low-frequency parts of the graph signals (See Appendix J for a more comprehensive comparison between ACM-GNNs, ACMII-GNNs and FAGCN,

---

[9]See more variants in Appendix B.

[10]See Appendix B.1 for the reasons of having 2 options for ACM-GNNs.

[11]See Appendix A.4 and A.5 for more discussion of the components in ACM architecture.

GPRGNN). BernNet [16] designs a scheme to learn arbitrary graph spectral filters with Bernstein polynomial to address heterophily. [33] points out that homophily is not necessary for GNNs and characterizes conditions that GNNs can perform well on heterophilic graphs.

## 6   Empirical Evaluation

In this section, we evaluate the proposed ACM and ACMII framework on real-world datasets (see Appendix D.2 for a performance comparison with basline models on synthetic datasets). We first conduct ablation studies in Sec. 6.1 to validate the effectiveness and efficiency of different components of ACM and ACMII. Then, we compare with state-of-the-art (SOTA) models in Sec. 6.2. The hyperparameter searching range and computing resources are described in Appendix C.

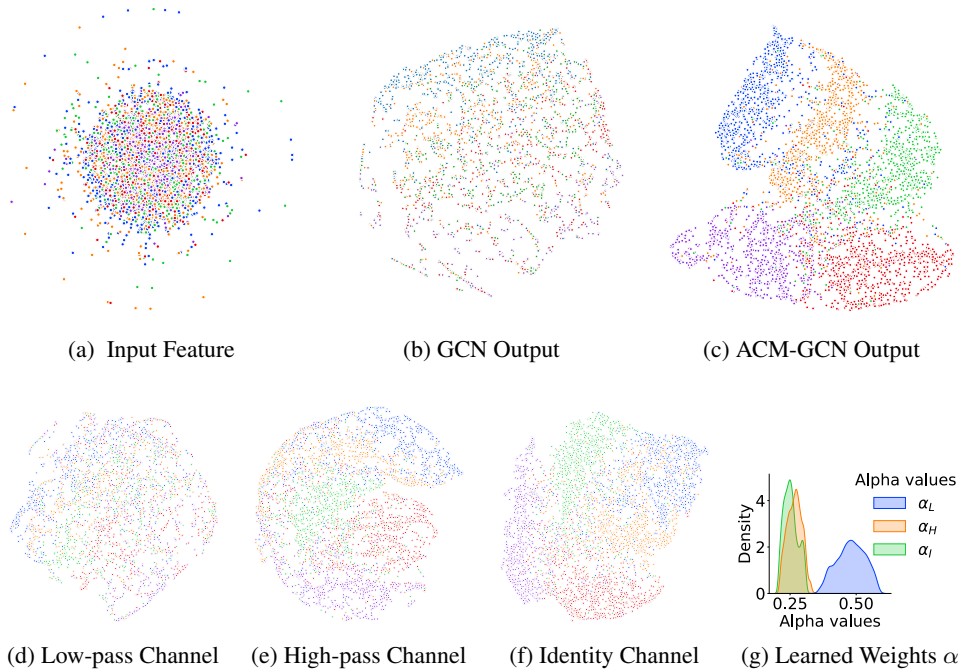

| (a)  Input Feature | (b) GCN Output | (c) ACM-GCN Output |
|---|---|---|

| (d) Low-pass Channel | (e) High-pass Channel | (f) Identity Channel | (g) Learned Weights $\alpha$ |
|---|---|---|---|

Figure 5: t-SNE visualization of the output layer of ACM-GCN and GCN trained on Squirrel

### 6.1   Ablation Study & Efficiency

We will now investigate the effectiveness and efficiency of adding HP, identity channels and the adaptive mixing mechanism in the proposed framework by performing an ablation study. Specifically, we apply the components of ACM to SGC-1 [42] [12] and the components of ACM and ACMII to GCN [19] separately. We run 10 times on each of the 9 benchmark datatsets, *Cornell*, *Wisconsin*, *Texas*, *Film*, *Chameleon*, *Squirrel*, *Cora*, *Citeseer* and *Pubmed* used in [37, 36], with the same 60%/20%/20% random splits for train/validation/test used in [8] and report the average test accuracy as well as the standard deviation. We also record the average running time per epoch (in milliseconds) to compare the computational efficiency. We set the temperature $T$ in equation 4.2 to be 3, which is the number of channels.

The results in Table 1 show that on most datasets, the additional HP and identity channels are helpful, even for strong homophily datasets such as *Cora, CiteSeer and PubMed*. The adaptive mixing mechanism also has an advantage over directly adding the three channels together. This illustrates the necessity of learning to customize the channel usage adaptively for different nodes. The t-SNE visualization in Figure 5 demonstrates that the high-pass channel(e) and identity channel(f) can extract meaningful patterns, which the low-pass channel(d) is not able to capture. The output of ACM-

---

[12]We only test ACM-SGC-1 because SGC-1 does not contain any non-linearity which makes ACM-SGC-1 and ACMII-SGC-1 exactly the same.

Ablation Study on Different Components in ACM-SGC and ACM-GCN (%)

| Baseline Models | LP | HP | Identity | Mixing | Cornell Acc ± Std | Wisconsin Acc ± Std | Texas Acc ± Std | Film Acc ± Std | Chameleon Acc ± Std | Squirrel Acc ± Std | Cora Acc ± Std | CiteSeer Acc ± Std | PubMed Acc ± Std | Rank |
|---|---|---|---|---|---|---|---|---|---|---|---|---|---|---|
| ACM-SGC-1 w/ | ✓ | | | | 70.98 ± 8.39 | 70.38 ± 2.85 | 83.28 ± 5.43 | 25.26 ± 1.18 | 64.86 ± 1.81 | 47.62 ± 1.27 | 85.12 ± 1.64 | 79.66 ± 0.75 | 85.5 ± 0.76 | 12.89 |
| | ✓ | ✓ | | ✓ | 83.28 ± 5.81 | 91.88 ± 1.61 | 90.98 ± 2.46 | 36.76 ± 1.01 | 65.27 ± 1.9 | 47.27 ± 1.37 | 86.8 ± 1.08 | 80.98 ± 1.68 | 87.21 ± 0.42 | 10.44 |
| | ✓ | | ✓ | ✓ | 93.93 ± 3.6 | 95.25 ± 1.84 | 93.93 ± 2.54 | 38.38 ± 1.13 | 63.83 ± 2.07 | 46.79 ± 0.75 | 86.73 ± 1.28 | 80.57 ± 0.99 | 87.8 ± 0.58 | 9.44 |
| | ✓ | ✓ | ✓ | | 88.2 ± 4.39 | 93.5 ± 2.95 | 92.95 ± 2.94 | 37.19 ± 0.87 | 62.82 ± 1.84 | 44.94 ± 0.93 | 85.22 ± 1.35 | 80.75 ± 1.68 | 88.11 ± 0.21 | 11.00 |
| | ✓ | ✓ | ✓ | ✓ | 93.77 ± 1.91 | 93.25 ± 2.92 | 93.61 ± 1.55 | 39.33 ± 1.25 | 63.68 ± 1.62 | 46.4 ± 1.13 | 86.63 ± 1.13 | 80.96 ± 0.93 | 87.75 ± 0.88 | 10.00 |
| ACM-GCN w/ | ✓ | | | | 82.46 ± 3.11 | 75.5 ± 2.92 | 83.11 ± 3.2 | 35.51 ± 0.99 | 64.18 ± 2.62 | 44.76 ± 1.39 | 87.78 ± 0.96 | 81.39 ± 1.23 | 88.9 ± 0.32 | 11.44 |
| | ✓ | ✓ | | ✓ | 82.13 ± 2.59 | 86.62 ± 4.61 | 89.19 ± 3.04 | 38.06 ± 1.35 | **69.21 ± 1.68** | 57.2 ± 1.01 | 88.93 ± 1.55 | **81.96 ± 0.91** | 90.01 ± 0.8 | 7.22 |
| | ✓ | | ✓ | ✓ | 94.26 ± 2.23 | 96.13 ± 2.2 | 94.1 ± 2.95 | 41.51 ± 0.99 | 67.44 ± 2.14 | 53.97 ± 1.39 | 88.95 ± 0.9 | 81.72 ± 1.22 | 90.88 ± 0.55 | 4.44 |
| | ✓ | ✓ | ✓ | | 91.64 ± 2 | 95.37 ± 3.31 | **95.25 ± 2.37** | 40.47 ± 1.49 | 68.93 ± 2.04 | 54.78 ± 1.27 | **89.13 ± 1.77** | **81.96 ± 2.03** | **91.01 ± 0.7** | 3.11 |
| | ✓ | ✓ | ✓ | ✓ | 94.75 ± 2.62 | **96.75 ± 1.6** | 95.08 ± 3.2 | 41.62 ± 1.15 | 69.04 ± 1.74 | **58.02 ± 1.86** | 88.95 ± 1.3 | 81.80 ± 1.26 | 90.69 ± 0.53 | 2.78 |
| ACMII-GCN w/ | ✓ | ✓ | | ✓ | 82.46 ± 3.03 | 91.00 ± 1.75 | 90.33 ± 2.69 | 38.39 ± 0.75 | 67.59 ± 2.14 | 53.67 ± 1.71 | **89.13 ± 1.14** | 81.75 ± 0.85 | 89.87 ± 0.39 | 7.44 |
| | ✓ | | ✓ | ✓ | 94.26 ± 2.57 | 96.00 ± 2.15 | 94.26 ± 2.96 | 40.96 ± 1.2 | 66.35 ± 1.76 | 50.78 ± 2.07 | 89.06 ± 1.07 | 81.86 ± 1.22 | 90.71 ± 0.67 | 4.67 |
| | ✓ | ✓ | ✓ | | 91.48 ± 1.43 | 96.25 ± 2.09 | 93.77 ± 2.91 | 40.27 ± 1.07 | 66.52 ± 2.65 | 52.9 ± 1.64 | 88.83 ± 1.16 | 81.54 ± 0.95 | 90.6 ± 0.47 | 6.67 |
| | ✓ | ✓ | ✓ | ✓ | **95.9 ± 1.83** | 96.62 ± 2.44 | 95.25 ± 3.15 | **41.84 ± 1.15** | 68.38 ± 1.36 | 54.53 ± 2.09 | 89.00 ± 0.72 | 81.79 ± 0.95 | 90.74 ± 0.5 | **2.78** |

Comparison of Average Running Time Per Epoch(ms)

| Baseline Models | LP | HP | Identity | Mixing | Cornell | Wisconsin | Texas | Film | Chameleon | Squirrel | Cora | CiteSeer | PubMed |
|---|---|---|---|---|---|---|---|---|---|---|---|---|---|
| ACM-SGC-1 w/ | ✓ | | | | 2.53 | 2.83 | 2.5 | 3.18 | 3.48 | 4.65 | 3.47 | 3.43 | 4.04 |
| | ✓ | ✓ | | ✓ | 4.01 | 4.57 | 4.24 | 4.55 | 4.76 | 5.09 | 5.39 | 4.69 | 4.75 |
| | ✓ | | ✓ | ✓ | 3.88 | 4.01 | 4.04 | 4.43 | 4.06 | 4.5 | 4.38 | 3.82 | 4.16 |
| | ✓ | ✓ | ✓ | | 3.31 | 3.49 | 3.18 | 3.7 | 3.53 | 4.83 | 3.92 | 3.87 | 4.24 |
| | ✓ | ✓ | ✓ | ✓ | 5.53 | 5.96 | 5.43 | 5.21 | 5.41 | 6.96 | 6 | 5.9 | 6.04 |
| ACM-GCN w/ | ✓ | | | | 3.67 | 3.74 | 3.59 | 4.86 | 4.96 | 6.41 | 4.24 | 4.18 | 5.08 |
| | ✓ | ✓ | | ✓ | 6.63 | 8.06 | 7.89 | 8.11 | 7.8 | 9.39 | 7.82 | 7.38 | 8.74 |
| | ✓ | | ✓ | ✓ | 5.73 | 5.91 | 5.93 | 6.86 | 6.35 | 7.15 | 7.34 | 6.65 | 6.8 |
| | ✓ | ✓ | ✓ | | 5.16 | 5.25 | 5.2 | 5.93 | 5.64 | 8.02 | 5.73 | 5.65 | 6.16 |
| | ✓ | ✓ | ✓ | ✓ | 8.25 | 8.11 | 7.89 | 7.97 | 8.41 | 11.9 | 8.84 | 8.38 | 8.63 |
| ACMII-GCN w/ | ✓ | ✓ | | ✓ | 6.62 | 7.35 | 7.39 | 7.62 | 7.33 | 9.69 | 7.49 | 7.58 | 7.97 |
| | ✓ | | ✓ | ✓ | 6.3 | 6.05 | 6.26 | 6.87 | 6.44 | 6.5 | 6.14 | 7.21 | 6.6 |
| | ✓ | ✓ | ✓ | | 5.24 | 5.27 | 5.46 | 5.72 | 5.65 | 7.87 | 5.48 | 5.65 | 6.33 |
| | ✓ | ✓ | ✓ | ✓ | 7.59 | 8.28 | 8.06 | 8.85 | 8 | 10 | 8.27 | 8.5 | 8.68 |

Table 1: Ablation study on 9 real-world datasets [36]. Cell with ✓ means the component is applied to the baseline model. The best test results are highlighted.

GCN(c) shows clearer boundaries among classes than GCN(b). The running time is approximately doubled in the ACM and ACMII framework compared to the original models.

## 6.2 Comparison with Baseline and SOTA Models

**Datasets & Experimental Setup** In this section, we evaluate SGC [42] with 1 hop and 2 hops (SGC-1, SGC-2), GCNII [7], GCNII* [7], GCN [19] and snowball networks [30] with 2 and 3 layers (snowball-2, snowball-3) and combine them with the ACM or ACMII framework[13]. We use $\hat{A}_{\mathrm{rw}}$ as the LP filter and the corresponding HP filter is $I - \hat{A}_{\mathrm{rw}}$[14]. Both filters are deterministic. We compare these approaches with several baselines and SOTA GNN models: MLP with 2 layers (MLP-2), GAT [41], APPNP [20], GPRGNN [8], $H_2$GCN [46], MixHop [1], GCN+JK [19, 43, 26], GAT+JK [41, 43, 26], FAGCN [4], GraphSAGE [15], Geom-GCN [36] and BernNet [16]. In addition to the 9 benchmark datasets used in section 6.1, we further test the above models on a new benchmark dataset, *Deezer-Europe* [38][15].

On each dataset used in [37, 36], we test the models 10 times following the same early stopping strategy, the same 60%/20%/20% random data split [16] and Adam [18] optimizer as used in GPRGNN [8]. For *Deezer-Europe*, we test the above models 5 times with the same early stopping strategy, the same fixed splits and Adam used in [26].

**Structure information channel and residual connection** Besides the filtered features, some recent SOTA models additionally use graph structure information, *i.e.,* $\mathrm{MLP}_\theta(A)$, and residual connection to address heterophily problem, *e.g.,* LINKX [25] and GloGNN [24]. $\mathrm{MLP}_\theta(A)$ and residual connection can be directly incorporated into ACM and ACMII framework, which leads us to ACM(II)-GCN+ and ACM(II)-GCN++. See the details of implementation in Appendix B.

---

[13]GCNII and GCNII* are hard to implement with the ACMII framework. See Appendix B for explanation.

[14]See Appendix A.3 for the comparison of $\hat{A}_{\mathrm{rw}}$ and $\hat{A}_{\mathrm{sym}}$.

[15]We choose *Deezer-Europe* because MLP outperforms GCN on it [26].

[16]See table 3 in Appendix A.2 for the performance comparison with several SOTA models, *e.g.,* LINKX [25] and GloGNN [24], on the fixed 48%/32%/20% splits provided by [36].

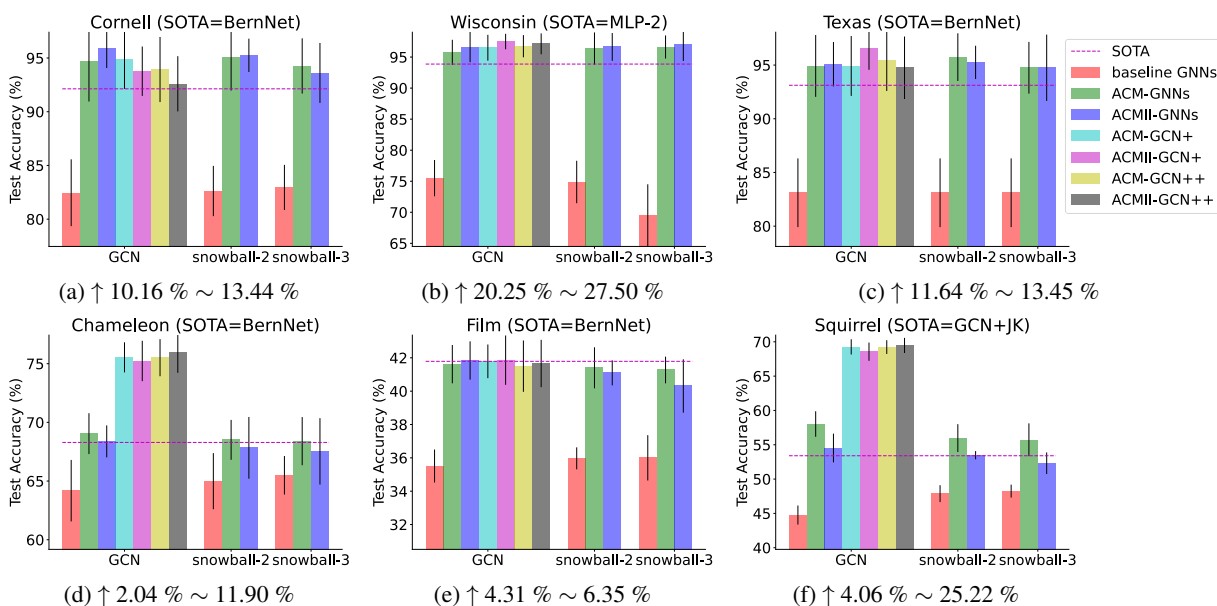

Figure 6: Comparison of baseline GNNs (red), ACM-GNNs (green), ACMII-GNNs (blue) with SOTA (magenta line) models on 6 selected datasets. The black lines indicate the standard deviation. The symbol "↑" shows the range of performance improvement (%) of ACM-GNNs and ACMII-GNNs over baseline GNNs. See Appendix I for a detailed discussion of the relation between $H_{\text{agg}}^M$ and GNN performance.

To visualize the performance, in Fig. 6, we plot the bar charts of the test accuracy of SOTA models, three selected baselines (GCN, snowball-2, snowball-3), their ACM(II) augmented models, ACM(II)-GCN+ and ACM(II)-GCN++ on the 6 most commonly used benchmark heterophily datasets (See Table 2 in Appendix A.1 for the full results, comparison and ranking). From Fig. 6, we can see that (1) after being combined with the ACM or ACMII framework, the performance of the three baseline models is **significantly boosted, by 2.04%∼27.50%** on all the 6 tasks. The ACM and ACMII in fact achieve SOTA performance. (2) On *Cornell, Wisconsin, Texas, Chameleon* and *Squirrel*, the augmented baseline models **significantly outperform the current SOTA models**. Overall, these results suggest that the proposed approach can help GNNs to generalize better on node classification tasks on heterophilic graphs, without adding too much computational cost.

## 7    Conclusions and Limitations

We have presented an analysis of existing homophily metrics and proposed new metrics which are more informative in terms of correlating with GNN performance. To our knowledge, this is the first work analyzing heterophily from the perspective of post-aggregation node similarity. The similarity matrix and the new metrics we defined mainly capture linear feature-independent relationships of each node. This might be insufficient when nonlinearity and feature-dependent information is important for classification. In the future, it would be useful to investigate if a similarity matrix could be defined which is capable of capturing nonlinear and feature-dependent relations between aggregated node.

We have also proposed a multi-channel mixing mechanism which leverages the intuitions gained in the first part of the paper and can be combined with different GNN architectures, enabling adaptive filtering (high-pass, low-pass or identity) at different nodes. Empirically, this approach shows very promising results, improving the performance of the base GNNs with which it is combined and achieving SOTA results at the cost of a reasonable increase in computation time. As discussed in Sec. 4.2, however, the filterbank method cannot properly handle all cases of harmful heterophily, and alternative ideas should be explored as well in the future.

## 8    Acknowledge

The authors would like to give very special thanks to William L. Hamilton for valuable discussion and advice. The project was partially supported by DeepMind and NSERC.

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
