# A  More Experimental Results

## A.1  Comparison with SOTA Models on 60%/20%/20% Random Splits

The main results of the full sets of experiments [17] with statistics of datasets are summarized in Table 2, where we report the mean accuracy (%) and standard deviation. We can see that after applied in ACM or ACMII framework, the performance of baseline models are boosted on almost all tasks and achieve SOTA performance on 9 out of 10 datasets. Especially, ACMII-GCN+ performs the best in terms of average rank (4.40) across all datasets. Overall, It suggests that ACM or ACMII framework can significantly increase the performance of GNNs on node classification tasks on heterophilic graphs and maintain highly competitive performance on homophilic datasets.

|  | Cornell | Wisconsin | Texas | Film | Chameleon | Squirrel | Deezer-Europe | Cora | CiteSeer | PubMed |  |
|---|---|---|---|---|---|---|---|---|---|---|---|
| #nodes | 183 | 251 | 183 | 7,600 | 2,277 | 5,201 | 28,281 | 2,708 | 3,327 | 19,717 | |
| #edges | 295 | 499 | 309 | 33,544 | 36,101 | 217,073 | 92,752 | 5,429 | 4,732 | 44,338 | |
| #features | 1,703 | 1,703 | 1,703 | 931 | 2,325 | 2,089 | 31,241 | 1,433 | 3,703 | 500 | |
| #classes | 5 | 5 | 5 | 5 | 5 | 5 | 2 | 7 | 6 | 3 | |
| $H_{edge}$ | 0.5669 | 0.4480 | 0.4106 | 0.3750 | 0.2795 | 0.2416 | 0.5251 | 0.8100 | 0.7362 | 0.8024 | |
| $H_{node}$ | 0.3855 | 0.1498 | 0.0968 | 0.2210 | 0.2470 | 0.2156 | 0.5299 | 0.8252 | 0.7175 | 0.7924 | |
| $H_{class}$ | 0.0468 | 0.0941 | 0.0013 | 0.0110 | 0.0620 | 0.0254 | 0.0304 | 0.7657 | 0.6270 | 0.6641 | |
| Data Splits | 60%/20%/20% | 60%/20%/20% | 60%/20%/20% | 60%/20%/20% | 60%/20%/20% | 60%/20%/20% | 60%/20%/20% | 50%/25%/25% | 60%/20%/20% | 60%/20%/20% | |
| $H_{agg}^{M}(G)$ | 0.8032 | 0.7768 | 0.694 | 0.6822 | 0.61 | 0.3566 | 0.5790 | 0.9904 | 0.9826 | 0.9432 | |

| | Test Accuracy (%) of State-of-the-art Models, Baseline GNN Models and ACM-GNN models | | | | | | | | | | Rank |
|---|---|---|---|---|---|---|---|---|---|---|---|
| MLP-2 | 91.30 ± 0.70 | 93.87 ± 3.33 | 92.26 ± 0.71 | 38.58 ± 0.25 | 46.72 ± 0.46 | 31.28 ± 0.27 | 66.55 ± 0.72 | 76.44 ± 0.30 | 76.25 ± 0.28 | 86.43 ± 0.13 | 23.40 |
| GAT | 76.00 ± 1.01 | 71.01 ± 4.66 | 78.87 ± 0.86 | 35.98 ± 0.23 | 63.9 ± 0.46 | 42.72 ± 0.33 | 61.09 ± 0.77 | 76.70 ± 0.42 | 67.20 ± 0.46 | 83.28 ± 0.12 | 26.20 |
| APPNP | 91.80 ± 0.63 | 92.00 ± 3.59 | 91.18 ± 0.70 | 38.86 ± 0.24 | 51.91 ± 0.56 | 34.77 ± 0.34 | 67.21 ± 0.56 | 79.41 ± 0.38 | 68.59 ± 0.30 | 85.02 ± 0.09 | 22.80 |
| GPRGNN | 91.36 ± 0.70 | 93.75 ± 2.37 | 92.92 ± 0.61 | 39.30 ± 0.27 | 67.48 ± 0.40 | 49.93 ± 0.53 | 66.90 ± 0.50 | 79.51 ± 0.36 | 67.63 ± 0.38 | 85.07 ± 0.09 | 19.20 |
| H2GCN | 86.23 ± 4.71 | 87.5 ± 1.77 | 85.90 ± 3.53 | 38.85 ± 1.17 | 52.30 ± 0.48 | 30.39 ± 1.22 | 67.22 ± 0.90 | 87.52 ± 0.61 | 79.97 ± 0.69 | 87.78 ± 0.28 | 21.80 |
| MixHop | 60.33 ± 28.53 | 77.25 ± 7.80 | 76.39 ± 7.66 | 33.13 ± 2.40 | 36.28 ± 10.22 | 24.55 ± 2.60 | 66.80 ± 0.58 | 65.65 ± 11.31 | 49.52 ± 13.35 | 87.04 ± 4.10 | 28.30 |
| GCN+JK | 66.56 ± 13.82 | 62.50 ± 15.75 | 80.66 ± 1.91 | 32.72 ± 2.62 | 64.68 ± 2.85 | 53.40 ± 1.90 | 60.99 ± 0.14 | 86.90 ± 1.51 | 73.77 ± 1.85 | 90.09 ± 0.68 | 23.40 |
| GAT+JK | 74.43 ± 10.24 | 69.50 ± 3.12 | 75.41 ± 7.18 | 35.41 ± 0.97 | 68.14 ± 1.18 | 52.28 ± 3.61 | 59.66 ± 0.92 | 89.52 ± 0.43 | 74.49 ± 2.76 | 89.15 ± 0.87 | 20.90 |
| FAGCN | 88.03 ± 5.6 | 89.75 ± 6.37 | 88.85 ± 4.39 | 31.59 ± 1.37 | 49.47 ± 2.84 | 42.24 ± 1.2 | 66.86 ± 0.53 | 88.85 ± 1.36 | 82.37 ± 1.46 | 89.98 ± 0.54 | 18.20 |
| BernNet | 92.13 ± 1.64 | NA | 93.12 ± 0.65 | 41.79 ± 1.01 | 68.29 ± 1.58 | 51.35 ± 0.73 | NA | 88.52 ± 0.95 | 80.09 ± 0.79 | 88.48 ± 0.41 | 14.75 |
| GraphSAGE | 71.41 ± 1.24 | 64.85 ± 5.14 | 79.03 ± 1.20 | 36.37 ± 0.21 | 62.15 ± 0.42 | 41.26 ± 0.26 | OOM | 86.58 ± 0.26 | 78.24 ± 0.30 | 86.85 ± 0.11 | 25.78 |
| Geom-GCN* | 60.81 | 64.12 | 67.57 | 31.63 | 60.9 | 38.14 | NA | 85.27 | 77.99 | 90.05 | 27.44 |
| SGC-1 | 70.98 ± 8.39 | 70.38 ± 2.85 | 83.28 ± 5.43 | 25.26 ± 1.18 | 64.86 ± 1.81 | 47.62 ± 1.27 | 59.73 ± 0.12 | 85.12 ± 1.64 | 79.66 ± 0.75 | 85.5 ± 0.76 | 24.90 |
| SGC-2 | 72.62 ± 9.92 | 74.75 ± 2.89 | 81.31 ± 3.3 | 28.81 ± 1.11 | 62.67 ± 2.41 | 41.25 ± 1.4 | 61.56 ± 0.51 | 85.48 ± 1.48 | 80.75 ± 1.15 | 85.36 ± 0.52 | 25.40 |
| GCNII | 89.18 ± 3.96 | 83.25 ± 2.69 | 82.46 ± 4.58 | 40.82 ± 1.79 | 60.35 ± 2.7 | 38.81 ± 1.97 | 66.38 ± 0.45 | 88.98 ± 1.33 | 81.58 ± 1.3 | 89.8 ± 0.3 | 19.30 |
| GCNII* | 90.49 ± 4.45 | 89.12 ± 3.06 | 88.52 ± 3.02 | 41.54 ± 0.99 | 62.8 ± 2.87 | 38.31 ± 1.3 | 66.42 ± 0.56 | 88.93 ± 1.37 | 81.83 ± 1.78 | 89.98 ± 0.52 | 16.40 |
| GCN | 82.46 ± 3.11 | 75.5 ± 2.92 | 83.11 ± 3.2 | 35.51 ± 0.99 | 64.18 ± 2.62 | 44.76 ± 1.39 | 62.23 ± 0.53 | 87.78 ± 0.96 | 81.39 ± 1.23 | 88.9 ± 0.32 | 20.90 |
| Snowball-2 | 82.62 ± 2.34 | 74.88 ± 3.42 | 83.11 ± 3.2 | 35.97 ± 0.66 | 64.99 ± 2.39 | 47.88 ± 1.23 | OOM | 88.64 ± 1.15 | 81.53 ± 1.71 | 89.04 ± 0.49 | 19.78 |
| Snowball-3 | 82.95 ± 2.1 | 69.5 ± 5.01 | 83.11 ± 3.2 | 36.00 ± 1.36 | 65.49 ± 1.64 | 48.25 ± 0.94 | OOM | 89.33 ± 1.3 | 80.93 ± 1.32 | 88.8 ± 0.82 | 19.11 |
| ACM-SGC-1 | 93.77 ± 1.91 | 93.25 ± 2.92 | 93.61 ± 1.55 | 39.33 ± 1.25 | 63.68 ± 1.62 | 46.4 ± 1.13 | 66.67 ± 0.56 | 86.63 ± 1.13 | 80.96 ± 0.93 | 87.75 ± 0.88 | 17.00 |
| ACM-SGC-2 | 93.77 ± 2.17 | 94.00 ± 2.61 | 93.44 ± 2.54 | 40.13 ± 1.21 | 60.48 ± 1.55 | 40.91 ± 1.39 | 66.53 ± 0.57 | 87.64 ± 0.99 | 80.93 ± 1.16 | 88.79 ± 0.5 | 17.70 |
| ACM-GCNII | 92.62 ± 3.13 | 94.63 ± 2.96 | 92.46 ± 1.97 | 41.37 ± 1.37 | 58.73 ± 2.52 | 40.9 ± 1.58 | 66.39 ± 0.56 | 89.1 ± 1.61 | 82.28 ± 1.12 | 90.12 ± 0.4 | 14.30 |
| ACM-GCNII* | 93.44 ± 2.74 | 94.37 ± 2.81 | 93.28 ± 2.79 | 41.27 ± 1.24 | 61.66 ± 2.29 | 38.32 ± 1.5 | 66.6 ± 0.57 | 89.00 ± 1.35 | 81.69 ± 1.25 | 90.18 ± 0.51 | 14.20 |
| ACM-GCN | 94.75 ± 3.8 | 95.75 ± 2.03 | 94.92 ± 2.88 | 41.62 ± 1.15 | 69.04 ± 1.74 | 58.02 ± 1.86 | 67.01 ± 0.38 | 88.62 ± 1.22 | 81.68 ± 0.97 | 90.66 ± 0.47 | 7.90 |
| ACM-GCN+ | 94.92 ± 2.79 | 96.5 ± 2.08 | 94.92 ± 2.79 | 41.79 ± 1.01 | 76.08 ± 2.13 | 69.26 ± 1.11 | 67.4 ± 0.44 | 89.75 ± 1.16 | 81.65 ± 1.48 | 90.46 ± 0.69 | 4.90 |
| ACM-GCN++ | 93.93 ± 1.05 | 97.5 ± 1.25 | 96.56 ± 2 | 41.86 ± 1.48 | 75.23 ± 1.72 | 68.56 ± 1.33 | 67.3 ± 0.48 | 89.33 ± 0.81 | 81.83 ± 1.65 | 90.39 ± 0.33 | 4.30 |
| ACM-Snowball-2 | 95.08 ± 3.11 | 96.38 ± 2.59 | 95.74 ± 2.22 | 41.4 ± 1.23 | 68.51 ± 1.7 | 55.97 ± 2.03 | OOM | 88.83 ± 1.49 | 81.58 ± 1.23 | 90.81 ± 0.52 | 7.44 |
| ACM-Snowball-3 | 94.26 ± 2.57 | 96.62 ± 1.86 | 94.75 ± 2.41 | 41.27 ± 0.8 | 68.4 ± 2.05 | 55.73 ± 2.39 | OOM | 89.59 ± 1.58 | 81.32 ± 0.97 | 91.44 ± 0.59 | 7.22 |
| ACMII-GCN | 95.9 ± 1.83 | 96.62 ± 2.44 | 95.08 ± 2.07 | 41.84 ± 1.15 | 68.38 ± 1.36 | 54.53 ± 2.09 | 67.15 ± 0.41 | 89.00 ± 0.72 | 81.79 ± 0.95 | 90.74 ± 0.5 | 5.90 |
| ACMII-Snowball-2 | 95.25 ± 1.55 | 96.63 ± 2.24 | 95.25 ± 1.55 | 41.1 ± 0.75 | 67.83 ± 2.63 | 53.48 ± 0.6 | OOM | 88.95 ± 1.04 | 82.07 ± 1.04 | 90.56 ± 0.39 | 7.56 |
| ACMII-Snowball-3 | 93.61 ± 2.79 | 97.00 ± 2.63 | 94.75 ± 3.09 | 40.31 ± 1.6 | 67.53 ± 2.83 | 52.31 ± 1.57 | OOM | 89.36 ± 1.26 | 81.56 ± 1.15 | 91.31 ± 0.6 | 9.00 |
| ACMII-GCN+ | 93.93 ± 3.03 | 96.75 ± 1.79 | 95.41 ± 2.82 | 41.5 ± 1.54 | 75.51 ± 1.58 | 69.81 ± 1.11 | 67.44 ± 0.31 | 89.18 ± 1.11 | 81.87 ± 1.38 | 90.96 ± 0.62 | 4.4 |
| ACMII-GCN++ | 92.62 ± 2.57 | 97.13 ± 1.68 | 94.75 ± 2.91 | 41.66 ± 1.42 | 75.93 ± 1.71 | 69.98 ± 1.53 | 67.5 ± 0.53 | 89.47 ± 1.08 | 81.76 ± 1.25 | 90.63 ± 0.56 | 5.10 |

Table 2: Experimental results: average test accuracy ± standard deviation on 10 real-world benchmark datasets. The best results are highlighted in grey and the best baseline results (SOTA in Figure 6) are underlined. Results "*" are reported from [8, 26] and results "†" are from [36]. NA means the reported results are not available and OOM means out of memory.

---

[17] The splits for all these experiments are random 60%/20%/20% splits for train/valid/test. The open source code we use is from https://github.com/jianhao2016/GPRGNN/blob/f4aaad6ca28c83d3121338a4c4fe5d162edfa9a2/src/utils.py#L16. See table 3 in Appendix A.2 for the performance comparison with several SOTA models on the fixed 48%/32%/20% splits provided by [36].

## A.2 Comparison with SOTA Models on Fixed 48%/32%/20% Splits

See table 3 for the results and table 13 14 the optimal searched hyperparameters. The results and comparison give us the same conclusion as in Appendix A.1.

| Datasets/Models | Cornell | Wisconsin | Texas | Film | Chameleon | Squirrel | Cora | Citeseer | PubMed | Average Rank |
|---|---|---|---|---|---|---|---|---|---|---|
| Geom-GCN | $60.54 \pm 3.67$ | $64.51 \pm 3.66$ | $66.76 \pm 2.72$ | $31.59 \pm 1.15$ | $60.00 \pm 2.81$ | $38.15 \pm 0.92$ | $85.35 \pm 1.57$ | $\mathbf{78.02 \pm 1.15}$ | $89.95 \pm 0.47$ | 18.22 |
| H2GCN | $82.70 \pm 5.28$ | $87.65 \pm 4.98$ | $84.86 \pm 7.23$ | $35.70 \pm 1.00$ | $60.11 \pm 2.15$ | $36.48 \pm 1.86$ | $87.87 \pm 1.20$ | $77.11 \pm 1.57$ | $89.49 \pm 0.38$ | 15.11 |
| GPRGCN | $78.11 \pm 6.55$ | $82.55 \pm 6.23$ | $81.35 \pm 5.32$ | $35.16 \pm 0.9$ | $62.59 \pm 2.04$ | $46.31 \pm 2.46$ | $87.95 \pm 1.18$ | $77.13 \pm 1.67$ | $87.54 \pm 0.38$ | 17.67 |
| FAGCN | $76.76 \pm 5.87$ | $79.61 \pm 1.58$ | $76.49 \pm 2.87$ | $34.82 \pm 1.35$ | $46.07 \pm 2.11$ | $30.83 \pm 0.69$ | $88.05 \pm 1.57$ | $77.07 \pm 2.05$ | $88.09 \pm 1.38$ | 20.00 |
| GCNII | $77.86 \pm 3.79$ | $80.39 \pm 3.40$ | $77.57 \pm 3.83$ | $37.44 \pm 1.30$ | $63.86 \pm 3.04$ | $38.47 \pm 1.58$ | $\mathbf{88.37 \pm 1.25}$ | $77.33 \pm 1.48$ | $\mathbf{90.15 \pm 0.43}$ | 12.44 |
| MixHop | $73.51 \pm 6.34$ | $75.88 \pm 4.90$ | $77.84 \pm 7.73$ | $32.22 \pm 2.34$ | $60.50 \pm 2.53$ | $43.80 \pm 1.48$ | $87.61 \pm 0.85$ | $76.26 \pm 1.33$ | $85.31 \pm 0.61$ | 20.78 |
| WRGAT | $81.62 \pm 3.90$ | $86.98 \pm 3.78$ | $83.62 \pm 5.50$ | $36.53 \pm 0.77$ | $65.24 \pm 0.87$ | $48.85 \pm 0.78$ | $88.20 \pm 2.26$ | $76.81 \pm 1.89$ | $88.52 \pm 0.92$ | 14.33 |
| GGCN | $85.68 \pm 6.63$ | $86.86 \pm 3.29$ | $84.86 \pm 4.55$ | $37.54 \pm 1.56$ | $71.14 \pm 1.84$ | $55.17 \pm 1.58$ | $87.95 \pm 1.05$ | $77.14 \pm 1.45$ | $89.15 \pm 0.37$ | 10.22 |
| LINKX | $77.84 \pm 5.81$ | $75.49 \pm 5.72$ | $74.60 \pm 8.37$ | $36.10 \pm 1.55$ | $68.42 \pm 1.38$ | $61.81 \pm 1.80$ | $84.64 \pm 1.13$ | $73.19 \pm 0.99$ | $87.86 \pm 0.77$ | 18.78 |
| GloGNN | $83.51 \pm 4.26$ | $87.06 \pm 3.53$ | $84.32 \pm 4.15$ | $37.35 \pm 1.30$ | $69.78 \pm 2.42$ | $57.54 \pm 1.39$ | $88.31 \pm 1.13$ | $77.41 \pm 1.65$ | $89.62 \pm 0.35$ | 8.78 |
| GloGNN++ | $85.95 \pm 5.10$ | $88.04 \pm 3.22$ | $84.05 \pm 4.90$ | $37.70 \pm 1.40$ | $71.21 \pm 1.84$ | $57.88 \pm 1.76$ | $88.33 \pm 1.09$ | $77.22 \pm 1.78$ | $89.24 \pm 0.39$ | 7.33 |
| ACM-SGC-1 | $82.43 \pm 5.44$ | $86.47 \pm 3.77$ | $81.89 \pm 4.53$ | $35.49 \pm 1.06$ | $63.99 \pm 1.66$ | $45.00 \pm 1.4$ | $86.9 \pm 1.38$ | $76.73 \pm 1.59$ | $88.49 \pm 0.51$ | 17.56 |
| ACM-SGC-2 | $82.43 \pm 5.44$ | $86.47 \pm 3.77$ | $81.89 \pm 4.53$ | $36.04 \pm 0.83$ | $59.21 \pm 2.22$ | $40.02 \pm 0.96$ | $87.69 \pm 1.07$ | $76.59 \pm 1.69$ | $89.01 \pm 0.6$ | 17.67 |
| Diag-NSD | $\mathbf{86.49 \pm 7.35}$ | $88.63 \pm 2.75$ | $85.67 \pm 6.95$ | $37.79 \pm 1.01$ | $68.68 \pm 1.73$ | $54.78 \pm 1.81$ | $87.14 \pm 1.06$ | $77.14 \pm 1.85$ | $89.42 \pm 0.43$ | 9.00 |
| O(d)-NSD | $84.86 \pm 4.71$ | $\mathbf{89.41 \pm 4.74}$ | $85.95 \pm 5.51$ | $37.81 \pm 1.15$ | $68.04 \pm 1.58$ | $56.34 \pm 1.32$ | $86.90 \pm 1.13$ | $76.70 \pm 1.57$ | $89.49 \pm 0.40$ | 10.44 |
| Gen-NSD | $85.68 \pm 6.51$ | $89.21 \pm 3.84$ | $82.97 \pm 5.13$ | $37.80 \pm 1.22$ | $67.93 \pm 1.58$ | $53.17 \pm 1.31$ | $87.30 \pm 1.15$ | $76.32 \pm 1.65$ | $89.33 \pm 0.35$ | 11.67 |
| NLMLP | $84.9 \pm 5.7$ | $87.3 \pm 4.3$ | $85.4 \pm 3.8$ | $\mathbf{37.9 \pm 1.4}$ | $50.7 \pm 2.2$ | $33.7 \pm 1.5$ | $76.9 \pm 1.8$ | $73.4 \pm 1.9$ | $88.2 \pm 0.5$ | 16.67 |
| NLGCN | $57.6 \pm 5.5$ | $60.2 \pm 5.3$ | $65.5 \pm 6.6$ | $31.6 \pm 1.0$ | $70.1 \pm 2.9$ | $59.0 \pm 1.2$ | $88.1 \pm 1.0$ | $75.2 \pm 1.4$ | $89.0 \pm 0.5$ | 17.44 |
| NLGAT | $54.7 \pm 7.6$ | $56.9 \pm 7.3$ | $62.6 \pm 7.1$ | $29.5 \pm 1.3$ | $65.7 \pm 1.4$ | $56.8 \pm 2.5$ | $88.5 \pm 1.8$ | $76.2 \pm 1.6$ | $88.2 \pm 0.3$ | 18.56 |
| ACM-GCN | $85.14 \pm 6.07$ | $88.43 \pm 3.22$ | $87.84 \pm 4.4$ | $36.63 \pm 0.84$ | $69.14 \pm 1.91$ | $55.19 \pm 1.49$ | $87.91 \pm 0.95$ | $77.32 \pm 1.7$ | $90.00 \pm 0.52$ | 8.11 |
| ACMII-GCN | $85.95 \pm 5.64$ | $87.45 \pm 3.74$ | $86.76 \pm 4.75$ | $36.31 \pm 1.2$ | $68.46 \pm 1.7$ | $51.8 \pm 1.5$ | $88.01 \pm 1.08$ | $77.15 \pm 1.45$ | $89.89 \pm 0.43$ | 9.33 |
| ACM-GCN+ | $85.68 \pm 4.84$ | $88.43 \pm 2.39$ | $\mathbf{88.38 \pm 3.64}$ | $36.26 \pm 1.34$ | $74.47 \pm 1.84$ | $66.98 \pm 1.71$ | $88.05 \pm 0.99$ | $77.67 \pm 1.19$ | $89.82 \pm 0.41$ | 5.33 |
| ACMII-GCN+ | $85.41 \pm 5.3$ | $88.04 \pm 3.66$ | $88.11 \pm 3.24$ | $36.14 \pm 1.44$ | $74.56 \pm 2.08$ | $67.07 \pm 1.65$ | $88.19 \pm 1.17$ | $77.2 \pm 1.61$ | $89.78 \pm 0.49$ | 6.78 |
| ACM-GCN++ | $85.68 \pm 5.8$ | $88.24 \pm 3.16$ | $\mathbf{88.38 \pm 3.43}$ | $37.31 \pm 1.09$ | $74.41 \pm 1.49$ | $67.06 \pm 1.66$ | $88.11 \pm 0.96$ | $77.46 \pm 1.65$ | $89.65 \pm 0.58$ | 5.33 |
| ACMII-GCN++ | $\mathbf{86.49 \pm 6.73}$ | $88.43 \pm 3.66$ | $\mathbf{88.38 \pm 3.43}$ | $37.09 \pm 1.32$ | $\mathbf{74.76 \pm 2.2}$ | $67.4 \pm 2.21$ | $88.25 \pm 0.96$ | $77.12 \pm 1.58$ | $89.71 \pm 0.48$ | $\mathbf{4.78}$ |

Table 3: Experimental results on fixed splits provided by [36]: average test accuracy $\pm$ standard deviation on 9 real-world benchmark datasets. The best results are highlighted. Results of Geom-GCN, H$_2$GCN and GPRGNN, LINX, GloGNN, GloGNN++, Diag-NSD, O(d)-NSD, Gen-NSD, NLMLP, NLGCN and NLGAT are from [36, 46, 27, 26, 24, 5, 29]; results on the rest models are run by ourselves and the hyperparameter searching range is the same as table 9.

## A.3 Discussion of Random Walk and Symmetric Renormalized Filters

| Datasets/Models | RW | | Symmetric | |
|---|---|---|---|---|
| | ACM | ACMII | ACM | ACMII |
| Cornell | $94.75 \pm 3.8$ | $\mathbf{95.9 \pm 1.83}$ | $94.92 \pm 2.48$ | $94.1 \pm 2.56$ |
| Wisconsin | $95.75 \pm 2.03$ | $\mathbf{96.62 \pm 2.44}$ | $95.63 \pm 2.81$ | $96.25 \pm 2.5$ |
| Texas | $94.92 \pm 2.88$ | $\mathbf{95.08 \pm 2.07}$ | $94.75 \pm 2.01$ | $94.59 \pm 2.65$ |
| Film | $41.62 \pm 1.15$ | $\mathbf{41.84 \pm 1.15}$ | $41.58 \pm 1.3$ | $41.65 \pm 0.6$ |
| Chameleon | $\mathbf{69.04 \pm 1.74}$ | $68.38 \pm 1.36$ | $67.9 \pm 2.76$ | $68.03 \pm 1.68$ |
| Squirrel | $\mathbf{58.02 \pm 1.86}$ | $54.53 \pm 2.09$ | $54.18 \pm 1.35$ | $53.68 \pm 1.74$ |
| Cora | $88.62 \pm 1.22$ | $\mathbf{89.00 \pm 0.72}$ | $88.65 \pm 1.26$ | $88.19 \pm 1.38$ |
| Citeseer | $81.68 \pm 0.97$ | $81.79 \pm 0.95$ | $\mathbf{81.84 \pm 1.15}$ | $81.81 \pm 0.86$ |
| PubMed | $90.66 \pm 0.47$ | $\mathbf{90.74 \pm 0.5}$ | $90.59 \pm 0.81$ | $90.54 \pm 0.59$ |

Table 4: Comparison of random walk and symmetric renormalized filters

The definitions of the similarity matrix, (modified) aggregation similarity score and diversification distinguishability value can be extended to symmetric normalized Laplacian or other aggregation operations. Yet unfortunately, we cannot extend Theorem 1 at this moment, because we need a condition that the row sum of $\hat{A}$ is not greater than 1 in the proof. This condition is guaranteed for random walk normalized Laplacian but not for symmetric normalized Laplacian. While in practice, we evaluate our models with symmetric filters and compare them with random walk filters. From table 4 we can see that, there are no big differences between these two filters.

| Datasets/Models | With $W_{\text{mix}}$ | | Without $W_{\text{mix}}$ | |
|---|---|---|---|---|
| | ACM | ACMII | ACM | ACMII |
| Cornell | $94.75 \pm 3.8$ | $\mathbf{95.9 \pm 1.83}$ | $93.61 \pm 2.37$ | $90.49 \pm 2.72$ |
| Wisconsin | $95.75 \pm 2.03$ | $96.62 \pm 2.44$ | $95 \pm 2.5$ | $\mathbf{97.50 \pm 1.25}$ |
| Texas | $94.92 \pm 2.88$ | $\mathbf{95.08 \pm 2.07}$ | $94.92 \pm 2.79$ | $94.92 \pm 2.79$ |
| Film | $41.62 \pm 1.15$ | $\mathbf{41.84 \pm 1.15}$ | $40.79 \pm 1.01$ | $40.86 \pm 1.48$ |
| Chameleon | $\mathbf{69.04 \pm 1.74}$ | $68.38 \pm 1.36$ | $68.16 \pm 1.79$ | $66.78 \pm 2.79$ |
| Squirrel | $\mathbf{58.02 \pm 1.86}$ | $54.53 \pm 2.09$ | $55.35 \pm 1.72$ | $52.98 \pm 1.66$ |
| Cora | $88.62 \pm 1.22$ | $\mathbf{89.00 \pm 0.72}$ | $88.41 \pm 1.63$ | $88.72 \pm 1.5$ |
| Citeseer | $81.68 \pm 0.97$ | $\mathbf{81.79 \pm 0.95}$ | $81.65 \pm 1.48$ | $81.72 \pm 1.58$ |
| PubMed | $90.66 \pm 0.47$ | $\mathbf{90.74 \pm 0.5}$ | $90.46 \pm 0.69$ | $90.39 \pm 1.33$ |

Table 5: Ablation study of $W_{\text{mix}}$

## A.4 Ablation Study of $W_{\text{mix}}$

From table 5 we can see that ACM(II) with $W_{\text{mix}}$ shows superiority in most datasets, although it is not statistically significant on some of them.

One possible explanation of the function of $W_{\text{mix}}$ is that it could help alleviate the dominance and bias to majority: Suppose in a dataset, most of the nodes need more information from LP channel than HP and identity channels, then $W_L, W_H, W_I$ tend to learn larger $\alpha_L$ than $\alpha_H$ and $\alpha_I$. For the minority nodes that need more information from HP or identity channels, they are hard to get large $\alpha_H$ or $\alpha_I$ values because $W_L, W_H, W_I$ are biased to the majority. And $W_{\text{mix}}$ can help us to learn more diverse alpha values when $W_L, W_H, W_I$ are biased.

Attention with more complicated design can be found for the node-wise adaptive channel mixing mechanism, but we do not explore this direction deeper in this paper because investigating attention function is not the main contribution of our paper.

## A.5 Learn Weights with Raw Features v.s. Combined Features

| Datasets/Models | With Raw Features | | With Combined Features | |
|---|---|---|---|---|
| | ACM | ACMII | ACM | ACMII |
| Cornell | $94.75 \pm 3.8$ | $\mathbf{95.9 \pm 1.83}$ | $95.08 \pm 2.64$ | $93.93 \pm 3.52$ |
| Wisconsin | $95.75 \pm 2.03$ | $\mathbf{96.62 \pm 2.44}$ | $96.12 \pm 1.31$ | $96 \pm 2$ |
| Texas | $94.92 \pm 2.88$ | $\mathbf{95.08 \pm 2.07}$ | $94.92 \pm 2.48$ | $94.59 \pm 2.94$ |
| Film | $41.62 \pm 1.15$ | $\mathbf{41.84 \pm 1.15}$ | $41.62 \pm 1.34$ | $41.44 \pm 1.18$ |
| Chameleon | $\mathbf{69.04 \pm 1.74}$ | $68.38 \pm 1.36$ | $68.82 \pm 2.18$ | $68.53 \pm 3.08$ |
| Squirrel | $\mathbf{58.02 \pm 1.86}$ | $54.53 \pm 2.09$ | $57.48 \pm 1.68$ | $53.28 \pm 1.08$ |
| Cora | $88.62 \pm 1.22$ | $\mathbf{89.00 \pm 0.72}$ | $88.59 \pm 1.04$ | $88.75 \pm 0.83$ |
| Citeseer | $81.68 \pm 0.97$ | $81.79 \pm 0.95$ | $\mathbf{81.9 \pm 1.27}$ | $81.76 \pm 1.05$ |
| PubMed | $90.66 \pm 0.47$ | $90.74 \pm 0.5$ | $\mathbf{90.75 \pm 0.77}$ | $90.58 \pm 0.64$ |

Table 6: Performance comparison between raw features and combined features

Construct the combined feature $H^l_{\text{Comb}} = [H^l_L, H^l_H, H^l_I]$, Replace the first line in Step 2 by the following lines:

$$\tilde{\alpha}^l_L = \sigma\left(H^l_{\text{Comb}}\tilde{W}^l_L\right), \ \tilde{\alpha}^l_H = \sigma\left(H^l_{\text{Comb}}\tilde{W}^l_H\right), \ \tilde{\alpha}^l_I = \sigma\left(H^l_{\text{Comb}}\tilde{W}^l_I\right), \ \tilde{W}^{l-1}_L, \tilde{W}^{l-1}_H, \tilde{W}^{l-1}_I \in \mathbb{R}^{3F_l \times 1}$$

$$[\alpha^l_L, \alpha^l_H, \alpha^l_I] = \text{Softmax}\left(\left(\left[\tilde{\alpha}^l_L, \tilde{\alpha}^l_H, \tilde{\alpha}^l_I\right]/T\right)W^l_{\text{Mix}}\right) \in \mathbb{R}^{N \times 3}, \ T \in \mathbb{R} \text{ temperature}, \ W^l_{\text{Mix}} \in \mathbb{R}^{3 \times 3};$$

The performance comparison can be found in table 6. From the results, we do not find significant difference between the frameworks with combined features and raw features. The reason is that the necessary nonlinear information from each channel is combined in $\left[\tilde{\alpha}^l_L, \tilde{\alpha}^l_H, \tilde{\alpha}^l_I\right]$ and $W^l_{\text{Mix}}$ is enough

to learn to mix the combined weights from different channels. The learning of redundant information in the feature extraction step for each channel will not improve the performance. Meanwhile, A disadvantage of the combined feature is that it increases the computational cost. Thus, we decide to use the raw features.

## A.6 $H_{\text{node}}^{v}$ Distributions of Different Datasets

See Figure 7 for $H_{\text{node}}^{v}$ distributions. We can see that *Wisconsin* and *Texas* have high density in low homophily area, *Cornell, Chameleon, Squirrel* and *Film* have high density in low and middle homophily area, *Cora, CiteSeer* and *PubMed* have high density in high homophily area.

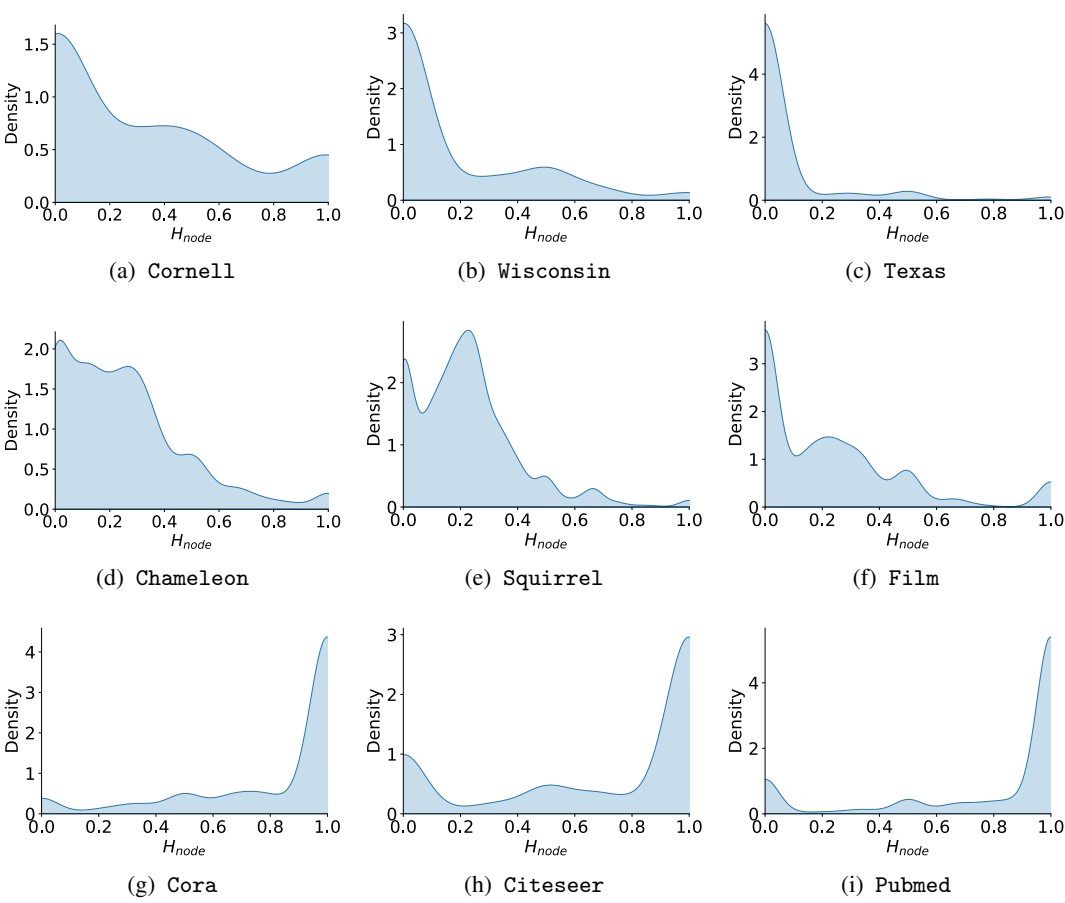

Figure 7: $H_{\text{node}}^{v}$ distributions of different datasets

## A.7 Distributions of Learned $\alpha_L, \alpha_H, \alpha_I$ in the Hidden and Output Layers of ACN-GCN

See Figure 8 for the distributions of weights in hidden layers and Figure 9 for the distributions of weights in output layers.

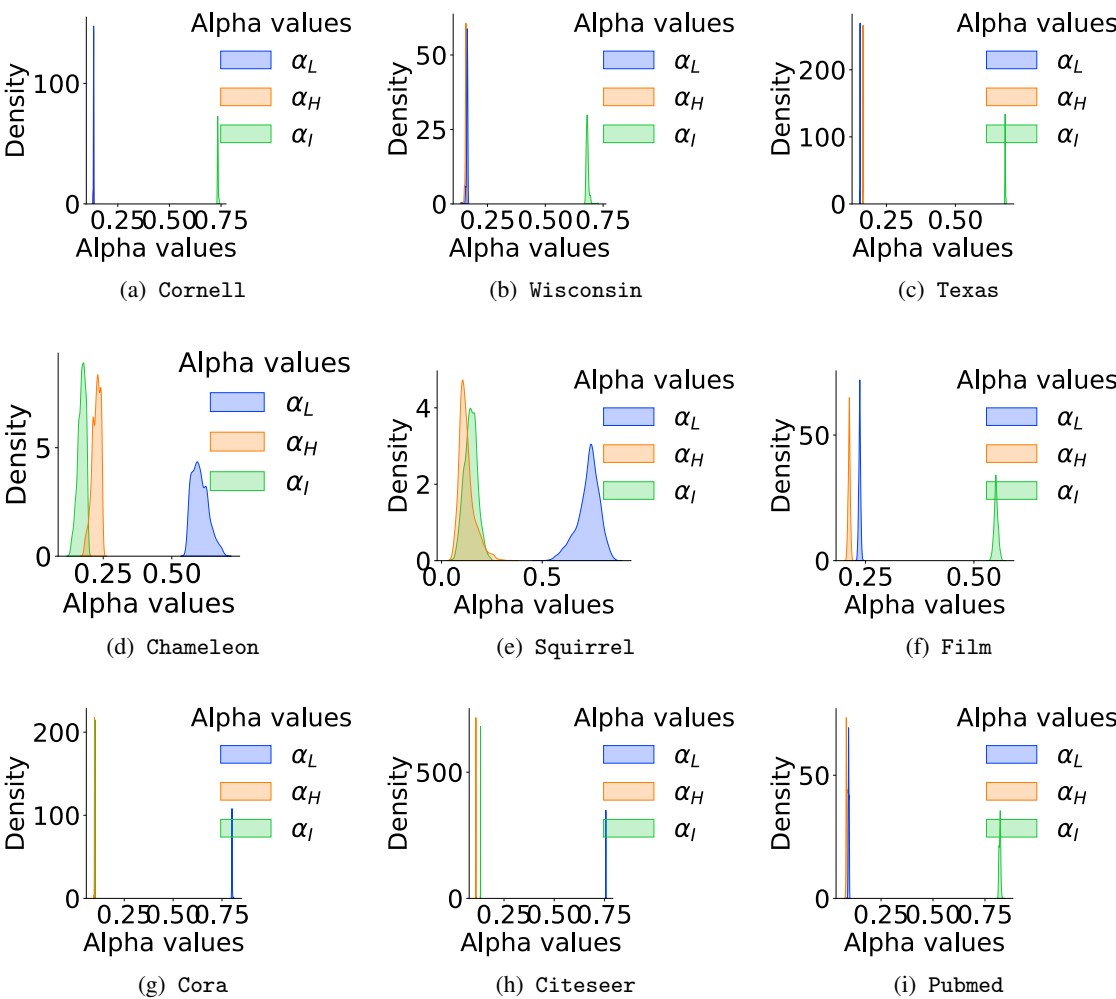

Figure 8: Distributions of the learned $\alpha_L, \alpha_H, \alpha_I$ in the hidden layer of ACM-GCN

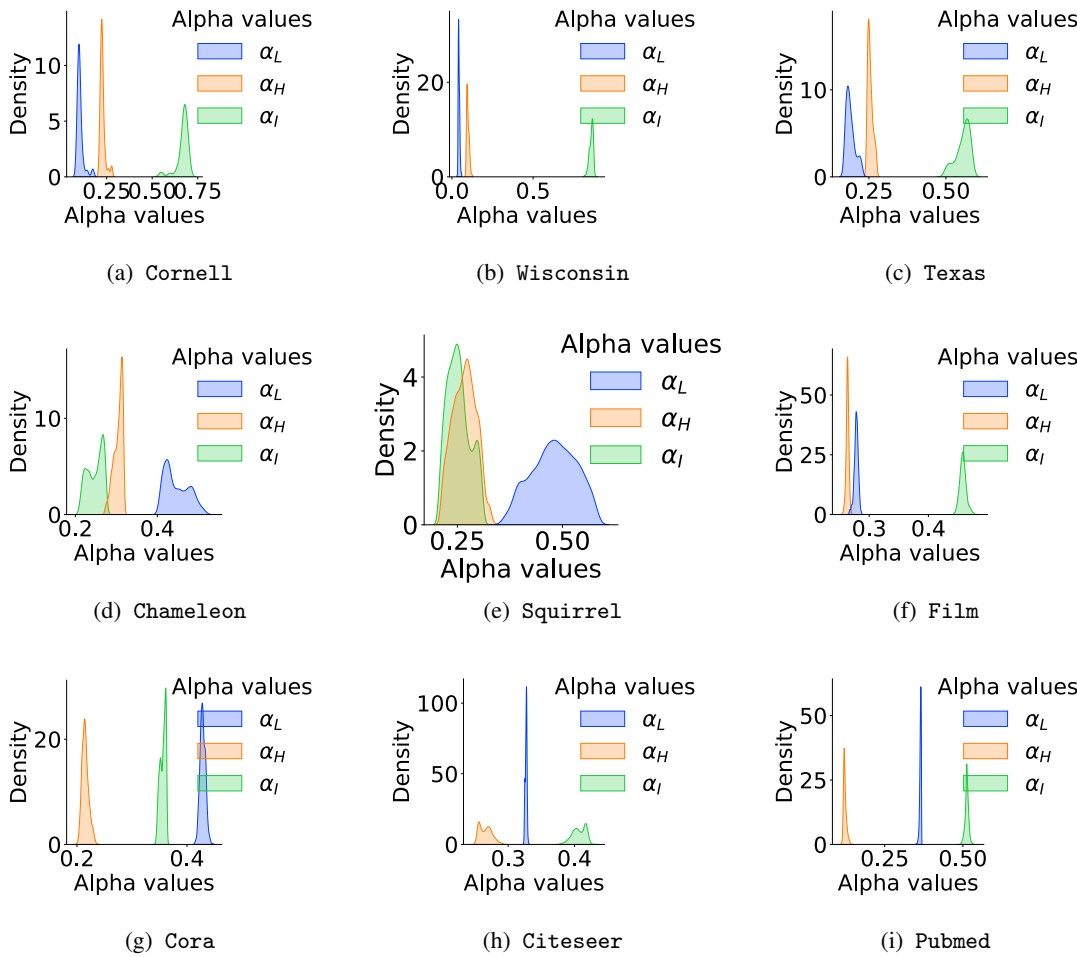

Figure 9: Distributions of the learned $\alpha_L, \alpha_H, \alpha_I$ in the output layer of ACM-GCN

## B    Details of the Implementation

### B.1    Why Do We Need 2 Options for ACM-GNNs

If we consider multiplying $\hat{A}$ as a filtering process, then in $\hat{A}XW$, we actually feed a linear transformed input $XW$ in to $\hat{A}$. In some cases, this linear feature extractor might not be strong enough. Thus, we try to provide an additional option, which uses a non-linear extractor for the filter $\hat{A}$ to process.

### B.2    Implementation of ACM-GCMII

Unlike other baseline GNN models, GCNII and GCNII* are not able to be applied under ACMII framework and we will make an explanation as follows.

GCNII: $\mathbf{H}^{(\ell+1)} = \sigma\left(\left((1-\alpha_\ell)\,\hat{\mathbf{A}}\mathbf{H}^{(\ell)} + \alpha_\ell\mathbf{H}^{(0)}\right)\left((1-\beta_\ell)\,\mathbf{I}_n + \beta_\ell\mathbf{W}^{(\ell)}\right)\right)$

GCNII*: $\mathbf{H}^{(\ell+1)} = \sigma\left((1-\alpha_\ell)\,\hat{\mathbf{A}}\mathbf{H}^{(\ell)}\left((1-\beta_\ell)\,\mathbf{I}_n + \beta_\ell\mathbf{W}_1^{(\ell)}\right) + +\alpha_\ell\mathbf{H}^{(0)}\left((1-\beta_\ell)\,\mathbf{I}_n + \beta_\ell\mathbf{W}_2^{(\ell)}\right)\right)$

From the above formulas of GCNII and GCNII* we cam see that, without major modification, GCNII and GCNII* are hard to be put into ACMII framework. In ACMII framework, before apply $\hat{A}$,

we first implement a nonlinear feature extractor $\sigma(H^\ell \mathbf{W}^{(\ell)})$. But in GCNII and GCNII*, before multiplying $W^\ell$(or $W_1^\ell, W_2^\ell$) to extract features, we need to add another term including $H^{(0)}$, which are not filtered by $\hat{A}$. This makes the order of aggregator $\hat{A}$ and nonlinear extractor unexchangable and thus, incompatible with ACMII framework. So we did not implement GCNII and GCNII* in ACMII framework.

### B.3   Implementation of ACM(II)-GCN+ and ACM(II)-GCN++

Besides the features extracted by different filters, some recent SOTA models use additional graph structure information explicitly, *i.e.,* $\text{MLP}_\theta(A)$ , to address heterophily problem, *e.g.,* LINKX [25] and GloGNN [24] and is found effective on some datasets, *e.g., Chameleon, Squirrel*. The explicit structure information can be directly incorporated into ACM and ACMII framework, and we have ACM(II)-GCN+ and ACM(II)-GCN++ as follows.

- ACM-GCN+ and ACMII-GCN+ have an option to include structure information channel (the 4-th channel) in each layer and their differences from ACM-GCN and ACMII-GCN are highlighted in red) as follows,

**Step 1. Feature Extraction for LP, HP, Identity and Structure Information Channel:**

$H_A^l = \text{ReLU}\left(A W_A^l\right), W_A^l \in \mathbb{R}^{N \times F_l}$, get $H_L^l, H_H^l, H_I^l$ with the same step as ACM-GCN and ACMII-GCN.

**Step 2. Row-wise Feature-based Weight Learning with Layer Normalization (LN)**

$\tilde{H}_L^l = \text{LN}(H_L^l), \ \tilde{H}_H^l = \text{LN}(H_H^l), \ \tilde{H}_I^l = \text{LN}(H_I^l), \ \tilde{H}_A^l = \text{LN}(H_A^l),$

$\tilde{\alpha}_L^l = \text{Sigmoid}\left(\tilde{H}_L^l \tilde{W}_L^l\right), \ \tilde{\alpha}_H^l = \text{Sigmoid}\left(\tilde{H}_H^l \tilde{W}_H^l\right), \tilde{\alpha}_I^l = \text{Sigmoid}\left(\tilde{H}_I^l \tilde{W}_I^l\right), \tilde{\alpha}_A^l = \text{Sigmoid}\left(\tilde{H}_A^l \tilde{W}_A^l\right),$

$\tilde{W}_L^{l-1}, \ \tilde{W}_H^{l-1}, \ \tilde{W}_I^{l-1}, \tilde{W}_A^l \in \mathbb{R}^{F_l \times 1}$

**Step 3. Node-wise Adaptive Channel Mixing:**

Option 1: without structure information

$\left[\alpha_L^l, \alpha_H^l, \alpha_I^l\right] = \text{Softmax}\left(\left(\left[\tilde{\alpha}_L^l, \tilde{\alpha}_H^l, \tilde{\alpha}_I^l\right]/T\right) W_{\text{Mix}}^l\right) \in \mathbb{R}^{N \times 3}, T = 3 \text{ temperature}, \ W_{\text{Mix}}^l \in \mathbb{R}^{3 \times 3};$

$H^l = \text{ReLU}\left(\text{diag}(\alpha_L^l) H_L^l + \text{diag}(\alpha_H^l) H_H^l + \text{diag}(\alpha_I^l) H_I^l\right)$

Option 2: with structure information

$\left[\alpha_L^l, \alpha_H^l, \alpha_I^l, \alpha_A^l\right] = \text{Softmax}\left(\left(\left[\tilde{\alpha}_L^l, \tilde{\alpha}_H^l, \tilde{\alpha}_I^l, \tilde{\alpha}_A^l\right]/T\right) W_{\text{Mix}}^l\right) \in \mathbb{R}^{N \times 4}, T = 4 \text{ temperature}, \ W_{\text{Mix}}^l \in \mathbb{R}^{4 \times 4};$

$H^l = \text{ReLU}\left(\text{diag}(\alpha_L^l) H_L^l + \text{diag}(\alpha_H^l) H_H^l + \text{diag}(\alpha_I^l) H_I^l + \text{diag}(\alpha_A^l) H_A^l\right)$

- ACM-GCN++ and ACMII-GCN++ have an option to include structure information channel (the 4-th channel) in each layer and residual connection and their differences from ACM-GCN+ and ACMII-GCN+ are highlighted in red) as follows,

**Step 1. Feature Extraction for LP, HP, Identity and Structure Information Channel, Get $H_X$:**

$H_X = \text{ReLU}(XW_X) \in \mathbb{R}^{F \times F'}, H_A^l = \text{ReLU}(AW_A^l), W_A^l \in \mathbb{R}^{N \times F'},$

get $H_L^l, H_H^l, H_I^l$ with the same step as ACM-GCN and ACMII-GCN.

**Step 2. Row-wise Feature-based Weight Learning with Layer Normalization (LN)**

$\tilde{H}_L^l = \text{LN}(H_L^l), \ \tilde{H}_H^l = \text{LN}(H_H^l), \ \tilde{H}_I^l = \text{LN}(H_I^l), \ \tilde{H}_A^l = \text{LN}(H_A^l),$

$\tilde{\alpha}_L^l = \text{Sigmoid}\left(\tilde{H}_L^l \tilde{W}_L^l\right), \ \tilde{\alpha}_H^l = \text{Sigmoid}\left(\tilde{H}_H^l \tilde{W}_H^l\right), \tilde{\alpha}_I^l = \text{Sigmoid}\left(\tilde{H}_I^l \tilde{W}_I^l\right), \tilde{\alpha}_A^l = \text{Sigmoid}\left(\tilde{H}_A^l \tilde{W}_A^l\right),$

$\tilde{W}_L^{l-1}, \ \tilde{W}_H^{l-1}, \ \tilde{W}_I^{l-1}, \tilde{W}_A^l \in \mathbb{R}^{F' \times 1}$

**Step 3. Node-wise Adaptive Channel Mixing:**

Option 1: without structure information

$\left[\alpha_L^l, \alpha_H^l, \alpha_I^l\right] = \text{Softmax}\left(\left(\left[\tilde{\alpha}_L^l, \tilde{\alpha}_H^l, \tilde{\alpha}_I^l\right]/T\right)W_{\text{Mix}}^l\right) \in \mathbb{R}^{N \times 3}, T = 3 \text{ temperature}, \ W_{\text{Mix}}^l \in \mathbb{R}^{3 \times 3};$

$H^l = \text{ReLU}\left(\text{diag}(\alpha_L^l)H_L^l + \text{diag}(\alpha_H^l)H_H^l + \text{diag}(\alpha_I^l)H_I^l\right) + H_X$

Option 2: with structure information

$\left[\alpha_L^l, \alpha_H^l, \alpha_I^l, \alpha_A^l\right] = \text{Softmax}\left(\left(\left[\tilde{\alpha}_L^l, \tilde{\alpha}_H^l, \tilde{\alpha}_I^l, \tilde{\alpha}_A^l\right]/T\right)W_{\text{Mix}}^l\right) \in \mathbb{R}^{N \times 4}, T = 4 \text{ temperature}, \ W_{\text{Mix}}^l \in \mathbb{R}^{4 \times 4};$

$H^l = \text{ReLU}\left(\text{diag}(\alpha_L^l)H_L^l + \text{diag}(\alpha_H^l)H_H^l + \text{diag}(\alpha_I^l)H_I^l + \text{diag}(\alpha_A^l)H_A^l\right) + H_X$

The results of ACM-GCN+, ACMII-GCN+, ACM-GCN++ and ACMII-GCN++ trained on random 60%/20%/20% splits are reported in table 2 in Appendix A.1. The results on fixed 48%/32%/20% splits are reported in table 3 in Appendix A.2.

**Practical Issues of ACM(II)-GCN+ and ACM(II)-GCN++**

**Computing Resources** For all experiments on synthetic datasets and real-world datasets, we use NVIDIA V100 GPUs with 16/32GB GPU memory, 8-core CPU, 16G Memory. The software implementation is based on PyTorch and PyTorch Geometric [12].

# C  Hyperparameter Searching Range & Optimal Hyperparameters

## C.1  Hyperparameter Searching Range for Synthetic Experiments

| Hyperparameter Searching Range for Synthetic Experiments | | | | |
|---|---|---|---|---|
| Models\Hyperparameters | lr | weight_decay | dropout | hidden |
| MLP-1 | 0.05 | {5e-5, 1e-4, 5e-4, 1e-3, 5e-3 } | - | - |
| SGC-1 | 0.05 | {5e-5, 1e-4, 5e-4, 1e-3, 5e-3} | - | - |
| ACM-SGC-1 | 0.05 | {5e-5, 1e-4, 5e-4, 1e-3, 5e-3} | { 0.1, 0.3, 0.5, 0.7, 0.9} | - |
| MLP-2 | 0.05 | {5e-5, 1e-4, 5e-4, 1e-3, 5e-3} | { 0.1, 0.3, 0.5, 0.7, 0.9} | 64 |
| GCN | 0.05 | {5e-5, 1e-4, 5e-4, 1e-3, 5e-3} | { 0.1, 0.3, 0.5, 0.7, 0.9} | 64 |
| ACM-GCN | 0.05 | {5e-5, 1e-4, 5e-4, 1e-3, 5e-3} | { 0.1, 0.3, 0.5, 0.7, 0.9} | 64 |

Table 7: Hyperparameter searching range for synthetic experiments

## C.2  Hyperparameter Searching Range for Ablation Study

## C.3  Hyperparameter Searching Range for GNNs on Real-world Datasets

See table 9 for the hyperparameter seaching range of baseline GNNs, ACM-GNNs, ACMII-GNNs and several SOTA models.

| | | | | | | | | | |
|---|---|---|---|---|---|---|---|---|---|
| | | | Hyperparameter Searching Range for Ablation Study | | | | | | |
| Models\Hyperparameters | lr | weight_decay | | | dropout | | | | hidden |
| SGC-LP+HP | {0.01, 0.05, 0.1} | {0, 5e-6, 1e-5, 5e-5, 1e-4, 5e-4, 1e-3, 5e-3, 1e-2} | | | - | | | | - |
| SGC-LP+Identity | {0.01, 0.05, 0.1} | {0, 5e-6, 1e-5, 5e-5, 1e-4, 5e-4, 1e-3, 5e-3, 1e-2} | | | - | | | | - |
| ACM-SGC-no adaptive mixing | {0.01, 0.05, 0.1} | {0, 5e-6, 1e-5, 5e-5, 1e-4, 5e-4, 1e-3, 5e-3, 1e-2} | | | {0, 0.1, 0.2, 0.3, 0.4, 0.5, 0.6, 0.7,0.8,0.9} | | | | - |
| GCN-LP+HP | {0.01, 0.05, 0.1} | {0, 5e-6, 1e-5, 5e-5, 1e-4, 5e-4, 1e-3, 5e-3, 1e-2} | | | {0, 0.1, 0.2, 0.3, 0.4, 0.5, 0.6, 0.7,0.8,0.9} | | | | 64 |
| GCN-LP+Identity | {0.01, 0.05, 0.1} | {0, 5e-6, 1e-5, 5e-5, 1e-4, 5e-4, 1e-3, 5e-3, 1e-2} | | | {0, 0.1, 0.2, 0.3, 0.4, 0.5, 0.6, 0.7,0.8,0.9} | | | | 64 |
| ACM-GCN-no adaptive mixing | {0.01, 0.05, 0.1} | {0, 5e-6, 1e-5, 5e-5, 1e-4, 5e-4, 1e-3, 5e-3, 1e-2} | | | {0, 0.1, 0.2, 0.3, 0.4, 0.5, 0.6, 0.7,0.8,0.9} | | | | 64 |

Table 8: Hyperparameter searching range for ablation study

| Models\Hyperparameters | lr | weight_decay | dropout | hidden | lambda | alpha_l | head | layers | JK type |
|---|---|---|---|---|---|---|---|---|---|
| H2GCN | 0.01 | 0.001 | {0, 0.5} | {8, 16, 32, 64} | - | - | - | {1, 2} | - |
| MixHop | 0.01 | 0.001 | 0.5 | {8, 16, 32} | - | - | - | {2, 3} | - |
| GCN+JK | {0.1, 0.01, 0.001} | 0.001 | 0.5 | {4, 8, 16, 32, 64} | - | - | - | 2 | {max, cat} |
| GAT+JK | {0.1, 0.01, 0.001} | 0.001 | 0.5 | {4, 8, 12, 32} | - | - | {2,4,8} | 2 | {max, cat} |
| GCNII, GCNII* | 0.01 | {0, 5e-6, 1e-5, 5e-5, 1e-4, 5e-4, 1e-3} for Deezer-Europe and {0, 5e-6, 1e-5, 5e-5, 1e-4, 5e-4, 1e-3, 5e-3, 1e-2} for others | 0.5 | 64 | {0.5, 1, 1.5} | {0.1,0.2,0.3,0,4,0.5} | | {4, 8, 16, 32} for Deezer-Europe and {4, 8, 16, 32, 64} for others | - |
| Baselines: {SGC-1, SGC-2, GCN, Snowball-2, Snowball-3, FAGCN}; ACM-{SGC-1, SGC-2, GCN, GCN+, GCN++, Snowball-2, Snowball-3}; ACMII-{SGC-1, SGC-2, GCN, GCN+, GCN++, Snowball-2, Snowball-3} | {0.002, 0.01, 0.05} for Deezer-Europe and {0.01, 0.05, 0.1} for others | {0, 5e-6, 1e-5, 5e-5, 1e-4, 5e-4, 1e-3} for Deezer-Europe and {0, 5e-6, 1e-5, 5e-5, 1e-4, 5e-4, 1e-3, 5e-3, 1e-2} for others | {0, 0.1, 0.2, 0.3, 0.4, 0.5, 0.6, 0.7, 0.8, 0.9} | 64 | - | - | - | - | - |
| GraphSAGE | {0.01,0.05, 0.1} | {0, 5e-6, 1e-5, 5e-5, 1e-4, 5e-4, 1e-3} for Deezer-Europe and {0, 5e-6, 1e-5, 5e-5, 1e-4, 5e-4, 1e-3, 5e-3, 1e-2} for others | { 0, 0.1, 0.2, 0.3, 0.4, 0.5, 0.6, 0.7, 0.8, 0.9} | 8 for Deezer-Europe and 64 for others | - | - | - | - | - |
| ACM-{GCNII, GCNII*} | 0.01 | {0, 5e-6, 1e-5, 5e-5, 1e-4, 5e-4, 1e-3} for Deezer-Europe and {0, 5e-6, 1e-5, 5e-5, 1e-4, 5e-4, 1e-3, 5e-3, 1e-2} for others | { 0, 0.1, 0.2, 0.3, 0.4, 0.5, 0.6, 0.7, 0.8, 0.9} | 64 | - | - | - | {1,2,3,4} | - |

Table 9: Hyperparameter searching range for training on real-world datasets

## C.4 Searched Optimal Hyperparameters for Baselines and ACM(II)-GNNs on Real-world Tasks

See the reported optimal hyperparameters on random 60%/20%/20% splits for baseline GNNs in table 10, for ACM-GNNs and ACMII-GNNs in table 11 and for ACM(II)-GCN+ and ACM(II)-GCN++ in table 12.

See the reported optimal hyperparameters on fixed 48%/32%/20% splits for ACM(II)-GNNs and FAGCN in table 13 and for ACM(II)-GCN+ and ACM(II)-GCN++ in table 14.

**Hyperparameters for Baseline GNNs**

| Datasets | Models\Hyperparameters | lr | weight_decay | dropout | hidden | # layers | Gat heads | JK Type | lambda | alpha_l | results | std | average epoch time/average total time |
|---|---|---|---|---|---|---|---|---|---|---|---|---|---|
| **Cornell** | SGC-1 | 0.05 | 1.00E-02 | 0 | 64 | - | - | - | - | - | 70.98 | 8.39 | 2.53ms/0.51s |
| | SGC-2 | 0.05 | 1.00E-03 | 0 | 64 | - | - | - | - | - | 72.62 | 9.92 | 2.46ms/0.53s |
| | GCN | 0.1 | 5.00E-03 | 0.5 | 64 | 2 | - | - | - | - | 82.46 | 3.11 | 3.67ms/0.74s |
| | Snowball-2 | 0.01 | 5.00E-03 | 0.4 | 64 | 2 | - | - | - | - | 82.62 | 2.34 | 4.24ms/0.87s |
| | Snowball-3 | 0.01 | 5.00E-03 | 0.4 | 64 | 3 | - | - | - | - | 82.95 | 2.1 | 6.66ms/1.36s |
| | GCNII | 0.01 | 1.00E-03 | 0.5 | 64 | 16 | - | - | 0.5 | 0.5 | 89.18 | 3.96 | 25.41ms/8.11s |
| | GCNII* | 0.01 | 1.00E-03 | 0.5 | 64 | 8 | - | - | 0.5 | 0.5 | 90.49 | 4.45 | 15.35ms/4.05s |
| | FAGCN | 0.01 | 1.00E-04 | 0.7 | 32 | 2 | - | - | - | - | 88.03 | 5.6 | 8.1ms/3.8858s |
| | Mixhop | 0.01 | 0.001 | 0.5 | 16 | 2 | - | - | - | - | 60.33 | 28.53 | 10.379ms/2.105s |
| | H2GCN | 0.01 | 0.001 | 0.5 | 64 | 1 | - | - | - | - | 86.23 | 4.71 | 4.381ms/1.123s |
| | GCN+JK | 0.1 | 0.001 | 0.5 | 64 | 2 | - | cat | - | - | 66.56 | 13.82 | 5.589ms/1.227s |
| | GAT+JK | 0.1 | 0.001 | 0.5 | 32 | 2 | 8 | max | - | - | 74.43 | 10.24 | 10.725ms/2.478s |
| **Wisconsin** | SGC-1 | 0.05 | 5.00E-03 | 0 | 64 | - | - | - | - | - | 70.38 | 2.85 | 2.83ms/0.57s |
| | SGC-2 | 0.1 | 1.00E-03 | 0 | 64 | - | - | - | - | - | 74.75 | 2.89 | 2.14ms/0.43s |
| | GCN | 0.1 | 1.00E-03 | 0.7 | 64 | 2 | - | - | - | - | 75.5 | 2.92 | 3.74ms/0.76s |
| | Snowball-2 | 0.1 | 1.00E-03 | 0.5 | 64 | 2 | - | - | - | - | 74.88 | 3.42 | 3.73ms/0.76s |
| | Snowball-3 | 0.05 | 5.00E-04 | 0.8 | 64 | 3 | - | - | - | - | 69.5 | 5.01 | 5.46ms/1.12s |
| | GCNII | 0.01 | 1.00E-03 | 0.5 | 64 | 8 | - | - | 0.5 | 0.5 | 83.25 | 2.69 | |
| | GCNII* | 0.01 | 1.00E-03 | 0.5 | 64 | 4 | - | - | 1.5 | 0.3 | 89.12 | 3.06 | 9.26ms/1.96s |
| | FAGCN | 0.05 | 1.00E-04 | 0 | 32 | 2 | - | - | - | - | 89.75 | 6.37 | 12.9ms/4.6359s |
| | Mixhop | 0.01 | 0.001 | 0.5 | 16 | 2 | - | - | - | - | 77.25 | 7.80 | 10.281ms/2.095s |
| | H2GCN | 0.01 | 0.001 | 0.5 | 32 | 1 | - | - | - | - | 87.5 | 1.77 | 4.324ms/1.134s |
| | GCN+JK | 0.1 | 0.001 | 0.5 | 32 | 2 | - | cat | - | - | 62.5 | 15.75 | 5.117ms/1.049s |
| | GAT+JK | 0.1 | 0.001 | 0.5 | 4 | 2 | 8 | max | - | - | 69.5 | 3.12 | 10.762ms/2.25s |
| | APPNP | 0.05 | 0.001 | 0.5 | 64 | 2 | - | - | - | - | 92 | 3.59 | 10.303ms/2.104s |
| | GPRGNN | 0.05 | 0.001 | 0.5 | 256 | 2 | - | - | - | - | 93.75 | 2.37 | 11.856ms/2.415s |
| **Texas** | SGC-1 | 0.05 | 1.00E-03 | 0 | 64 | - | - | - | - | - | 83.28 | 5.43 | 2.55ms/0.54s |
| | SGC-2 | 0.01 | 1.00E-03 | 0 | 64 | - | - | - | - | - | 81.31 | 3.3 | 2.61ms/2.53s |
| | GCN | 0.05 | 1.00E-02 | 0.9 | 64 | 2 | - | - | - | - | 83.11 | 3.2 | 3.59ms/0.73s |
| | Snowball-2 | 0.05 | 1.00E-02 | 0.9 | 64 | 2 | - | - | - | - | 83.11 | 3.2 | 3.98ms/0.82s |
| | Snowball-3 | 0.05 | 1.00E-02 | 0.9 | 64 | 3 | - | - | - | - | 83.11 | 3.2 | 5.56ms/1.12s |
| | GCNII | 0.01 | 1.00E-04 | 0.5 | 64 | 4 | - | - | 1.5 | 0.5 | 82.46 | 4.58 | |
| | GCNII* | 0.01 | 1.00E-04 | 0.5 | 64 | 8 | - | - | 0.5 | 0.5 | 88.52 | 3.02 | 15.64ms/3.47s |
| | FAGCN | 0.01 | 5.00E-04 | 0 | 32 | 2 | - | - | - | - | 88.85 | 4.39 | 8.8ms/6.5252s |
| | Mixhop | 0.01 | 0.001 | 0.5 | 32 | 2 | - | - | - | - | 76.39 | 7.66 | 11.099ms/2.329s |
| | H2GCN | 0.01 | 0.001 | 0.5 | 64 | 1 | - | - | - | - | 85.90 | 3.53 | 4.197ms/0.95s |
| | GCN+JK | 0.1 | 0.001 | 0.5 | 32 | 2 | - | cat | - | - | 80.66 | 1.91 | 5.28ms/1.085s |
| | GAT+JK | 0.1 | 0.001 | 0.5 | 8 | 2 | 2 | cat | - | - | 75.41 | 7.18 | 10.937ms/2.402s |
| **Film** | SGC-1 | 0.01 | 5.00E-06 | 0 | 64 | - | - | - | - | - | 25.26 | 1.18 | 3.18ms/0.70s |
| | SGC-2 | 0.01 | 5.00E-06 | 0 | 64 | - | - | - | - | - | 28.81 | 1.11 | 2.13ms/0.43s |
| | GCN | 0.1 | 5.00E-04 | 0 | 64 | 2 | - | - | - | - | 35.51 | 0.99 | 4.86ms/0.99s |
| | Snowball-2 | 0.1 | 5.00E-04 | 0 | 64 | 2 | - | - | - | - | 35.97 | 0.66 | 5.59ms/1.14s |
| | Snowball-3 | 0.1 | 5.00E-04 | 0.2 | 64 | 3 | - | - | - | - | 36 | 1.36 | 7.89ms/1.60s |
| | GCNII | 0.01 | 1.00E-04 | 0.5 | 64 | 8 | - | - | 1.5 | 0.3 | 40.82 | 1.79 | 15.85ms/3.22s |
| | GCNII* | 0.01 | 1.00E-06 | 0.5 | 64 | 4 | - | - | 1 | 0.1 | 41.54 | 0.99 | |
| | FAGCN | 0.01 | 5.00E-05 | 0.6 | 32 | 2 | - | - | - | - | 31.59 | 1.37 | 45.4ms/11.107s |
| | Mixhop | 0.01 | 0.001 | 0.5 | 8 | 3 | 8 | max | - | - | 33.13 | 2.40 | 17.651ms/3.566s |
| | H2GCN | 0.01 | 0.001 | 0 | 64 | 1 | 8 | max | - | - | 38.85 | 1.17 | 8.101ms/1.695s |
| | GCN+JK | 0.1 | 0.001 | 0.5 | 64 | 2 | 8 | cat | - | - | 32.72 | 2.62 | 8.946ms/1.807s |
| | GAT+JK | 0.001 | 0.001 | 0.5 | 32 | 2 | 4 | cat | - | - | 35.41 | 0.97 | 20.726ms/4.187s |
| **Chameleon** | SGC-1 | 0.1 | 5.00E-06 | 0 | 64 | - | - | - | - | - | 64.86 | 1.81 | 3.48ms/2.96s |
| | SGC-2 | 0.1 | 0.00E+00 | 0 | 64 | - | - | - | - | - | 62.67 | 2.41 | 4.43ms/1.12s |
| | GCN | 0.01 | 1.00E-05 | 0.9 | 64 | 2 | - | - | - | - | 64.18 | 2.62 | 4.96ms/1.18s |
| | Snowball-2 | 1.00E-01 | 1.00E-05 | 0.9 | 64 | 2 | - | - | - | - | 64.99 | 2.39 | 4.96ms/1.00s |
| | Snowball-3 | 0.1 | 5.00E-06 | 0.9 | 64 | 3 | - | - | - | - | 65.49 | 1.64 | 7.44ms/1.50s |
| | GCNII | 0.01 | 5.00E-06 | 0.5 | 64 | 4 | - | - | 0.5 | 0.1 | 60.35 | 2.7 | 9.76ms/2.26s |
| | GCNII* | 0.01 | 5.00E-04 | 0.5 | 64 | 4 | - | - | 1.5 | 0.5 | 62.8 | 2.87 | 10.40ms/2.17s |
| | FAGCN | 0.002 | 1.00E-04 | 0 | 32 | 2 | - | - | - | - | 49.47 | 2.84 | 8.4ms/13.8696s |
| | Mixhop | 0.01 | 0.001 | 0.5 | 16 | 2 | 8 | max | - | - | 36.28 | 10.2 | 11.372ms/2.297s |
| | H2GCN | 0.01 | 0.001 | 0 | 32 | 1 | 8 | max | - | - | 52.3 | 0.48 | 4.059ms/0.82s |
| | GCN+JK | 0.001 | 0.001 | 0.5 | 32 | 2 | 8 | cat | - | - | 64.68 | 2.85 | 5.211ms/1.053s |
| | GAT+JK | 0.001 | 0.001 | 0.5 | 4 | 2 | 8 | max | - | - | 68.14 | 1.18 | 13.772ms/2.788s |
| **Squirrel** | SGC-1 | 0.05 | 0.00E+00 | 0 | 64 | - | - | - | - | - | 47.62 | 1.27 | 4.65ms/1.44s |
| | SGC-2 | 0.1 | 0.00E+00 | 0.9 | 64 | - | - | - | - | - | 41.25 | 1.4 | 35.06ms/7.81s |
| | GCN | 0.01 | 5.00E-05 | 0.7 | 64 | 2 | - | - | - | - | 44.76 | 1.39 | 8.41ms/2.50s |
| | Snowball-2 | 0.1 | 0.00E+00 | 0.9 | 64 | 2 | - | - | - | - | 47.88 | 1.23 | 8.96ms/1.92s |
| | Snowball-3 | 0.1 | 0.00E+00 | 0.8 | 64 | 3 | - | - | - | - | 48.25 | 0.94 | 14.00ms/2.90s |
| | GCNII | 0.01 | 1.00E-04 | 0.5 | 64 | 4 | - | - | 1.5 | 0.2 | 38.81 | 1.97 | 13.35ms/2.70s |
| | GCNII* | 0.01 | 5.00E-04 | 0.5 | 64 | 4 | - | - | 1.5 | 0.3 | 38.31 | 1.3 | 13.81ms/2.78s |
| | FAGCN | 0.05 | 1.00E-04 | 0 | 32 | 2 | - | - | - | - | 42.24 | 1.2 | 16ms/6.7961s |
| | Mixhop | 0.01 | 0.001 | 0.5 | 32 | 2 | - | - | - | - | 24.55 | 2.6 | 17.634ms/3.562s |
| | H2GCN | 0.01 | 0.001 | 0 | 16 | 1 | - | - | - | - | 30.39 | 1.22 | 9.315ms/1.882s |
| | GCN+JK | 0.001 | 0.001 | 0.5 | 32 | 2 | - | max | - | - | 53.4 | 1.9 | 14.321ms/2.905s |
| | GAT+JK | 0.001 | 0.001 | 0.5 | 8 | 2 | 4 | max | - | - | 52.28 | 3.61 | 29.097ms/5.878s |
| **Cora** | SGC-1 | 0.1 | 5.00E-06 | 0 | 64 | - | - | - | - | - | 85.12 | 1.64 | 3.47ms/11.55s |
| | SGC-2 | 0.1 | 1.00E-05 | 0 | 64 | - | - | - | - | - | 85.48 | 1.48 | 2.91ms/6.85s |
| | GCN | 0.1 | 5.00E-04 | 0.2 | 64 | 2 | - | - | - | - | 87.78 | 0.96 | 4.24ms/0.86s |
| | Snowball-2 | 0.1 | 5.00E-04 | 0.1 | 64 | 2 | - | - | - | - | 88.64 | 1.15 | 4.65ms/0.94s |
| | Snowball-3 | 0.05 | 1.00E-03 | 0.6 | 64 | 3 | - | - | - | - | 89.33 | 1.3 | 6.41ms/1.32s |
| | GCNII | 0.01 | 1.00E-04 | 0.5 | 64 | 16 | - | - | 0.5 | 0.2 | 88.98 | 1.33 | |
| | GCNII* | 0.01 | 5.00E-04 | 0.5 | 64 | 4 | - | - | 0.5 | 0.5 | 88.93 | 1.37 | 10.16ms/2.24s |
| | FAGCN | 0.05 | 5.00E-04 | 0 | 32 | 2 | - | - | - | - | 88.85 | 1.36 | 8.4ms/3.3183s |
| | Mixhop | 0.01 | 0.001 | 0.5 | 16 | 2 | - | - | - | - | 65.65 | 11.31 | 11.177ms/2.278s |
| | H2GCN | 0.01 | 0.001 | 0 | 32 | 1 | - | - | - | - | 87.52 | 0.61 | 4.335ms/1.209s |
| | GCN+JK | 0.001 | 0.001 | 0.5 | 64 | 2 | - | cat | - | - | 86.90 | 1.51 | 6.656ms/1.346s |
| | GAT+JK | 0.001 | 0.001 | 0.5 | 32 | 2 | 2 | cat | - | - | 89.52 | 0.43 | 12.91ms/2.608s |
| **CiteSeer** | SGC-1 | 0.1 | 5.00E-04 | 0 | 64 | - | - | - | - | - | 79.66 | 0.75 | 3.43ms/7.30s |
| | SGC-2 | 0.01 | 5.00E-04 | 0.9 | 64 | - | - | - | - | - | 80.75 | 1.15 | 5.33ms/4.40s |
| | GCN | 0.1 | 1.00E-03 | 0.9 | 64 | 2 | - | - | - | - | 81.39 | 1.23 | 4.18ms/0.86s |
| | Snowball-2 | 0.1 | 1.00E-03 | 0.8 | 64 | 2 | - | - | - | - | 81.53 | 1.71 | 5.19ms/1.11s |
| | Snowball-3 | 0.1 | 1.00E-03 | 0.9 | 64 | 3 | - | - | - | - | 80.93 | 1.32 | 7.64ms/1.69s |
| | GCNII | 0.01 | 1.00E-03 | 0.5 | 64 | 16 | - | - | 0.5 | 0.2 | 81.58 | 1.3 | |
| | GCNII* | 0.01 | 1.00E-03 | 0.5 | 64 | 16 | - | - | 0.5 | 0.2 | 81.83 | 1.78 | 32.50ms/10.29s |
| | FAGCN | 0.05 | 5.00E-04 | 0 | 32 | 2 | - | - | - | - | 82.37 | 1.46 | 9.4ms/4.7648s |
| | Mixhop | 0.01 | 0.001 | 0.5 | 16 | 2 | - | - | - | - | 49.52 | 13.35 | 13.793ms/2.786s |
| | H2GCN | 0.01 | 0.001 | 0 | 8 | 1 | - | - | - | - | 79.97 | 0.69 | 5.794ms/3.049s |
| | GCN+JK | 0.001 | 0.001 | 0.5 | 32 | 2 | - | max | - | - | 73.77 | 1.85 | 5.264ms/1.063s |
| | GAT+JK | 0.001 | 0.001 | 0.5 | 8 | 2 | 4 | max | - | - | 74.49 | 2.76 | 12.326ms/2.49s |
| **PubMed** | SGC-1 | 0.05 | 5.00E-06 | 0.3 | 64 | - | - | - | - | - | 87.75 | 0.88 | 6.04ms/2.61s |
| | SGC-2 | 0.05 | 5.00E-05 | 0.1 | 64 | - | - | - | - | - | 88.79 | 0.5 | 8.62ms/3.18s |
| | GCN | 0.1 | 5.00E-05 | 0.6 | 64 | 2 | - | - | - | - | 88.9 | 0.32 | 5.08ms/1.03s |
| | Snowball-2 | 0.1 | 5.00E-04 | 0 | 64 | 2 | - | - | - | - | 89.04 | 0.49 | 5.68ms/1.19s |
| | Snowball-3 | 0.1 | 5.00E-06 | 0 | 64 | 3 | - | - | - | - | 88.8 | 0.82 | 8.54ms/1.75s |
| | GCNII | 0.01 | 1.00E-06 | 0.5 | 64 | 4 | - | - | 0.5 | 0.5 | 89.8 | 0.3 | 10.98ms/3.21s |
| | GCNII* | 0.01 | 1.00E-06 | 0.5 | 64 | 4 | - | - | 0.5 | 0.1 | 89.98 | 0.52 | 11.47ms/3.24s |
| | FAGCN | 0.05 | 5.00E-04 | 0 | 32 | 2 | - | - | - | - | 89.98 | 0.54 | 14.5ms/6.411s |
| | Mixhop | 0.01 | 0.001 | 0.5 | 16 | 2 | - | - | - | - | 87.04 | 4.10 | 17.459ms/3.527s |
| | H2GCN | 0.01 | 0.001 | 0 | 64 | 1 | - | - | - | - | 87.78 | 0.28 | 8.039ms/2.28s |
| | GCN+JK | 0.01 | 0.001 | 0.5 | 32 | 2 | - | cat | - | - | 90.09 | 0.68 | 12.001ms/2.424s |
| | GAT+JK | 0.1 | 0.001 | 0.5 | 8 | 2 | 4 | max | - | - | 89.15 | 0.87 | 20.403ms/4.125s |
| **Deezer-Europe** | FAGCN | 0.01 | 0.0001 | 0 | 32 | 2 | - | - | - | - | 66.86 | 0.53 | 41.7ms/20.8362s |
| | GCNII | 0.01 | 5e-6,1e-5 | 0.5 | 64 | 32 | - | - | 0.5 | 0.5 | 66.38 | 0.45 | 126.58ms/63.16s |
| | GCNII* | 0.01 | 1e-4,1e-3 | 0.5 | 64 | 32 | - | - | 0.5 | 0.5 | 66.42 | 0.56 | 134.05ms/66.89s |

Table 10: Optimal hyperparameters for baseline models on random 60%/20%/20% splits

| Datasets | Models\Hyperparameters | lr | weight_decay | dropout | hidden | # layers | Gat heads | JK Type | lambda | alpha_l | results | std | average epoch time/average total time |
|---|---|---|---|---|---|---|---|---|---|---|---|---|---|
| | | | | | | | | | | | **Hyperparameters for ACM-GNNs and ACMII-GNNs** | | |
| Cornell | ACM-SGC-1 | 0.01 | 5.00E-03 | 0.6 | 64 | - | - | - | - | - | 93.77 | 1.91 | 5.53ms/2.31s |
| | ACM-SGC-2 | 0.01 | 5.00E-03 | 0.6 | 64 | - | - | - | - | - | 93.77 | 2.17 | 4.73ms/1.87s |
| | ACM-GCN | 0.05 | 1.00E-02 | 0.2 | 64 | 2 | - | - | - | - | 94.75 | 3.8 | 8.25ms/1.69s |
| | ACMII-GCN | 0.1 | 1.00E-02 | 0.5 | 64 | 2 | - | - | - | - | 95.25 | 2.79 | 8.43ms/1.71s |
| | ACM-GCNII | 0.01 | 1.00E-03 | 0.5 | 64 | 1 | - | - | 0.5 | 0.4 | 92.62 | 3.13 | 6.81ms/1.43s |
| | ACM-GCNII* | 0.01 | 5.00E-04 | 0.5 | 64 | 1 | - | - | 0.5 | 0.1 | 93.44 | 2.74 | 6.76ms/1.39s |
| | ACM-Snowball-2 | 0.05 | 1.00E-02 | 0.2 | 64 | 2 | - | - | - | - | 95.08 | 3.11 | 9.15ms/1.86s |
| | ACM-Snowball-3 | 0.1 | 1.00E-02 | 0.4 | 64 | 3 | - | - | - | - | 94.26 | 2.57 | 13.20ms/2.68s |
| | ACMII-Snowball-2 | 0.05 | 1.00E-02 | 0.6 | 64 | 2 | - | - | - | - | 95.25 | 1.55 | 8.23ms/1.72s |
| | ACMII-Snowball-3 | 0.05 | 1.00E-02 | 0.7 | 64 | 3 | - | - | - | - | 93.61 | 2.79 | 11.70ms/2.37s |
| Wisconsin | ACM-SGC-1 | 0.05 | 5.00E-03 | 0.7 | 64 | - | - | - | - | - | 93.25 | 2.92 | 5.96ms/1.34s |
| | ACM-SGC-2 | 0.1 | 5.00E-03 | 0.2 | 64 | - | - | - | - | - | 94 | 2.61 | 4.60ms/0.95s |
| | ACM-GCN | 0.1 | 5.00E-03 | 0 | 64 | 2 | - | - | - | - | 95.75 | 2.03 | 8.11ms/1.64s |
| | ACMII-GCN | 0.1 | 1.00E-02 | 0.2 | 64 | 2 | - | - | - | - | 96.62 | 2.44 | 8.28ms/1.68s |
| | ACM-GCNII | 0.01 | 5.00E-03 | 0.5 | 64 | 1 | - | - | 1 | 0.1 | 94.63 | 2.96 | 9.31ms/2.19s |
| | ACM-GCNII* | 0.01 | 1.00E-03 | 0.5 | 64 | 1 | - | - | 1.5 | 0.4 | 94.37 | 2.81 | 7.11ms/1.45s |
| | ACM-Snowball-2 | 0.1 | 5.00E-03 | 0.1 | 64 | 2 | - | - | - | - | 96.38 | 2.59 | 8.63ms/1.74s |
| | ACM-Snowball-3 | 0.05 | 1.00E-02 | 0.3 | 64 | 3 | - | - | - | - | 96.62 | 1.86 | 12.79ms/2.58s |
| | ACMII-Snowball-2 | 0.1 | 5.00E-03 | 0.1 | 64 | 2 | - | - | - | - | 96.63 | 2.24 | 8.11ms/1.65s |
| | ACMII-Snowball-3 | 0.1 | 5.00E-03 | 0.1 | 64 | 3 | - | - | - | - | 97 | 2.63 | 12.38ms/2.51s |
| Texas | ACM-SGC-1 | 0.01 | 5.00E-03 | 0.6 | 64 | - | - | - | - | - | 93.61 | 1.55 | 5.43ms/2.18s |
| | ACM-SGC-2 | 0.05 | 5.00E-03 | 0.4 | 64 | - | - | - | - | - | 93.44 | 2.54 | 4.59ms/1.01s |
| | ACM-GCN | 0.05 | 1.00E-02 | 0.6 | 64 | 2 | - | - | - | - | 94.92 | 2.88 | 8.33ms/1.70s |
| | ACMII-GCN | 0.1 | 5.00E-03 | 0.4 | 64 | 2 | - | - | - | - | 95.08 | 2.54 | 8.49ms/1.72s |
| | ACM-GCNII | 0.01 | 1.00E-03 | 0.5 | 64 | 1 | - | - | 0.5 | 0.4 | 92.46 | 1.97 | 6.47ms/1.36s |
| | ACM-GCNII* | 0.01 | 1.00E-03 | 0.5 | 64 | 1 | - | - | 0.5 | 0.4 | 93.28 | 2.79 | 7.03ms/1.45s |
| | ACM-Snowball-2 | 0.05 | 1.00E-02 | 0.1 | 64 | 2 | - | - | - | - | 95.74 | 2.22 | 8.35ms/1.71s |
| | ACM-Snowball-3 | 0.01 | 5.00E-03 | 0.6 | 64 | 3 | - | - | - | - | 94.75 | 2.41 | 12.56ms/2.63s |
| | ACMII-Snowball-2 | 0.1 | 1.00E-02 | 0.4 | 64 | 2 | - | - | - | - | 95.25 | 1.55 | 9.74ms/1.97s |
| | ACMII-Snowball-3 | 0.05 | 1.00E-02 | 0.6 | 64 | 3 | - | - | - | - | 94.75 | 3.09 | 11.91ms/2.42s |
| Film | ACM-SGC-1 | 0.05 | 5.00E-05 | 0.7 | 64 | - | - | - | - | - | 39.33 | 1.25 | 5.21ms/2.33s |
| | ACM-SGC-2 | 0.1 | 5.00E-05 | 0.7 | 64 | - | - | - | - | - | 40.13 | 1.21 | 12.41ms/4.87s |
| | ACM-GCN | 0.1 | 5.00E-04 | 0.5 | 64 | 2 | - | - | - | - | 41.62 | 1.15 | 10.72ms/2.66s |
| | ACMII-GCN | 0.1 | 5.00E-04 | 0.5 | 64 | 2 | - | - | - | - | 41.24 | 1.16 | 10.51ms/2.44s |
| | ACM-GCNII | 0.01 | 0.00E+00 | 0.5 | 64 | 3 | - | - | 1.5 | 0.2 | 41.37 | 1.37 | 13.65ms/2.74s |
| | ACM-GCNII* | 0.01 | 1.00E-05 | 0.5 | 64 | 3 | - | - | 1.5 | 0.1 | 41.27 | 1.24 | 14.98ms/3.01s |
| | ACM-Snowball-2 | 0.1 | 5.00E-03 | 0 | 64 | 2 | - | - | - | - | 41.4 | 1.23 | 10.30ms/2.08s |
| | ACM-Snowball-3 | 0.05 | 1.00E-02 | 0 | 64 | 3 | - | - | - | - | 41.27 | 0.8 | 16.43ms/3.52s |
| | ACMII-Snowball-2 | 0.1 | 5.00E-03 | 0 | 64 | 2 | - | - | - | - | 41.1 | 0.75 | 10.74ms/2.19s |
| | ACMII-Snowball-3 | 0.05 | 1.00E-04 | 0.2 | 64 | 3 | - | - | - | - | 40.31 | 1.6 | 16.31ms/3.29s |
| Chameleon | ACM-SGC-1 | 0.1 | 5.00E-06 | 0.9 | 64 | - | - | - | - | - | 63.68 | 1.62 | 5.41ms/1.21s |
| | ACM-SGC-2 | 0.1 | 5.00E-06 | 0.9 | 64 | - | - | - | - | - | 60.48 | 1.55 | 7.86ms/1.81s |
| | ACM-GCN | 0.01 | 5.00E-05 | 0.8 | 64 | 2 | - | - | - | - | 68.18 | 1.67 | 10.55ms/3.12s |
| | ACMII-GCN | 0.05 | 5.00E-05 | 0.7 | 64 | 2 | - | - | - | - | 68.38 | 1.36 | 10.90ms/2.39s |
| | ACM-GCNII | 0.01 | 5.00E-06 | 0.5 | 64 | 4 | - | - | 0.5 | 0.1 | 58.73 | 2.52 | 18.31ms/3.68s |
| | ACM-GCNII* | 0.01 | 1.00E-03 | 0.5 | 64 | 1 | - | - | 1 | 0.1 | 61.66 | 2.29 | 6.68ms/1.40s |
| | ACM-Snowball-2 | 0.05 | 5.00E-05 | 0.7 | 64 | 2 | - | - | - | - | 68.51 | 1.7 | 9.92ms/2.06s |
| | ACM-Snowball-3 | 0.01 | 1.00E-04 | 0.7 | 64 | 3 | - | - | - | - | 68.4 | 2.05 | 14.49ms/3.15s |
| | ACMII-Snowball-2 | 0.1 | 5.00E-05 | 0.6 | 64 | 2 | - | - | - | - | 67.83 | 2.63 | 9.99ms/2.10s |
| | ACMII-Snowball-3 | 0.05 | 1.00E-04 | 0.7 | 64 | 3 | - | - | - | - | 67.53 | 2.83 | 15.03ms/3.29s |
| Squirrel | ACM-SGC-1 | 0.05 | 0.00E+00 | 0.9 | 64 | - | - | - | - | - | 46.4 | 1.13 | 6.96ms/2.16s |
| | ACM-SGC-2 | 0.05 | 0.00E+00 | 0.9 | 64 | - | - | - | - | - | 40.91 | 1.39 | 35.20ms/10.66s |
| | ACM-GCN | 0.05 | 5.00E-06 | 0.6 | 64 | 2 | - | - | - | - | 58.02 | 1.86 | 14.35ms/2.98s |
| | ACMII-GCN | 0.05 | 0.00E+00 | 0.7 | 64 | 2 | - | - | - | - | 53.76 | 1.63 | 14.08ms/3.39s |
| | ACM-GCNII | 0.01 | 1.00E-05 | 0.5 | 64 | 4 | - | - | 0.5 | 0.1 | 40.9 | 1.58 | 20.72ms/4.17s |
| | ACM-GCNII* | 0.05 | 1.00E-05 | 0.5 | 64 | 4 | - | - | 0.5 | 0.3 | 38.32 | 1.5 | 21.78ms/4.38s |
| | ACM-Snowball-2 | 0.05 | 5.00E-06 | 0.6 | 64 | 2 | - | - | - | - | 55.97 | 2.03 | 15.38ms/3.15s |
| | ACM-Snowball-3 | 0.01 | 1.00E-04 | 0.6 | 64 | 3 | - | - | - | - | 55.73 | 2.39 | 26.15ms/5.94s |
| | ACMII-Snowball-2 | 0.1 | 5.00E-06 | 0.6 | 64 | 2 | - | - | - | - | 53.48 | 0.6 | 15.54ms/3.19s |
| | ACMII-Snowball-3 | 0.05 | 5.00E-05 | 0.5 | 64 | 3 | - | - | - | - | 52.31 | 1.57 | 26.24ms/5.30s |
| Cora | ACM-SGC-1 | 0.01 | 5.00E-06 | 0.9 | 64 | - | - | - | - | - | 86.63 | 1.13 | 6.00ms/7.40s |
| | ACM-SGC-2 | 0.1 | 5.00E-05 | 0.6 | 64 | - | - | - | - | - | 87.64 | 0.99 | 4.85ms/1.17s |
| | ACM-GCN | 0.1 | 5.00E-03 | 0.5 | 64 | 2 | - | - | - | - | 88.62 | 1.22 | 8.84ms/1.81s |
| | ACMII-GCN | 0.1 | 5.00E-04 | 0.4 | 64 | 2 | - | - | - | - | 89 | 0.72 | 8.93ms/1.83s |
| | ACM-GCNII | 0.01 | 1.00E-03 | 0.5 | 64 | 3 | - | - | 1 | 0.2 | 89.1 | 1.61 | 14.07ms/3.04s |
| | ACM-GCNII* | 0.01 | 1.00E-02 | 0.5 | 64 | 4 | - | - | 1 | 0.2 | 89 | 1.35 | 11.36ms/2.48s |
| | ACM-Snowball-2 | 0.05 | 1.00E-03 | 0.6 | 64 | 2 | - | - | - | - | 88.83 | 1.49 | 9.34ms/1.92s |
| | ACM-Snowball-3 | 0.1 | 1.00E-02 | 0.3 | 64 | 3 | - | - | - | - | 89.59 | 1.58 | 13.33ms/2.75s |
| | ACMII-Snowball-2 | 0.1 | 1.00E-03 | 0.5 | 64 | 2 | - | - | - | - | 88.95 | 1.04 | 9.29ms/1.90s |
| | ACMII-Snowball-3 | 0.1 | 5.00E-03 | 0.5 | 64 | 3 | - | - | - | - | 89.36 | 1.26 | 14.18ms/2.89s |
| CiteSeer | ACM-SGC-1 | 0.01 | 5.00E-04 | 0.9 | 64 | - | - | - | - | - | 80.96 | 0.93 | 5.90ms/4.31s |
| | ACM-SGC-2 | 0.05 | 5.00E-04 | 0.9 | 64 | - | - | - | - | - | 80.93 | 1.16 | 5.01ms/1.42s |
| | ACM-GCN | 0.05 | 5.00E-03 | 0.7 | 64 | 2 | - | - | - | - | 81.68 | 0.97 | 11.35ms/2.57s |
| | ACMII-GCN | 0.05 | 5.00E-03 | 0.7 | 64 | 2 | - | - | - | - | 81.58 | 1.77 | 9.55ms/1.94s |
| | ACM-GCNII | 0.01 | 1.00E-02 | 0.5 | 64 | 3 | - | - | 0.5 | 0.3 | 82.28 | 1.12 | 15.61ms/3.56s |
| | ACM-GCNII* | 0.01 | 1.00E-02 | 0.5 | 64 | 3 | - | - | 0.5 | 0.5 | 81.69 | 1.25 | 15.56ms/3.61s |
| | ACM-Snowball-2 | 0.05 | 5.00E-03 | 0.7 | 64 | 2 | - | - | - | - | 81.58 | 1.23 | 11.14ms/2.50s |
| | ACM-Snowball-3 | 0.01 | 5.00E-03 | 0.9 | 64 | 3 | - | - | - | - | 81.32 | 0.97 | 15.91ms/5.36s |
| | ACMII-Snowball-2 | 0.05 | 5.00E-03 | 0.7 | 64 | 2 | - | - | - | - | 82.07 | 1.04 | 10.97ms/2.55s |
| | ACMII-Snowball-3 | 0.05 | 1.00E-04 | 0.6 | 64 | 3 | - | - | - | - | 81.56 | 1.15 | 14.95ms/3.03s |
| PubMed | ACM-SGC-1 | 0.05 | 5.00E-06 | 0.3 | 64 | - | - | - | - | - | 87.75 | 0.88 | 6.04ms/2.61s |
| | ACM-SGC-2 | 0.05 | 5.00E-05 | 0.1 | 64 | - | - | - | - | - | 88.79 | 0.5 | 8.62ms/3.18s |
| | ACM-GCN | 0.1 | 5.00E-04 | 0.2 | 64 | 2 | - | - | - | - | 90.54 | 0.63 | 10.20ms/2.08s |
| | ACMII-GCN | 0.1 | 5.00E-04 | 0.2 | 64 | 2 | - | - | - | - | 90.74 | 0.5 | 10.20ms/2.07s |
| | ACM-GCNII | 0.01 | 1.00E-04 | 0.5 | 64 | 3 | - | - | 1.5 | 0.5 | 90.12 | 0.4 | 15.07ms/3.35s |
| | ACM-GCNII* | 0.01 | 1.00E-04 | 0.5 | 64 | 3 | - | - | 1.5 | 0.5 | 90.18 | 0.51 | 16.62ms/3.72s |
| | ACM-Snowball-2 | 0.1 | 1.00E-04 | 0.3 | 64 | 2 | - | - | - | - | 90.81 | 0.52 | 11.52ms/2.36s |
| | ACM-Snowball-3 | 0.05 | 1.00E-03 | 0.2 | 64 | 3 | - | - | - | - | 91.44 | 0.59 | 18.06ms/3.69s |
| | ACMII-Snowball-2 | 0.1 | 1.00E-04 | 0.3 | 64 | 2 | - | - | - | - | 90.56 | 0.39 | 11.74ms/2.39s |
| | ACMII-Snowball-3 | 0.1 | 5.00E-04 | 0.2 | 64 | 3 | - | - | - | - | 91.31 | 0.6 | 18.61ms/3.88s |
| Deezer-Europe | ACM-SGC-1 | 0.05 | 0,5e-6,1e-5,5e-5 | 0.3 | 64 | - | - | - | - | - | 66.67 | 0.56 | 146.41ms/73.06s |
| | ACM-SGC-2 | 0.002 | 5e-5,1e-4 | 0.3 | 64 | - | - | - | - | - | 66.53 | 0.57 | 195.21ms/97.41s |
| | ACM-GCN | 0.002 | 5.00E-04 | 0.5 | 64 | 2 | - | - | - | - | 67.01 | 0.38 | 136.45ms/68.09s |
| | ACMII-GCN | 0.01 | 5.00E-05 | 0.8 | 64 | 2 | - | - | - | - | 67.15 | 0.41 | 135.24ms/67.48s |
| | ACM-GCNII | 0.01 | 0,5e-6 | 0.5 | 64 | 1 | - | - | 0.5 | 0.4 | 66.39 | 0.56 | 80.82ms/40.33s |
| | ACM-GCNII* | 0.01 | 0.0001,1e-3 | 0.5 | 64 | 1 | - | - | 1.5 | 0.2 | 66.6 | 0.57 | 80.95ms/40.40s |

Table 11: Optimal hyperparameters for ACM(II)-GNNs on random 60%/20%/20% splits

| Datasets | Models\Hyperparameters | lr | weight_decay | dropout | hidden | with A | results | std | average epoch time/average total time |
|---|---|---|---|---|---|---|---|---|---|
| **Cornell** | ACM-GCN+ | 0.05 | 1.00E-02 | 0.1 | 64 | Y | 94.92 | 2.79 | 16.66ms/3.37s |
| | ACMII-GCN+ | 0.05 | 1.00E-02 | 0.3 | 64 | Y | 93.93 | 1.05 | 12.55ms/2.56s |
| | ACM-GCN++(with xX) | 0.1 | 5.00E-03 | 0.4 | 64 | N | 93.93 | 3.03 | 12.89ms/2.62s |
| | ACMII-GCN++(with xX) | 0.05 | 1.00E-02 | 0.6 | 64 | Y | 92.62 | 2.57 | 18.25ms/3.69s |
| **Wisconsin** | ACM-GCN+ | 0.05 | 1.00E-02 | 0.3 | 64 | Y | 96.5 | 2.08 | 16.54ms/3.35s |
| | ACMII-GCN+ | 0.01 | 1.00E-02 | 0.1 | 64 | N | 97.5 | 1.25 | 12.09ms/2.88s |
| | ACM-GCN++(with xX) | 0.05 | 1.00E-02 | 0.1 | 64 | Y | 96.75 | 1.79 | 18.12ms/3.66s |
| | ACMII-GCN++(with xX) | 0.01 | 1.00E-02 | 0.1 | 64 | Y | 97.13 | 1.68 | 17.32ms/3.53s |
| **Texas** | ACM-GCN+ | 0.05 | 1.00E-03 | 0.3 | 64 | N | 94.92 | 2.79 | 12.05ms/2.44s |
| | ACMII-GCN+ | 0.05 | 1.00E-02 | 0.1 | 64 | Y | 96.56 | 2 | 22.63ms/4.58s |
| | ACM-GCN++(with xX) | 0.05 | 5.00E-04 | 0.2 | 64 | N | 95.41 | 2.82 | 13.20ms/2.67s |
| | ACMII-GCN++(with xX) | 0.05 | 5.00E-04 | 0.1 | 64 | N | 94.75 | 2.91 | 12.82ms/2.60s |
| **Film** | ACM-GCN+ | 0.01 | 1.00E-03 | 0.8 | 64 | N | 41.79 | 1.01 | 13.57ms/3.59s |
| | ACMII-GCN+ | 0.1 | 5.00E-05 | 0.7 | 64 | N | 41.86 | 1.48 | 13.38ms/3.59s |
| | ACM-GCN++(with xX) | 0.002 | 5.00E-03 | 0.9 | 64 | N | 41.5 | 1.54 | 13.76ms/2.77s |
| | ACMII-GCN++(with xX) | 0.002 | 5.00E-03 | 0.9 | 64 | N | 41.66 | 1.42 | 13.67ms/2.77s |
| **Chameleon** | ACM-GCN+ | 0.002 | 1.00E-03 | 0.4 | 64 | Y | 76.08 | 2.13 | 18.19ms/8.60s |
| | ACMII-GCN+ | 0.1 | 1.00E-04 | 0.7 | 64 | Y | 75.23 | 1.72 | 17.39ms/3.57s |
| | ACM-GCN++(with xX) | 0.1 | 5.00E-05 | 0.8 | 64 | Y | 75.51 | 1.58 | 18.69ms/4.17s |
| | ACMII-GCN++(with xX) | 0.01 | 1.00E-04 | 0.8 | 64 | Y | 75.93 | 1.71 | 18.70ms/4.53s |
| **Squirrel** | ACM-GCN+ | 0.01 | 1.00E-04 | 0.6 | 64 | Y | 69.26 | 1.11 | 24.71ms/4.97s |
| | ACMII-GCN+ | 0.01 | 1.00E-04 | 0.6 | 64 | Y | 68.56 | 1.33 | 21.21ms/4.26s |
| | ACM-GCN++(with xX) | 0.002 | 1.00E-03 | 0.7 | 64 | Y | 69.81 | 1.11 | 22.14ms/5.34s |
| | ACMII-GCN++(with xX) | 0.002 | 1.00E-04 | 0.7 | 64 | Y | 69.98 | 1.53 | 21.78ms/4.38s |
| **Cora** | ACM-GCN+ | 0.1 | 5.00E-03 | 0.3 | 64 | Y | 89.75 | 1.16 | 17.29ms/3.52s |
| | ACMII-GCN+ | 0.1 | 5.00E-03 | 0.5 | 64 | Y | 89.33 | 0.81 | 18.08ms/3.69s |
| | ACM-GCN++(with xX) | 0.05 | 5.00E-03 | 0.4 | 64 | Y | 89.18 | 1.11 | 18.21ms/3.69s |
| | ACMII-GCN++(with xX) | 0.1 | 1.00E-02 | 0.1 | 64 | Y | 89.47 | 1.08 | 18.53ms/3.76s |
| **CiteSeer** | ACM-GCN+ | 0.1 | 1.00E-05 | 0.5 | 64 | N | 81.65 | 1.48 | 12.44ms/2.50s |
| | ACMII-GCN+ | 0.002 | 5.00E-03 | 0.8 | 64 | N | 81.83 | 1.65 | 14.87ms/15.36s |
| | ACM-GCN++(with xX) | 0.05 | 5.00E-03 | 0.3 | 64 | N | 81.87 | 1.38 | 13.35ms/2.86s |
| | ACMII-GCN++(with xX) | 0.01 | 5.00E-04 | 0.9 | 64 | N | 81.76 | 1.25 | 14.04ms/3.88s |
| **PubMed** | ACM-GCN+ | 0.1 | 1.00E-04 | 0.1 | 64 | N | 90.46 | 0.69 | 15.15ms/3.09s |
| | ACMII-GCN+ | 0.1 | 1.00E-04 | 0.3 | 64 | N | 90.39 | 0.33 | 17.36ms/3.55s |
| | ACM-GCN++(with xX) | 0.1 | 1.00E-04 | 0.1 | 64 | N | 90.96 | 0.62 | 16.35ms/3.47s |
| | ACMII-GCN++(with xX) | 0.1 | 1.00E-04 | 0.3 | 64 | N | 90.63 | 0.56 | 16.18ms/3.39s |
| **Deezer-Europe** | ACM-GCN+ | 0.002 | 1.00E-06 | 0.7 | 64 | N | 67.4 | 0.44 | 281.97ms/140.70s |
| | ACMII-GCN+ | 0.002 | 1.00E-04 | 0.8 | 64 | N | 67.3 | 0.48 | 281.48ms/140.46s |
| | ACM-GCN++(with xX) | 0.002 | 1.00E-03 | 0.7 | 64 | Y | 67.44 | 0.31 | 332.92ms/166.13s |
| | ACMII-GCN++(with xX) | 0.002 | 1.00E-05 | 0.8 | 64 | N | 67.5 | 0.53 | 326.09ms/162.72s |

Table 12: Optimal hyperparameters for ACM(II)-GCN+ and ACM(II)-GCN++ on random 60%/20%/20% splits

| Datasets | Models\Hyperparameters | lr | weight_decay | dropout | hidden | results | std | average epoch time/average total time |
|---|---|---|---|---|---|---|---|---|
| **Cornell** | ACM-SGC-1 | 0.01 | 5.00E-06 | 0 | 64 | 82.43 | 5.44 | 5.37ms/23.05s |
| | ACM-SGC-2 | 0.01 | 5.00E-06 | 0 | 64 | 82.43 | 5.44 | 5.93ms/25.66s |
| | ACM-GCN | 0.05 | 5.00E-04 | 0.5 | 64 | 85.14 | 6.07 | 8.04ms/1.67s |
| | ACMII-GCN | 0.1 | 1.00E-04 | 0 | 64 | 85.95 | 5.64 | 7.83ms/2.66s |
| | FAGCN | 0.01 | 1.00E-04 | 0.6 | 64 | 76.76 | 5.87 | 8.80ms/7.67s |
| | ACM-Snowball-2 | 0.05 | 5.00E-03 | 0.3 | 64 | 85.41 | 5.43 | 11.50ms/2.35s |
| | ACM-Snowball-3 | 0.05 | 5.00E-03 | 0.2 | 64 | 83.24 | 5.38 | 15.06ms/3.12s |
| | ACMII-Snowball-2 | 0.1 | 5.00E-03 | 0.2 | 64 | 85.68 | 5.93 | 12.63ms/2.58s |
| | ACMII-Snowball-3 | 0.05 | 5.00E-03 | 0.2 | 64 | 82.7 | 4.86 | 14.59ms/3.06s |
| **Wisconsin** | ACM-SGC-1 | 0.1 | 5.00E-06 | 0 | 64 | 86.47 | 3.77 | 5.07ms/14.07s |
| | ACM-SGC-2 | 0.1 | 5.00E-06 | 0 | 64 | 86.47 | 3.77 | 5.30ms/16.05s |
| | ACM-GCN | 0.05 | 1.00E-03 | 0.4 | 64 | 88.43 | 3.22 | 8.04ms/1.66s |
| | ACMII-GCN | 0.01 | 5.00E-05 | 0.1 | 64 | 87.45 | 3.74 | 8.40ms/2.19s |
| | FAGCN | 0.01 | 5.00E-05 | 0.5 | 64 | 79.61 | 1.59 | 8.61ms/5.84s |
| | ACM-Snowball-2 | 0.01 | 1.00E-03 | 0.4 | 64 | 87.06 | 2 | 12.51ms/2.60s |
| | ACM-Snowball-3 | 0.01 | 1.00E-02 | 0.1 | 64 | 86.67 | 4.37 | 14.92ms/3.15s |
| | ACMII-Snowball-2 | 0.01 | 5.00E-04 | 0.1 | 64 | 87.45 | 2.8 | 11.96ms/2.63s |
| | ACMII-Snowball-3 | 0.01 | 5.00E-03 | 0.5 | 64 | 85.29 | 4.23 | 14.87ms/3.10s |
| **Texas** | ACM-SGC-1 | 0.01 | 1.00E-05 | 0 | 64 | 81.89 | 4.53 | 5.34ms/19.00s |
| | ACM-SGC-2 | 0.05 | 1.00E-05 | 0 | 64 | 81.89 | 4.53 | 5.50ms/9.26s |
| | ACM-GCN | 0.05 | 5.00E-04 | 0.5 | 64 | 87.84 | 4.4 | 9.62ms/1.99s |
| | ACMII-GCN | 0.01 | 1.00E-03 | 0.1 | 64 | 86.76 | 4.75 | 9.98ms/2.22s |
| | FAGCN | 0.01 | 1.00E-05 | 0 | 64 | 76.49 | 2.87 | 10.45ms/5.70s |
| | ACM-Snowball-2 | 0.01 | 5.00E-03 | 0.2 | 64 | 87.57 | 4.86 | 11.56ms/2.45s |
| | ACM-Snowball-3 | 0.01 | 5.00E-03 | 0.2 | 64 | 87.84 | 3.87 | 15.17ms/3.15s |
| | ACMII-Snowball-2 | 0.01 | 1.00E-03 | 0.2 | 64 | 86.76 | 4.43 | 11.36ms/2.30 |
| | ACMII-Snowball-3 | 0.01 | 5.00E-03 | 0.6 | 64 | 85.41 | 6.42 | 15.84ms/3.48s |
| **Film** | ACM-SGC-1 | 0.05 | 5.00E-04 | 0 | 64 | 35.49 | 1.06 | 5.39ms/1.17s |
| | ACM-SGC-2 | 0.05 | 5.00E-04 | 0.1 | 64 | 36.04 | 0.83 | 13.22ms/3.31s |
| | ACM-GCN | 0.01 | 5.00E-03 | 0 | 64 | 36.28 | 1.09 | 8.96ms/1.82s |
| | ACMII-GCN | 0.01 | 5.00E-03 | 0 | 64 | 36.16 | 1.11 | 9.06ms/1.83s |
| | FAGCN | 0.01 | 5.00E-05 | 0.4 | 64 | 34.82 | 1.35 | 15.60ms/2.51s |
| | ACM-Snowball-2 | 0.01 | 1.00E-02 | 0 | 64 | 36.89 | 1.18 | 14.77ms/3.01s |
| | ACM-Snowball-3 | 0.01 | 1.00E-02 | 0.2 | 64 | 36.82 | 0.94 | 16.57ms/3.36s |
| | ACMII-Snowball-2 | 0.01 | 5.00E-03 | 0.1 | 64 | 36.55 | 1.24 | 12.76ms/2.57s |
| | ACMII-Snowball-3 | 0.05 | 5.00E-03 | 0.3 | 64 | 36.49 | 1.41 | 16.51ms/3.49s |
| **Chameleon** | ACM-SGC-1 | 0.1 | 5.00E-06 | 0.9 | 64 | 63.99 | 1.66 | 5.92ms/1.74s |
| | ACM-SGC-2 | 0.1 | 0.00E+00 | 0.9 | 64 | 59.21 | 2.22 | 8.84ms/1.78s |
| | ACM-GCN | 0.05 | 5.00E-05 | 0.7 | 64 | 66.93 | 1.85 | 8.40ms/1.71s |
| | ACMII-GCN | 0.05 | 5.00E-06 | 0.8 | 64 | 66.91 | 2.55 | 8.90ms/2.10s |
| | FAGCN | 0.01 | 5.00E-05 | 0 | 64 | 46.07 | 2.11 | 16.90ms/7.94s |
| | ACM-Snowball-2 | 0.01 | 1.00E-04 | 0.7 | 64 | 67.08 | 2.04 | 12.50ms/2.69s |
| | ACM-Snowball-3 | 0.01 | 1.00E-05 | 0.8 | 64 | 66.91 | 1.73 | 16.12ms/4.91s |
| | ACMII-Snowball-2 | 0.01 | 5.00E-05 | 0.8 | 64 | 66.49 | 1.75 | 12.65ms/3.42s |
| | ACMII-Snowball-3 | 0.05 | 5.00E-05 | 0.7 | 64 | 66.86 | 1.74 | 17.60ms/4.06s |
| **Squirrel** | ACM-SGC-1 | 0.05 | 5.00E-06 | 0.9 | 64 | 45 | 1.4 | 6.10ms/2.18s |
| | ACM-SGC-2 | 0.05 | 0.00E+00 | 0.9 | 64 | 40.02 | 0.96 | 35.75ms/9.62s |
| | ACM-GCN | 0.05 | 5.00E-06 | 0.7 | 64 | 54.4 | 1.88 | 10.48ms/2.68s |
| | ACMII-GCN | 0.05 | 5.00E-06 | 0.7 | 64 | 51.85 | 1.38 | 11.69ms/2.91s |
| | FAGCN | 0 | 5.00E-03 | 0 | 64 | 30.86 | 0.69 | 10.90ms/13.91s |
| | ACM-Snowball-2 | 0.01 | 1.00E-04 | 0.7 | 64 | 52.5 | 1.49 | 17.89ms/5.78s |
| | ACM-Snowball-3 | 0.01 | 5.00E-05 | 0.7 | 64 | 53.31 | 1.88 | 22.60ms/7.53s |
| | ACMII-Snowball-2 | 0.05 | 5.00E-05 | 0.6 | 64 | 50.15 | 1.4 | 16.95ms/3.45s |
| | ACMII-Snowball-3 | 0.01 | 5.00E-04 | 0.6 | 64 | 48.87 | 1.23 | 23.52ms/4.94s |
| **Cora** | ACM-SGC-1 | 0.05 | 5.00E-05 | 0.7 | 64 | 86.9 | 1.38 | 4.99ms/2.40s |
| | ACM-SGC-2 | 0.1 | 0 | 0.8 | 64 | 87.69 | 1.07 | 5.16ms/1.16s |
| | ACM-GCN | 0.01 | 5.00E-05 | 0.6 | 64 | 87.91 | 0.95 | 8.41ms/1.84s |
| | ACMII-GCN | 0.01 | 1.00E-04 | 0.6 | 64 | 88.01 | 1.08 | 8.59ms/1.96s |
| | FAGCN | 0.02 | 1.00E-04 | 0.5 | 64 | 88.05 | 1.57 | 9.30ms/10.64s |
| | ACM-Snowball-2 | 0.01 | 1.00E-03 | 0.5 | 64 | 87.42 | 1.09 | 12.54ms/2.72s |
| | ACM-Snowball-3 | 0.01 | 5.00E-06 | 0.9 | 64 | 87.1 | 0.93 | 15.83ms/11.33s |
| | ACMII-Snowball-2 | 0.01 | 1.00E-03 | 0.6 | 64 | 87.57 | 0.86 | 12.06ms/2.64s |
| | ACMII-Snowball-3 | 0.01 | 5.00E-03 | 0.5 | 64 | 87.16 | 1.01 | 16.29ms/3.62s |
| **CiteSeer** | ACM-SGC-1 | 0.05 | 0.00E+00 | 0.7 | 64 | 76.73 | 1.59 | 5.24ms/1.14s |
| | ACM-SGC-2 | 0.1 | 0.00E+00 | 0.8 | 64 | 76.59 | 1.69 | 5.14ms/1.03s |
| | ACM-GCN | 0.01 | 5.00E-06 | 0.3 | 64 | 77.32 | 1.7 | 8.89ms/1.79s |
| | ACMII-GCN | 0.01 | 5.00E-05 | 0.5 | 64 | 77.15 | 1.45 | 8.95ms/1.80s |
| | FAGCN | 0.02 | 5.00E-05 | 0.4 | 64 | 77.07 | 2.05 | 10.05ms/5.69s |
| | ACM-Snowball-2 | 0.01 | 5.00E-05 | 0 | 64 | 76.41 | 1.38 | 12.87ms/2.59s |
| | ACM-Snowball-3 | 0.01 | 5.00E-06 | 0.9 | 64 | 75.91 | 1.57 | 17.40ms/11.92s |
| | ACMII-Snowball-2 | 0.01 | 5.00E-03 | 0.5 | 64 | 76.92 | 1.45 | 13.10ms/2.94s |
| | ACMII-Snowball-3 | 0.1 | 5.00E-05 | 0.9 | 64 | 76.18 | 1.55 | 17.47ms/5.88s |
| **PubMed** | ACM-SGC-1 | 0.05 | 5.00E-06 | 0.4 | 64 | 88.49 | 0.51 | 5.77ms/3.65s |
| | ACM-SGC-2 | 0.05 | 5.00E-06 | 0.3 | 64 | 89.01 | 0.6 | 8.50ms/5.18s |
| | ACM-GCN | 0.01 | 5.00E-05 | 0.4 | 64 | 90 | 0.52 | 8.99ms/2.51s |
| | ACMII-GCN | 0.01 | 1.00E-04 | 0.3 | 64 | 89.89 | 0.43 | 9.70ms/2.57s |
| | FAGCN | 0.01 | 1.00E-04 | 0 | 64 | 88.09 | 1.38 | 10.30ms/8.75s |
| | ACM-Snowball-2 | 0.01 | 1.00E-03 | 0.3 | 64 | 89.89 | 0.57 | 15.05ms/3.11s |
| | ACM-Snowball-3 | 0.01 | 5.00E-03 | 0.1 | 64 | 89.81 | 0.43 | 20.51ms/4.63s |
| | ACMII-Snowball-2 | 0.01 | 5.00E-04 | 0.4 | 64 | 89.84 | 0.48 | 15.10ms/3.2s |
| | ACMII-Snowball-3 | 0.01 | 1.00E-03 | 0.4 | 64 | 89.73 | 0.52 | 20.46ms/4.32s |

Table 13: Optimal hyperparameters for FAGCN and ACM(II)-GNNs on fixed 48%/32%/20% splits

| Datasets | Models\Hyperparameters | lr | weight_decay | dropout | hidden | with A | results | std | average epoch time/average total time |
|---|---|---|---|---|---|---|---|---|---|
| Cornell | ACM-GCN+ | 0.05 | 1.00E-03 | 0.1 | 64 | N | 85.68 | 4.84 | 10.86ms/2.28s |
| | ACMII-GCN+ | 0.05 | 5.00E-03 | 0 | 64 | Y | 85.41 | 5.3 | 14.42ms/2.97s |
| | ACM-GCN++ | 0.01 | 5.00E-04 | 0.1 | 64 | N | 85.68 | 5.8 | 14.15ms/3.17s |
| | ACMII-GCN++ | 0.01 | 5.00E-03 | 0.3 | 64 | N | 86.49 | 6.73 | 14.11ms/3.19s |
| Wisconsin | ACM-GCN+ | 0.01 | 1.00E-03 | 0.1 | 64 | Y | 88.43 | 2.39 | 14.50ms/3.18s |
| | ACMII-GCN+ | 0.01 | 5.00E-03 | 0.3 | 64 | Y | 88.04 | 3.66 | 17.71ms/3.75s |
| | ACM-GCN++ | 0.05 | 5.00E-03 | 0.1 | 64 | Y | 88.24 | 3.16 | 20.61ms/4.29s |
| | ACMII-GCN++ | 0.01 | 5.00E-03 | 0.2 | 64 | Y | 88.43 | 3.66 | 18.28ms/3.75s |
| Texas | ACM-GCN+ | 0.01 | 5.00E-04 | 0.2 | 64 | Y | 88.38 | 3.64 | 22.63ms/4.63s |
| | ACMII-GCN+ | 0.05 | 1.00E-02 | 0.4 | 64 | Y | 88.11 | 3.24 | 16.92ms/3.44s |
| | ACM-GCN++ | 0.01 | 5.00E-03 | 0.3 | 64 | Y | 88.38 | 3.43 | 20.69ms/4.25s |
| | ACMII-GCN++ | 0.01 | 5.00E-03 | 0.6 | 64 | Y | 88.38 | 3.43 | 18.58ms/3.84s |
| Film | ACM-GCN+ | 0.05 | 5.00E-03 | 0 | 64 | N | 36.13 | 1.19 | 18.33ms/3.68s |
| | ACMII-GCN+ | 0.05 | 5.00E-03 | 0 | 64 | N | 35.95 | 1.33 | 19.07ms/3.83s |
| | ACM-GCN++ | 0.01 | 5.00E-03 | 0 | 64 | N | 37.31 | 1.09 | 18.57ms/3.73s |
| | ACMII-GCN++ | 0.01 | 5.00E-03 | 0 | 64 | N | 36.68 | 1.35 | 15.79ms/3.17s |
| Chameleon | ACM-GCN+ | 0.05 | 1.00E-04 | 0.7 | 64 | Y | 74.23 | 2.25 | 25.31ms/5.14s |
| | ACMII-GCN+ | 0.05 | 1.00E-04 | 0.7 | 64 | Y | 74.3 | 2.03 | 25.04ms/5.04s |
| | ACM-GCN++ | 0.002 | 5.00E-04 | 0.8 | 64 | Y | 74.3 | 2.23 | 19.44ms/8.58s |
| | ACMII-GCN++ | 0.01 | 1.00E-04 | 0.8 | 64 | Y | 74.45 | 1.34 | 21.24ms/4.92s |
| Squirrel | ACM-GCN+ | 0.002 | 1.00E-04 | 0.6 | 64 | Y | 66.06 | 2.16 | 36.96ms/7.82s |
| | ACMII-GCN+ | 0.01 | 5.00E-04 | 0.8 | 64 | Y | 65.95 | 1.74 | 35.56ms/9.18s |
| | ACM-GCN++ | 0.01 | 1.00E-04 | 0.8 | 64 | Y | 66.45 | 1.83 | 26.34ms/6.20s |
| | ACMII-GCN++ | 0.002 | 5.00E-04 | 0.8 | 64 | Y | 66.75 | 1.82 | 24.55ms/10.49s |
| Cora | ACM-GCN+ | 0.002 | 0.00E+00 | 0.6 | 64 | N | 88.05 | 0.99 | 15.21ms/5.00s |
| | ACMII-GCN+ | 0.002 | 5.00E-05 | 0.7 | 64 | Y | 88.19 | 1.17 | 13.74ms/5.67s |
| | ACM-GCN++ | 0.002 | 5.00E-06 | 0.7 | 64 | N | 88.11 | 0.96 | 14.59ms/5.05s |
| | ACMII-GCN++ | 0.002 | 5.00E-05 | 0.7 | 64 | N | 88.25 | 0.96 | 15.75ms/5.87s |
| CiteSeer | ACM-GCN+ | 0.01 | 5.00E-05 | 0.3 | 64 | N | 77.67 | 1.19 | 17.36ms/3.49s |
| | ACMII-GCN+ | 0.01 | 5.00E-03 | 0.2 | 64 | Y | 77.2 | 1.61 | 22.99ms/4.74s |
| | ACM-GCN++ | 0.002 | 5.00E-06 | 0.6 | 64 | N | 77.46 | 1.65 | 14.51ms/3.88s |
| | ACMII-GCN++ | 0.01 | 5.00E-05 | 0.6 | 64 | N | 77.12 | 1.58 | 18.69ms/3.76s |
| PubMed | ACM-GCN+ | 0.05 | 5.00E-05 | 0.3 | 64 | N | 89.82 | 0.41 | 24.63ms/4.95s |
| | ACMII-GCN+ | 0.01 | 1.00E-04 | 0.3 | 64 | N | 89.78 | 0.49 | 25.10ms/5.61s |
| | ACM-GCN++ | 0.01 | 5.00E-05 | 0.3 | 64 | N | 89.65 | 0.58 | 18.36ms/3.76s |
| | ACMII-GCN++ | 0.002 | 5.00E-06 | 0.4 | 64 | N | 89.71 | 0.48 | 16.98ms/9.44s |

Table 14: Optimal hyperparameters for ACM(II)-GCN+ and ACM(II)-GCN++ on fixed 48%/32%/20% splits

# D    Experimental Setup and Further Discussion on Synthetic Graphs

## D.1    Detailed Description of Data Generation Process

- For each node $v$, we first randomly generate its degree $d_v$.

- Given $d_v$, for any $h$, we sample $hd_v$ intra-class edges and $(1-h)d_v$ inter-class edges.

More specifically in our synthetic experiments, for a given $h$,

- we generate node degree $d_v$ for nodes in each class from multinomial distribution with `numpy.random.multinomial(800/h, numpy.ones(400)/400, size=1)[0]`.

- For a sampled $d_v$, we generate intra-class edges from (does not include self-loop) `numpy.random.multinomial(hd_v, numpy.ones(399)/399, size=1)[0]` and inter-class edges from `numpy.random.multinomial((1-h) d_v, numpy.ones(1600)/1600, size=1)[0]`.

For each generated graph, we calculate their $H_{\text{node}}, H_{\text{class}}, H_{\text{agg}}^M$. Then, we reorder the value of the metrics in ascend order for x-axis and plot the corresponding test accuracy.

Here is a simplified example of how we draw Figure 2. Suppose we generate 3 graphs with $H_{\text{edge}} = 0.1, 0.5, 0.9$, the test accuracy of GCN on these 3 synthetic graphs are $0.8, 0.5, 0.9$. For the generated graphs, we calculate their $H_{\text{agg}}^M$, and suppose we get $H_{\text{agg}}^M = 0.7, 0.4, 0.8$. Then we will draw the performance of GCN under $H_{\text{agg}}^M$ with ascend x-axis order $[0.4, 0.7, 0.8]$ and the corresponding reordered y-axis is $[0.5, 0.8, 0.9]$. Other figures are drawn with the same process.

## D.2    Model Comparison on Synthetic Graphs

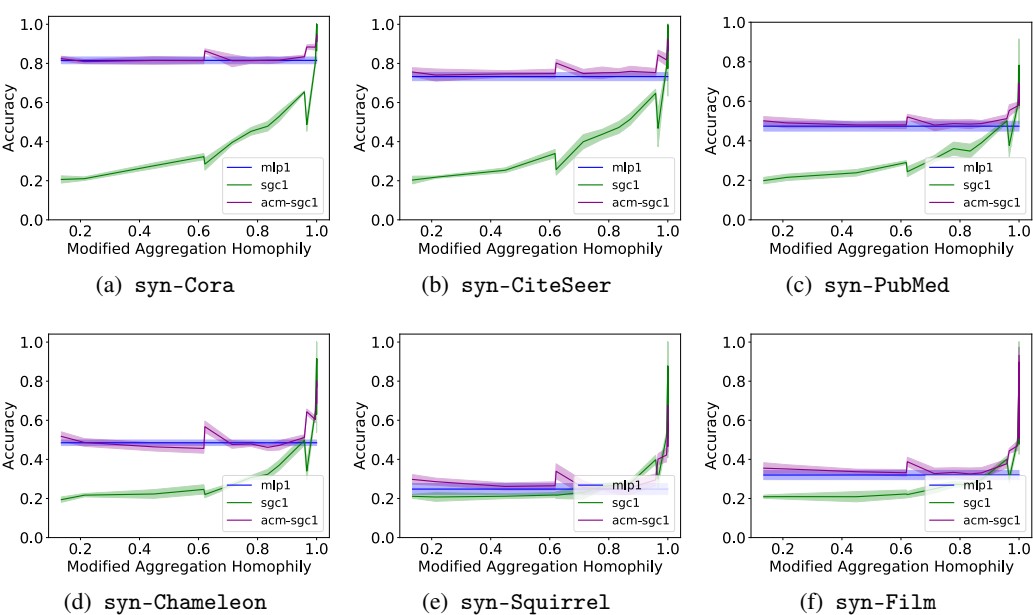

Figure 10: Comparison of test accuracy (mean $\pm$ std) of MLP-1, SGC-1 and ACM-SGC-1 on synthetic datasets

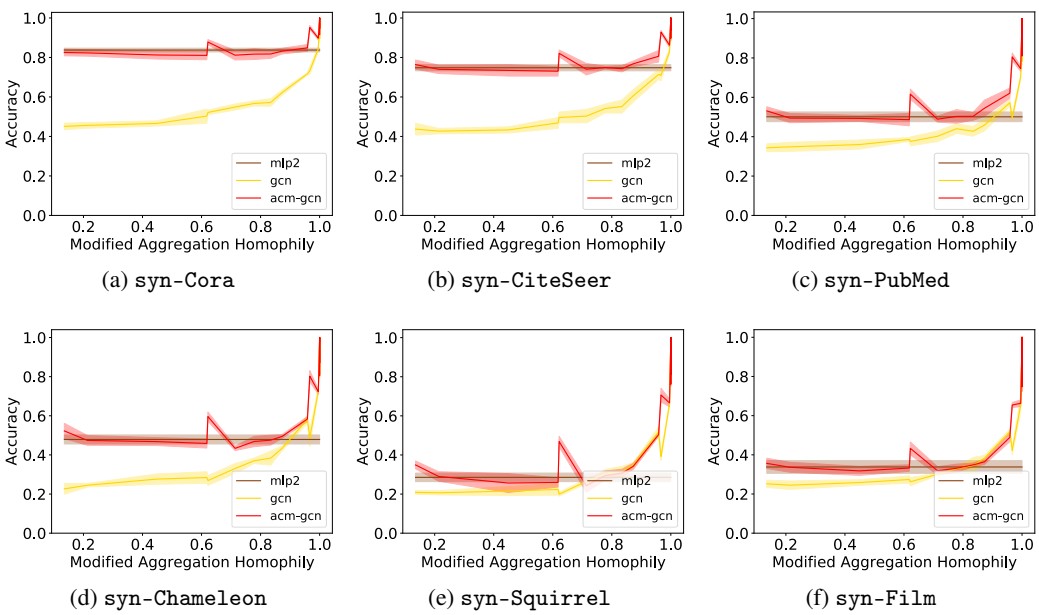

(a) `syn-Cora`     (b) `syn-CiteSeer`     (c) `syn-PubMed`

(d) `syn-Chameleon`     (e) `syn-Squirrel`     (f) `syn-Film`

Figure 11: Comparison of test accuracy (mean $\pm$ std) of MLP-2, GCN and ACM-GCN on synthetic datasets

In order to separate the effects of nonlinearity and graph structure, we compare SGC with 1 hop (sgc-1) with MLP-1 (linear model). For GCN which includes nonlinearity, we use MLP-2 as its corresponding graph-agnostic baseline model. We train the above GNN models, graph-agnostic baseline models and ACM-GNN models on all synthetic datasets and plot the mean test accuracy with standard deviation on each dataset. From Figure 10 and Figure 11, we can see that on each $H_{\text{agg}}^{M}(\mathcal{G})$ level, ACM-GNNs will not underperform baseline GNNs and the graph-agnostic models. But when $H_{\text{agg}}^{M}(\mathcal{G})$ is small, baseline GNNs will be outperformed by graph-agnostic models by a large margin. This demonstrate that the ACM framework can help GNNs to perform well on harmful graphs while keep competitive on less harmful graphs.

### D.3 Further Discussion of Aggregation Homophily on Regular Graphs

We notice that in Figure 2(a), the performance of SGC-1 and GCN both have a turning point, *i.e.,* when $H_{\text{edge}}(\mathcal{G})$ is smaller than a certain value, the performance will get better instead of getting worse. With some extra restriction on node degree in data generation process, we find that this interesting phenomenon can be theoretically explained by the following proposition 1 based on our proposed similarity matrix which can verify the usefulness of $H_{\text{agg}}^{M}(\mathcal{G})$. We first generate regular graphs ,*i.e.,* each node has the same degree, as follows,

**Generate Synthetic Regular Graphs**    We first generate 180 graphs in total with 18 edge homophily levels varied from 0.05 to 0.9, each corresponding to 10 graphs. For every generated graph, we have 5 classes with 400 nodes in each class. For each node, we randomly generate 10 intra-class edges and $[\frac{10}{H_{\text{edge}}(\mathcal{G})} - 10]$ inter-class edges. The features of nodes in each class are sampled from node features in the corresponding class of the base dataset. Nodes are randomly split into 60%/20%/20% for train/validation/test. We train 1-hop SGC (*sgc-1*) [42] and GCN [19] on synthetic data (see Appendix C.1 for hyperparameter searching range). For each value of $H_{\text{edge}}(\mathcal{G})$, we take the average test accuracy and standard deviation of runs over 10 generated graphs. We plot the performance curves in Figure 12.

From Figure 12 we can see that the turning point is a bit less than 0.2. We derive the following proposition for $d$-regular graph to explain and predict it.

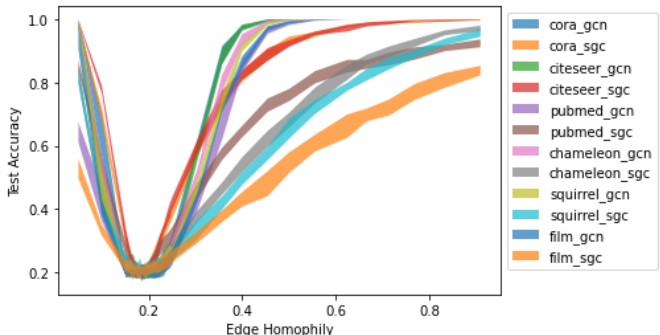

Figure 12: Synthetic experiments for edge homophily on regular graphs.

**Proposition 1.** (See Appendix F for proof). Suppose there are $C$ classes in the graph $\mathcal{G}$ and $\mathcal{G}$ is a $d$-regular graph (each node has $d$ neighbors). Given $d$, edges for each node are *i.i.d.* generated, such that each edge of any node has probability $h$ to connect with nodes in the same class and probability $1 - h$ to connect with nodes in different classes. Let the aggregation operator $\hat{A} = \hat{A}_{\text{rw}}$. Then, for nodes $v$, $u_1$ and $u_2$, where $Z_{u_1,:} = Z_{v,:}$ and $Z_{u_2,:} \neq Z_{v,:}$, we have

$$g(h) \equiv \mathbb{E}\left(S(\hat{A}, Z)_{v,u_1}\right) - \mathbb{E}\left(S(\hat{A}, Z)_{v,u_2}\right) = \left(\frac{(C-1)(hd+1) - (1-h)d}{(C-1)(d+1)}\right)^2 \quad (12)$$

and the minimum of $g(h)$ is reached at

$$h = \frac{d + 1 - C}{Cd} = \frac{d_{\text{intra}}/h + 1 - C}{C(d_{\text{intra}}/h)} \Rightarrow h = \frac{d_{\text{intra}}}{Cd_{\text{intra}} + C - 1}$$

where $d_{\text{intra}} = dh$, which is the expected number of neighbors of a node that have the same label as the node.

The value of $g(h)$ in equation 12 is the expected differences of the similarity values between nodes in the same class as $v$ and nodes in other classes. $g(h)$ is strongly related to the definition of aggregation homophily and its minimum potentially implies the turning point of performance curves. In the synthetic experiments, we have $d_{\text{intra}} = 10, C = 5$ and the minimum of $g(h)$ is reached at $h = 5/27 \approx 0.1852$, which corresponds to the lowest point in the performance curve in Figure 12. In other words, the $H_{\text{edge}}(\mathcal{G})$ where SGC-1 and GCN perform worst is where $g(h)$ gets the smallest value, instead of the point with the smallest edge homophily value, *i.e.*, $H_{\text{edge}}(\mathcal{G}) = 0$. This reveals the advantage of $H_{\text{agg}}(\mathcal{G})$ over $H_{\text{edge}}(\mathcal{G})$ by taking use of the similarity matrix.

# E   Details of Gradient Calculation in equation 6

## E.1   Derivation in Matrix Form

This derivation is similar to [31].

In output layer, we have

$$Y = \text{softmax}(\hat{A}XW) \equiv \text{softmax}(Y') = \left(\exp(Y')\mathbf{1}_C\mathbf{1}_C^T\right)^{-1} \odot \exp(Y') > 0$$
$$\mathcal{L} = -\text{trace}(Z^T \log Y)$$

where $\mathbf{1}_C \in \mathcal{R}^{C \times 1}$, $(\cdot)^{-1}$ is point-wise inverse function and each element of $Y$ is positive. Then

$$d\mathcal{L} = -\text{trace}\left(Z^T((Y)^{-1} \odot dY)\right) = -\text{trace}\left(Z^T\left((\text{softmax}(Y'))^{-1} \odot d\,\text{softmax}(Y')\right)\right)$$

Note that

$$
\begin{aligned}
d\,\mathrm{softmax}(Y') =& - \left(\exp(Y')\mathbf{1}_C\mathbf{1}_C^T\right)^{-2} \odot [(\exp(Y') \odot dY')\mathbf{1}_C\mathbf{1}_C^T] \odot \exp(Y') \\
& + \left(\exp(Y')\mathbf{1}_C\mathbf{1}_C^T\right)^{-1} \odot (\exp(Y') \odot dY') \\
=& - \mathrm{softmax}(Y') \odot \left(\exp(Y')\mathbf{1}_C\mathbf{1}_C^T\right)^{-1} \odot [(\exp(Y') \odot dY')\mathbf{1}_C\mathbf{1}_C^T] \\
& + \mathrm{softmax}(Y') \odot dY' \\
=& \ \mathrm{softmax}(Y') \odot \left( - \left(\exp(Y')\mathbf{1}_C\mathbf{1}_C^T\right)^{-1} \odot [(\exp(Y') \odot dY')\mathbf{1}_C\mathbf{1}_C^T] + dY'\right)
\end{aligned}
$$

Then,

$$
\begin{aligned}
d\mathcal{L} =& -\mathrm{trace}\left(Z^T\left((\mathrm{softmax}(Y'))^{-1} \odot \left[\mathrm{softmax}(Y') \odot \left( - \left(\exp(Y')\mathbf{1}_C\mathbf{1}_C^T\right)^{-1}\right.\right.\right.\right. \\
& \left.\left.\left.\left. \odot [(\exp(Y') \odot dY')\mathbf{1}_C\mathbf{1}_C^T] + dY'\right)\right]\right)\right) \\
=& -\mathrm{trace}\left(Z^T\left( - \left(\exp(Y')\mathbf{1}_C\mathbf{1}_C^T\right)^{-1} \odot [(\exp(Y') \odot dY')\mathbf{1}_C\mathbf{1}_C^T] + dY'\right)\right) \\
=& \ \mathrm{trace}\left(\left(\left(Z \odot \left(\exp(Y')\mathbf{1}_C\mathbf{1}_C^T\right)^{-1}\right)\mathbf{1}_C\mathbf{1}_C^T\right)^T [\exp(Y') \odot dY'] - Z^T dY'\right) \\
=& \ \mathrm{trace}\left(\left(\exp(Y') \odot \left(\left(Z \odot \left(\exp(Y')\mathbf{1}_C\mathbf{1}_C^T\right)^{-1}\right)\mathbf{1}_C\mathbf{1}_C^T\right)\right)^T dY' - Z^T dY'\right) \\
=& \ \mathrm{trace}\left(\left(\exp(Y') \odot \left(\exp(Y')\mathbf{1}_C\mathbf{1}_C^T\right)^{-1}\right)^T dY' - Z^T dY'\right) \\
=& \ \mathrm{trace}\left((\mathrm{softmax}(Y') - Z)^T dY'\right)
\end{aligned}
$$

where the 4-th equation holds due to $\left(Z \odot \left(\exp(Y')\mathbf{1}_C\mathbf{1}_C^T\right)^{-1}\right)\mathbf{1}_C\mathbf{1}_C^T = \left(\exp(Y')\mathbf{1}_C\mathbf{1}_C^T\right)^{-1}$. Thus, we have

$$
\frac{d\mathcal{L}}{dY'} = \mathrm{softmax}(Y') - Z = Y - Z
$$

For $Y'$ and $W$, we have

$$
dY' = \hat{A}X\,dW \text{ and } d\mathcal{L} = \mathrm{trace}\left(\frac{d\mathcal{L}}{dY'}^T dY'\right) = \mathrm{trace}\left(\frac{d\mathcal{L}}{dY'}^T \hat{A}X\,dW\right) = \mathrm{trace}\left(\frac{d\mathcal{L}}{dW}^T dW\right)
$$

To get $\frac{d\mathcal{L}}{dW}$ we have,

$$
\frac{d\mathcal{L}}{dW} = X^T\hat{A}^T\frac{d\mathcal{L}}{dY'} = X^T\hat{A}^T(Y - Z) \tag{13}
$$

### E.2 Component-wise Derivation

Denote $\tilde{X} = XW$. We rewrite $\mathcal{L}$ as follows:

$$
\begin{aligned}
\mathcal{L} =& -\mathrm{trace}\left(Z^T \log\left((\exp(Y')\mathbf{1}_C\mathbf{1}_C^T)^{-1} \odot \exp(Y')\right)\right) \\
=& -\mathrm{trace}\left(Z^T\left(-\log(\exp(Y')\mathbf{1}_C\mathbf{1}_C^T) + Y'\right)\right) \\
=& -\mathrm{trace}\left(Z^T Y'\right) + \mathrm{trace}\left(Z^T \log\left(\exp(Y')\mathbf{1}_C\mathbf{1}_C^T\right)\right) \\
=& -\mathrm{trace}\left(Z^T\hat{A}XW\right) + \mathrm{trace}\left(Z^T \log\left(\exp(Y')\mathbf{1}_C\mathbf{1}_C^T\right)\right) \\
=& -\mathrm{trace}\left(Z^T\hat{A}XW\right) + \mathrm{trace}\left(\mathbf{1}_C^T \log\left(\exp(Y')\mathbf{1}_C\right)\right)
\end{aligned}
$$

Expand $\mathcal{L}$ component-wisely, we have

$$
\mathcal{L} = -\sum_{i=1}^{N}\sum_{j\in\mathcal{N}_i} \hat{A}_{i,j} Z_{i,:}\tilde{X}_{j:}^T + \sum_{i=1}^{N}\log\left(\sum_{c=1}^{C}\exp(\sum_{j\in\mathcal{N}_i}\hat{A}_{i,j}\tilde{X}_{j,c})\right)
$$

$$
= -\sum_{i=1}^{N}\log\left(\exp\left(\sum_{c=1}^{C}\sum_{j\in\mathcal{N}_i}\hat{A}_{i,j}Z_{i,c}\tilde{X}_{j,c}\right)\right) + \sum_{i=1}^{N}\log\left(\sum_{c=1}^{C}\exp\left(\sum_{j\in\mathcal{N}_i}\hat{A}_{i,j}\tilde{X}_{j,c}\right)\right)
$$

$$
= -\sum_{i=1}^{N}\log\frac{\exp\left(\sum_{c=1}^{C}\sum_{j\in\mathcal{N}_i}\hat{A}_{i,j}Z_{i,c}\tilde{X}_{j,c}\right)}{\left(\sum_{c=1}^{C}\exp(\sum_{j\in\mathcal{N}_i}\hat{A}_{i,j}\tilde{X}_{j,c})\right)}
$$

Note that $\sum_{c=1}^{C} Z_{j,c} = 1$ for any $j$. Consider the derivation of $\mathcal{L}$ over $\tilde{X}_{j',c'}$:

$$
\frac{d\mathcal{L}}{d\tilde{X}_{j',c'}} = -\sum_{i=1}^{N}\frac{\sum_{c=1}^{C}\exp(\sum_{j\in\mathcal{N}_i}\hat{A}_{i,j}\tilde{X}_{j,c})}{\exp\left(\sum_{c=1}^{C}\sum_{j\in\mathcal{N}_i}\hat{A}_{i,j}Z_{i,c}\tilde{X}_{j,c}\right)}
$$

$$
\times\left(\frac{\left(\hat{A}_{i,j'}Z_{i,c'}\right)\exp\left(\sum_{c=1}^{C}\sum_{j\in\mathcal{N}_i}\hat{A}_{i,j}Z_{i,c}\tilde{X}_{j,c}\right)\left(\sum_{c=1}^{C}\exp(\sum_{j\in\mathcal{N}_i}\hat{A}_{i,j}\tilde{X}_{j,c})\right)}{\left(\sum_{c=1}^{C}\exp(\sum_{j\in\mathcal{N}_i}\hat{A}_{i,j}\tilde{X}_{j,c})\right)^2}\right.
$$

$$
\left. -\frac{\left(\hat{A}_{i,j'}\right)\exp\left(\sum_{c=1}^{C}\sum_{j\in\mathcal{N}_i}\hat{A}_{i,j}Z_{i,c}\tilde{X}_{j,c}\right)\left(\exp(\sum_{j\in\mathcal{N}_i}\hat{A}_{i,j}\tilde{X}_{j,c'})\right)}{\left(\sum_{c=1}^{C}\exp(\sum_{j\in\mathcal{N}_i}\hat{A}_{i,j}\tilde{X}_{j,c})\right)^2}\right)
$$

$$
= -\sum_{i=1}^{N}\left(\frac{\left(\hat{A}_{i,j'}Z_{i,c'}\right)\left(\sum_{c=1}^{C}\exp(\sum_{j\in\mathcal{N}_i}\hat{A}_{i,j}\tilde{X}_{j,c})\right) - \left(\hat{A}_{i,j'}\right)\left(\exp(\sum_{j\in\mathcal{N}_i}\hat{A}_{i,j}\tilde{X}_{j,c'})\right)}{\left(\sum_{c=1}^{C}\exp(\sum_{j\in\mathcal{N}_i}\hat{A}_{i,j}\tilde{X}_{j,c})\right)}\right)
$$

$$
= -\sum_{i=1}^{N}\left(\hat{A}_{i,j'}\frac{\left(\sum_{c=1,c\neq c'}^{C}(Z_{i,c'})\exp(\sum_{j\in\mathcal{N}_i}\hat{A}_{i,j}\tilde{X}_{j,c})\right) + (Z_{i,c'}-1)\left(\exp(\sum_{j\in\mathcal{N}_i}\hat{A}_{i,j}\tilde{X}_{j,c'})\right)}{\left(\sum_{c=1}^{C}\exp(\sum_{j\in\mathcal{N}_i}\hat{A}_{i,j}\tilde{X}_{j,c})\right)}\right)
$$

$$
= -\sum_{i=1}^{N}\hat{A}_{i,j'}\left(Z_{i,c'}\hat{P}(Y_i\neq c') + (Z_{i,c'}-1)\hat{P}(Y_i = c')\right)
$$

$$
= -\sum_{i=1}^{N}\hat{A}_{i,j'}\left(Z_{i,c'} - \hat{P}(Y_i = c')\right)
$$

(14)

Writing the above in matrix form, we have

$$
\frac{d\mathcal{L}}{d\tilde{X}} = \hat{A}(Z-Y),\ \frac{d\mathcal{L}}{d\tilde{W}} = X^T\hat{A}^T(Z-Y),\ \Delta Y' \propto \hat{A}XX^T\hat{A}^T(Z-Y) \tag{15}
$$

# F Proof of Proposition 1

*Proof.* According to the given assumptions, for node $v$, we have $\hat{A}_{v,k} = \frac{1}{d+1}$, the expected number of intra-class edges is $dh$ (here the self-loop edge introduced by $\hat{A}$ is not counted based on the definition of edge homophily and data generation process) and inter-class edges is $(1-h)d$. Suppose there are $C \geq 2$ classes. Consider matrix $\hat{A}Z$,

Then, we have $\mathbb{E}\left[(\hat{A}Z)_{v,c}\right] = \mathbb{E}\left[\sum_{k \in \mathcal{V}} \hat{A}_{v,k} \mathbf{1}_{\{Z_{k,:}=e_c^T\}}\right] = \sum_{k \in \mathcal{V}} \frac{\mathbb{E}\left[\mathbf{1}_{\{Z_{k,:}=e_c^T\}}\right]}{d+1}$, where $\mathbf{1}$ is the indicator function.

When $v$ is in class $c$, we have $\sum_{k \in \mathcal{V}} \frac{\mathbb{E}\left[\mathbf{1}_{\{Z_{k,:}=e_c^T\}}\right]}{d+1} = \frac{hd+1}{d+1}$ ($hd+1 = hd$ intra-class edges $+ 1$ self-loop introduced by $\hat{A}$).

When $v$ is not in class $c$, we have $\sum_{k \in \mathcal{V}} \frac{\mathbb{E}\left[\mathbf{1}_{\{Z_{k,:}=e_c^T\}}\right]}{d+1} = \frac{(1-h)d}{(C-1)(d+1)}$ ($(1-h)d$ inter-class edges uniformly distributed in the other $C-1$ classes).

For nodes $v, u$, we have $(\hat{A}Z)_{v,:}, (\hat{A}Z)_{u,:} \in \mathbb{R}^C$ and since elements in $\hat{A}_{v,k}$ and $\hat{A}_{u,k'}$ are independently generated for all $k, k' \in \mathcal{V}$, we have

$$\mathbb{E}\left[(\hat{A}Z)_{v,c}(\hat{A}Z)_{u,c}\right] = \mathbb{E}\left[(\sum_{k \in \mathcal{V}} \hat{A}_{v,k} \mathbf{1}_{\{Z_{k,:}=e_c^T\}})(\sum_{k' \in \mathcal{V}} \hat{A}_{u,k'} \mathbf{1}_{\{Z_{k',:}=e_c^T\}})\right]$$

$$= \mathbb{E}\left[(\sum_{k \in \mathcal{V}} \hat{A}_{v,k} \mathbf{1}_{\{Z_{k,:}=e_c^T\}})\right] \mathbb{E}\left[(\sum_{k' \in \mathcal{V}} \hat{A}_{u,k'} \mathbf{1}_{\{Z_{k',:}=e_c^T\}})\right]$$

Thus,

$$\mathbb{E}\left[S(\hat{A}, Z)_{v,u}\right] = \mathbb{E}\left[< (\hat{A}Z)_{v,:}, (\hat{A}Z)_{u,:} >\right] = \sum_c \mathbb{E}\left[(\sum_{k \in \mathcal{V}} \hat{A}_{v,k} \mathbf{1}_{\{Z_{k,:}=e_c^T\}})\right] \mathbb{E}\left[(\sum_{k' \in \mathcal{V}} \hat{A}_{u,k'} \mathbf{1}_{\{Z_{k',:}=e_c^T\}})\right]$$

$$= \begin{cases} \left(\frac{hd+1}{d+1}\right)^2 + \frac{((1-h)d)^2}{(C-1)(d+1)^2}, & u, v \text{ are in the same class} \\ \frac{2(hd+1)(1-h)d}{(C-1)(d+1)^2} + \frac{(C-2)(1-h)^2d^2}{(C-1)^2(d+1)^2}, & u, v \text{ are in different classes} \end{cases}$$

For nodes $u_1, u_2$, and $v$, where $Z_{u_1,:} = Z_{v,:}$ and $Z_{u_2,:} \neq Z_{v,:}$,

$$g(h) \equiv \mathbb{E}\left[S(\hat{A}, Z)_{v,u_1}\right] - \mathbb{E}\left[S(\hat{A}, Z)_{v,u_2}\right] \tag{16}$$

$$= \frac{(C-1)^2(hd+1)^2 + (C-1)\left[(1-h)d\right]^2 - (C-1)\left(2(hd+1)(1-h)d\right) - (C-2)\left[(1-h)d\right]^2}{(C-1)^2(d+1)^2}$$

$$= \left(\frac{(C-1)(hd+1) - (1-h)d}{(C-1)(d+1)}\right)^2$$

Setting $g(h) = 0$, we obtain the optimal $h$:

$$h = \frac{d+1-C}{Cd} \tag{17}$$

For the data generation process in the synthetic experiments, we fix $d_{\text{intra}}$, then $d = d_{\text{intra}}/h$, which is a function of $h$. We change $d$ in equation 17 to $d_{\text{intra}}/h$, leading to

$$h = \frac{d_{\text{intra}}/h + 1 - C}{Cd_{\text{intra}}/h} \tag{18}$$

It is easy to observe that $h$ satisfying equation 18 still makes $g(h) = 0$, when $d$ in $g(h)$ is replaced by $d_{\text{intra}}/h$. From equation 18 we obtain the optimal $h$ in terms of $d_{\text{intra}}$:

$$h = \frac{d_{\text{intra}}}{Cd_{\text{intra}} + C - 1}$$

$\square$

### F.1 An Extension of Proposition 1

Base on the definition of aggregation similarity, we have

$$S_{\text{agg}}\left(S(\hat{A}, Z)\right) = \frac{\left|\left\{v \mid \text{Mean}_u\left(\{S(\hat{A}, Z)_{v,u}|Z_{u,:} = Z_{v,:}\}\right) \geq \text{Mean}_u\left(\{S(\hat{A}, Z)_{v,u}|Z_{u,:} \neq Z_{v,:}\}\right)\right\}\right|}{|\mathcal{V}|}$$

$$= \frac{\sum_{v \in \mathcal{V}} \mathbf{1}_{\left\{\text{Mean}_u\left(\{S(\hat{A},Z)_{v,u}|Z_{u,:}=Z_{v,:}\}\right) \geq \text{Mean}_u\left(\{S(\hat{A},Z)_{v,u}|Z_{u,:}\neq Z_{v,:}\}\right)\right\}}}{|\mathcal{V}|}$$

Then,

$$\mathbb{E}\left(S_{\text{agg}}\left(S(\hat{A}, Z)\right)\right) = \mathbb{E}\left(\frac{\sum_{v \in \mathcal{V}} \mathbf{1}_{\left\{\text{Mean}_u\left(\{S(\hat{A},Z)_{v,u}|Z_{u,:}=Z_{v,:}\}\right) \geq \text{Mean}_u\left(\{S(\hat{A},Z)_{v,u}|Z_{u,:}\neq Z_{v,:}\}\right)\right\}}}{|\mathcal{V}|}\right)$$

$$= \frac{\sum_{v \in \mathcal{V}} \mathbb{P}\left(\text{Mean}_u\left(\{S(\hat{A}, Z)_{v,u}|Z_{u,:} = Z_{v,:}\}\right) \geq \text{Mean}_u\left(\{S(\hat{A}, Z)_{v,u}|Z_{u,:} \neq Z_{v,:}\}\right)\right)}{|\mathcal{V}|}$$

$$= \mathbb{P}\left(\text{Mean}_u\left(\{S(\hat{A}, Z)_{v,u}|Z_{u,:} = Z_{v,:}\}\right) - \text{Mean}_u\left(\{S(\hat{A}, Z)_{v,u}|Z_{u,:} \neq Z_{v,:}\}\right) \geq 0\right)$$

Consider the random variable

$$RV = \text{Mean}_u\left(\{S(\hat{A}, Z)_{v,u}|Z_{u,:} = Z_{v,:}\}\right) - \text{Mean}_u\left(\{S(\hat{A}, Z)_{v,u}|Z_{u,:} \neq Z_{v,:}\}\right)$$

Since $RV$ is symmetrically distributed and under the conditions in proposition 1, its expectation is $\mathbb{E}[RV] = g(h)$ as showed in equation 16. Since the minimum of $g(h)$ is 0 and $RV$ is symmetrically distributed, we have $\mathbb{P}(RV \geq 0) \geq 0.5$ and this can explain why $H_{\text{agg}}(\mathcal{G})$ is always greater than 0.5 in many real-world tasks.

## G  Proof of Theorem 1

*Proof.* Define $W_v^c = (\hat{A}Z)_{v,c}$. Then,

$$W_v^c = \sum_{k \in \mathcal{V}} \hat{A}_{v,k} \mathbf{1}_{\{Z_{k,:} = e_c^T\}} \in [0, 1], \quad \sum_{c=1}^{C} W_v^c = 1$$

Note that

$$S(I - \hat{A}, Z) = (I - \hat{A})ZZ^T(I - \hat{A})^T = ZZ^T + \hat{A}ZZ^T\hat{A}^T - \hat{A}ZZ^T - ZZ^T\hat{A}^T \quad (19)$$

For any node $v$, let the class $v$ belongs to be denoted by $c_v$. For two nodes $v, u$, if $Z_{v,:} \neq Z_{u,:}$, we have

$$(ZZ^T)_{v,u} = 0$$

$$(\hat{A}ZZ^T\hat{A}^T)_{v,u} = \sum_{c=1}^{C} W_v^c W_u^c$$

$$(\hat{A}ZZ^T)_{v,u} = W_v^{c_u}$$

$$(ZZ^T\hat{A}^T)_{v,u} = (\hat{A}ZZ^T)_{u,v} = W_u^{c_v}$$

Then, from equation 19 it follows that

$$(S(I - \hat{A}, Z))_{v,u} = \sum_{c=1}^{C} W_v^c W_u^c - W_v^{c_u} - W_u^{c_v}$$

When $C = 2$,

$$S(I - \hat{A}, Z)_{v,u} = W_v^{c_u}(W_u^{c_u} - 1) + W_u^{c_v}(W_v^{c_v} - 1) \leq 0$$

If $Z_{v,:} = Z_{u,:}$, *i.e.*, $c_v = c_u$, we have

$$(ZZ^T)_{v,u} = 1$$

$$(\hat{A}ZZ^T\hat{A}^T)_{v,u} = \sum_{c=1}^{C} W_v^c W_u^c$$

$$(\hat{A}ZZ^T)_{v,u} = W_v^{c_v}$$

$$(ZZ^T\hat{A}^T)_{v,u} = (\hat{A}ZZ^T)_{u,v} = W_u^{c_u} = W_u^{c_v}$$

Then, from equation 19 it follows that

$$S(I - \hat{A}, Z)_{v,u} = 1 + \sum_{c=1}^{C} W_v^c W_u^c - W_v^{c_v} - W_u^{c_v}$$

$$= \sum_{c=1,c\neq c_v}^{C} W_v^c W_u^c + 1 + W_v^{c_v} W_u^{c_v} - W_v^{c_v} - W_u^{c_v}$$

$$= \sum_{c=1,c\neq c_v}^{C} W_v^c W_u^c + (1 - W_v^{c_v})(1 - W_u^{c_v}) \geq 0$$

Thus, if $C = 2$, for any $v \in \mathcal{V}$, if $Z_{u,:} \neq Z_{v,:}$, we have $S(I - \hat{A}, Z)_{v,u} \leq 0$; if $Z_{u,:} = Z_{v,:}$, we have $S(I - \hat{A}, Z)_{v,u} \geq 0$. Apparently, the two conditions in equation 10 are satisfied. Thus $v$ is diversification distinguishable and $\text{DD}_{\hat{A},X}(\mathcal{G}) = 1$. The theorem is proved. $\square$

## H  Discussion of the Limitations of Diversification Operation

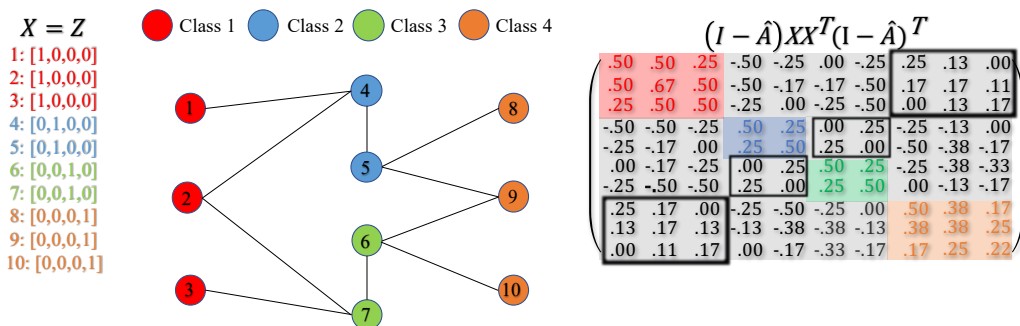

Figure 13: Example of the case (the area in black box) that HP filter does not work well for harmful heterophily

From the black box area of $S(I - \hat{A}, X)$ in the example in Figure 13 we can see that nodes in class 1 and 4 assign non-negative weights to each other although there is no edge between them; nodes in class 2 and 3 assign non-negative weights to each other as well. This is because the surrounding differences of class 1 are similar as class 4, so are class 2 and 3. In real-world applications, when nodes in several small clusters connect to a large cluster, the surrounding differences of the nodes in the small clusters will become similar. In such case, HP filter are not able to distinguish the nodes from different small clusters.

## I  The Similarity, Homophily and $\text{DD}_{\hat{A},X}(\mathcal{G})$ Metrics and Their Estimations

Firstly, we would like to clarify that, for each curve in the synthetic experiments, the node features are fixed and we only change the homophily values. But in real-world tasks, different datasets have different features and aggregated features. Thus, to get more instructive information for different datasets and compare them, we need to consider more metrics, e.g. feature-label consistency and

aggregated-feature-label consistency. With the similarity score of the features $S_{\text{agg}}\left(S(I,X)\right)$ and aggregated features $S_{\text{agg}}\left(S(\hat{A},X)\right)$ listed in Table 15, our methods open up a new perspective on analyzing and comparing the performance of graph-agnostic models and graph-aware models in real-world tasks. Here are 2 examples.

**Example 1:** People observe that GCN (graph-aware model) underperforms MLP-2 (graph-agnostic model) on *Cornell, Wisconsin, Texas, Film* and people commonly believe that the bad graph structure (low $H_{\text{edge}}, H_{\text{node}}, H_{\text{class}}$ values) is the reason for performance degradation. But based on the high aggregation homophily values, the graph structure inconsistency is not the main cause of the performance degradation. And from Table 15 we can see that the $S_{\text{agg}}\left(S(\hat{A},X)\right)$ for those 4 datasets are lower than their corresponding $S_{\text{agg}}\left(S(I,X)\right)$, which implies that it is the aggregated-feature-label inconsistency that causes the performance degradation, i.e. the aggregation step actually decrease the quality of node features rather than making them more distinguishable.

For the rest 5 datasets *Chameleon, Squirrel, Cora, Citeseer, PubMed*, we all have $S_{\text{agg}}\left(S(\hat{A},X)\right)$ larger than $S_{\text{agg}}\left(S(I,X)\right)$ except *PubMed*, which means the aggregated features have higher quality than raw features. We can see that the proposed metrics are much more instructive than the existing ones.

**Example 2:** According to $H_{\text{edge}}, H_{\text{node}}, H_{\text{class}}$, the value for *Chameleon*, and *Squirrel* are extremely low indicating graph structure are bad for GNNs. But on contrary, GCN outperforms MLP-2 on those 2 datasets. Traditional homophily metrics fail to explain such phenomenon but our method can give an explanation from different angles: For Chameleon, its modified aggregation homophily is not low and its $S_{\text{agg}}\left(S(\hat{A},X)\right)$ is higher than its $S_{\text{agg}}\left(S(I,X)\right)$, which means its graph-label consistency together with aggregated-feature-label consistency help the graph-aware model obtain the performance gain; for Squirrel, its modified aggregation homophily is low but its $S_{\text{agg}}\left(S(\hat{A},X)\right)$ is higher than its $S_{\text{agg}}\left(S(I,X)\right)$, which means although its graph-label consistency is bad, the aggregated-feature-label consistency is the key factor to help the graph-aware model perform better.

We also need to point out that (modified) aggregation similarity score, $S_{\text{agg}}\left(S(\hat{A},X)\right)$ and $S_{\text{agg}}\left(S(I,X)\right)$ are not deciding values because they do not consider the nonlinear structure in the features. In practice, a low score does not tell us the GNN models will definitely perform bad.

|  | Cornell | Wisconsin | Texas | Film | Chameleon | Squirrel | Cora | CiteSeer | PubMed |
|---|---|---|---|---|---|---|---|---|---|
| $H_{\text{agg}}(\mathcal{G})$ | 0.9016 | 0.8884 | 0.847 | 0.8411 | 0.805 | 0.6783 | 0.9952 | 0.9913 | 0.9716 |
| $S_{\text{agg}}\left(S(\hat{A},X)\right)$ | 0.8251 | 0.7769 | 0.6557 | 0.5118 | 0.8292 | 0.7216 | 0.9439 | 0.9393 | 0.8623 |
| $S_{\text{agg}}\left(S(I,X)\right)$ | 0.9672 | 0.8287 | 0.9672 | 0.5405 | 0.7931 | 0.701 | 0.9103 | 0.9315 | 0.8823 |
| $DD_{\hat{A},X}(\mathcal{G})$ | 0.3497 | 0.6096 | 0.459 | 0.3279 | 0.3109 | 0.2711 | 0.2681 | 0.4124 | 0.1889 |
| $\hat{H}_{\text{agg}}(\mathcal{G})$ | 0.9046 ± 0.0282 | 0.9147 ± 0.0260 | 0.8596 ± 0.0299 | 0.8451 ± 0.0041 | 0.8041 ± 0.0078 | 0.6788 ± 0.0077 | 0.9959 ± 0.0011 | 0.9907 ± 0.0015 | 0.9724 ± 0.0015 |
| $\hat{S}_{\text{agg}}\left(S(\hat{A},X)\right)$ | 0.8266 ± 0.0526 | 0.8280 ± 0.0351 | 0.6835 ± 0.0498 | 0.5345 ± 0.0421 | 0.8433 ± 0.0070 | 0.7352 ± 0.0132 | 0.9487 ± 0.0023 | 0.9451 ± 0.0038 | 0.8626 ± 0.0021 |
| $\hat{S}_{\text{agg}}\left(S(I,X)\right)$ | 0.9752 ± 0.0174 | 0.8680 ± 0.0270 | 0.9661 ± 0.0336 | 0.5438 ± 0.0184 | 0.8257 ± 0.0050 | 0.7472 ± 0.0089 | 0.9204 ± 0.0044 | 0.9441 ± 0.0036 | 0.8835 ± 0.0019 |
| $DD_{\hat{A},X}(\mathcal{G})$ | 0.3936 ± 0.0663 | 0.6073 ± 0.0436 | 0.4817 ± 0.0762 | 0.3300 ± 0.0136 | 0.3329 ± 0.0151 | 0.3021 ± 0.0101 | 0.3198 ± 0.0225 | 0.4424 ± 0.0136 | 0.1919 ± 0.0046 |

Table 15: Additional metrics and their estimations with only training labels (mean ± std)

Furthermore, in most real-world applications, not all labels are available to calculate the dataset statistics. Thus, we randomly split the data into 60%/20%/20% for training/validation/test, and only use the training labels for the estimation of the statistics. We repeat each estimation for 10 times and report the mean with standard deviation. The results are shown in table 15.

**Analysis** From the reported results we can see that the estimations are accurate and the errors are within the acceptable range, which means the proposed metrics and similarity scores can be accurately estimated with a subset of labels and this is important for real-world applications.

## J A Detailed Explanation of the Differences Between ACM(II)-GNNs and GPRGNN, FAGCN

Differences with GPRGNN [8]:

- *GPRGNN does not feed distinct **node-wise feature transformation** to different "multi-scale channels"*

  We first rewrite GPRGNN as

  $$\mathbf{Z} = \sum_{k=0}^{K} \gamma_k \mathbf{H}^{(k)} = \sum_{k=0}^{K} \gamma_k I \mathbf{H}^{(k)} = \sum_{k=0}^{K} diag(\gamma_k, \gamma_k, \ldots, \gamma_k) \mathbf{H}^{(k)}, \text{ where } \mathbf{H}^{(k)} = \hat{A}_{\text{sym}} \mathbf{H}^{(k-1)}, \mathbf{H}_{i:}^{(0)} = f_\theta(X_{i:}).$$

  From the above equation we can see that $\mathbf{Z} = \sum_{k=0}^{K} \gamma_k \hat{A}_{\text{sym}}^k f_\theta(X_{i:})$, *i.e.,* the **node-wise feature transformation** in GPRGNN is only learned by the same $\theta$ for all the "multi-scale channels". But in the ACM framework, different channels extract distinct information with different parameters separately.

- *GPRGNN does not have node-wise mixing mechanism.*

  There is no node-wise mixing in GPRGNN. The mixing mechanism in GPRGNN is $\mathbf{Z} = \sum_{k=0}^{K} diag(\gamma_k, \gamma_k, \ldots, \gamma_k) \mathbf{H}^{(k)}$, i.e. for each "multi-scale channel $k$", all nodes share the same mixing parameter $\gamma_k$. But in the ACM framework, the node-wise channel mixing can be written as $\mathbf{Z} = \sum_{k=0}^{K} diag(\gamma_k^1, \gamma_k^2, \ldots, \gamma_k^N) \mathbf{H}^{(k)}$ where $K$ is the number of channels, $N$ is the number of nodes and $\gamma_k^i, i = 1, \ldots, N$ are the mixing weights that are learned by node $i$ to mix channel $k$. ACM and ACMII allow GNNs to learn more diverse mixing parameters in diagonal than GPRGNN and thus, have stronger expressive power than GPRGNN.

Differences with FAGCN [4]:

- *The targets of node-wise operations in ACM (channel mixing) and FAGCN (negative message passing) are different.*

  Instead of using a fixed low-pass filter $\hat{A}$, FAGCN tries to learn a more powerful aggregator $\hat{A}'$ based on $\hat{A}$ by allowing negative message passing. The node-wise operation in FAGCN is similar to GAT [3] which is trying to modify the **node-wise filtering (message passing) process**, i.e. for each node $i$, it assigns different weights $\alpha_{ij} \in [-1, 1]$ to different neighborhood nodes (equation 7 in FAGCN paper). The goal of this node-wise operation in FAGCN is **to learn a new filter during the filtering process node-wisely**. But in ACM, the node-wise operation is to mix the **filtered information** from each channel which is processed by different fixed filters. The targets of two the node-wise operations are actually different things.

- *FAGCN does not learn distinct information from different "channels". FAGCN only uses simple addition to mix information instead of node-wise channel mixing mechanism*

  The learned filter $\hat{A}'$ can be decomposed as $\hat{A}' = \hat{A}_1' + (-\hat{A}_2')$, where $\hat{A}_1'$ and $-\hat{A}_2'$ represent positive and negative edge (propagation) information, respectively. But FAGCN does not feed distinct information to $\hat{A}_1'$ and $-\hat{A}_2'$. Moreover, the aggregated $\hat{A}_1' X$ and "diversified" information $(-\hat{A}_2') X$ are simply added together instead of using any node-wise mixing mechanism. In ACM, we learn distinct information separately in each channel with different parameters and add them adaptively and node-wisely instead of just adding them together. In section 6.1, the ablation study has empirically shown that node-wise adaptive channel mixing is better than simple addition.

## K On Expressive Power and Frequency Analysis

GPRGNN, BernNet and many other GNN models share the common belief that "expressiveness of GNN == expressiveness of graph filters" and this motivates them to design graph filters with strong

expressive power. This might be true in many cases but not always because we have other parts besides the filters in GNN that can be improved. Thus at the first glance, the 3-channel architecture might look ordinary from the perspective of graph filter. But when combined with the node-wise channel mixing mechanism, this simple architecture becomes powerful. One simple guess is that it can fit more complex node label pattern (distribution).

We did not compare the spectral filters against GPRGNN, because our analysis and derivation of ACM are from node level instead of spectral domain. Also, ACM-GCN is not learning a spectral filter and each channel has different input information rather than the same information.

In addition, we would like to share an opinion that are against the mainstream opinion about the spectral analysis of the filters in GNNs: The spectral analysis is based on graph Laplacian, whose smoothness is defined on the given graph structure. Therefore, when the given graph structure is "bad" or "trivial", the smoothness becomes trivial as well. The traditional spectral analysis is valid for "good" graph structures, e.g. Cora, Citeseer, PubMed, but for "bad" graphs, we might need some other analysis. This is not conclusive and we just share it here for you interest.