# OpenReview forum: "Revisiting Heterophily For Graph Neural Networks"
_NeurIPS.cc/2022/Conference — NeurIPS 2022 Accept_

### Official Review · Reviewer_S2KT · 2022-07-09

**Rating:** 7
**Confidence:** 5
**Ethics Flag:** Yes
**Soundness:** 4 excellent
**Presentation:** 2 fair
**Contribution:** 3 good

**Summary:**

This paper considers heterophily in GNN for node classification. It has two major contributions to the community. First, it pointed out that mainstream homophily measures (i.e. edge, node, class) does not align with GCN/SGC classification accuracy, since those measures does not distinguish harmless/harmful heterophilty. It proposes a new mesaure that inspired by SGC gradient updates, which shows good alignment between homophily and classification accuracy on synthetic datasets. Second, this paper propose ACM which uses low pass high pass and identity channels, combine together with softmax operation, ACM is based on the intuition that high pass filter would help distinguish nodes with harmful heterophily. ACM is empirically verified to be good.

**Questions:**

N/A

**Limitations:**

Suggestions
  - in abstract, make clear this paper is about node classification, (real-world tasks -> nodeclassification)
  - line 137, i assume you meant to write [-1, 1]
  - line 142, I am not sure what you mean here
  - line 554, i assume you meant to write 1_{N}^{T}
  - my experience is that graphsage typically works well for heterophily graphs, adding that as a baseline to acm would be useful
  - add citations
    - (gnn heterophily) Residual correlation in graph neural network regression
    - (gnn heterophily) Beyond Homophily in Graph Neural Networks
    - (gnn heterophily) New benchmarks for learning on non-homophilous graphs


**Strengths And Weaknesses:**

Strengths
  - the new homophily is novel and practical connection with SGC gradient is very intuitive
    - this is a very notable contribution to the community, although homophily and SGC performance has been discussed in the past by many, it is a known issue that the edge homophily is not fully correlated with SGC performance. the analysis and new hopophily measure solved this problem
  - ACM is novel effective and intuitive

Weakness
  - writing could improve

---

> ### Author Response · Authors · 2022-08-02
> **Author Response to Reviewer S2KT**
>
> Thanks for you valuable comments. We will add the suggested references and keep improving the writing.
>
> Authors

---

### Official Review · Reviewer_vQE7 · 2022-07-11

**Rating:** 6
**Confidence:** 3
**Soundness:** 3 good
**Presentation:** 2 fair
**Contribution:** 3 good

**Summary:**

This paper first addresses the limitation of the previously proposed metrics in analyzing the performance of GNNs on heterophilic graph datasets. To solve the limitation, a new metric based on the post-aggregation node similarity is proposed. The newly proposed metric better reflects the performances of GNNs on node classification tasks. To further improve the GNNs, the authors propose an adaptive channel mixing mechanism that uses both high and low-frequency graph signals. The adaptive channel mixing mechanism is employed in existing GNN models and shows improved performances on various node classification datasets.

**Questions:**

- Two variants of the adaptive channel mixing mechanism are proposed. From a practical point of view, which model should be used?
- The new metric can be extended to the multi-hop aggregation setting. Was there any further finding when the authors consider the multi-hop aggregation matrix?
- The intuition behind the mixing matrix is unclear. Could authors elaborate more on the necessity of the mixing matrix W_Mix? Why the attention is insufficient to mix the channel outputs?

**Limitations:**

Although the limitation of the proposed approach is addressed in the appendix, the limitations are shown based on a curated example and fail to show general conditions where the model fails.


**Strengths And Weaknesses:**

Strengths

- The paper addresses the limitation of previously suggested metrics well and proposes a new metric that can better reflect the behaviors of GNN models.
- The adaptive channel mixing mechanism is intuitive and can be applied to many existing GNN models without adding too much computational complexity.
- Extensive experimental results are shown on various datasets to show the performance of the proposed framework.

Weaknesses

- Although the metric shows a strong correlation with many existing GNN models, the analysis of post-aggregation is based on SGC and cannot be generalized to the other models directly.
- Representation of some parts can further be improved. For example, the connection between the diversification distinguishability and the adaptive channel mixing framework seems vague.  The definition of diversification distinguishability and the following theorem seem not necessary to introduce the necessity of the proposed framework. Also, the connection between the metric part and the model part seems not very clear although both parts are inspired by the post-aggregation similarity.
- The proposed metric seems to be correlated with model performances. However, the correlation between the performances of real-world datasets and the proposed metric hasn’t been shown.

---

> ### Author Response · Authors · 2022-08-02
> **Author Response to Reviewer vQE7 (Part 1)**
>
> ### Q1.
> Although the metric shows a strong correlation with many existing GNN models, the analysis of post-aggregation is based on SGC and cannot be generalized to the other models directly.
>
> ### R1.
>
> Just like what we mentioned in the limitation part (section 7), all the existing homophily metrics and our metrics only consider the linear feature-independent relation between graph structure and labels. Although the proposed post-aggregation similarity principle shows advantages over homophily principle, people can consider designing a non-linear feature-dependent metrics in the future which can be generalized to other models with non-linear activation functions. We believe our paper can be a good starting point for this future research.
>
>
> ### Q2.
> Representation of some parts can further be improved. For example, the connection between the diversification distinguishability and the adaptive channel mixing framework seems vague. The definition of diversification distinguishability and the following theorem seem not necessary to introduce the necessity of the proposed framework. Also, the connection between the metric part and the model part seems not very clear although both parts are inspired by the post-aggregation similarity.
>
> ### R2.
>
> #### (1) "the connection between the diversification distinguishability and the adaptive channel mixing framework seems vague"
>
> Diversification distinguishability is proposed to show the effectiveness of high-pass filter, which leads us to the 3-channel GNN architecture. The node-wise adaptive channel mixing mechanism is based on the observation from the example in Figure 3, not diversification distinguishability. Diversification distinguishability and adaptive channel mixing are two different contributions in the ACM framework.
>
> #### (2) "The definition of diversification distinguishability and the following theorem seem not necessary to introduce the necessity of the proposed framework"
>
> As mentioned in our paper, Theorem 1, which is based on the definition of diversification distinguishability, theoretically shows the effectiveness of high-pass filter on addressing the heterophily problem, which leads us to the 3-channel GNN framework.
>
> #### (3) "the connection between the metric part and the model part seems not very clear"
>
> As you mentioned in your question, the key part to connect the investigation section (heterophily; new metrics) and the methodology section (new model) is the post-aggregation node similarity matrix. The metrics are the by-products of the similarity matrix.
>
> Like the existing homophily metrics, the purpose of designing aggregation homophily is just to measure whether the aggregation (message passing) step will help $\textbf{uni-channel}$ graph-aware model outperform graph-agnostic model. Our proposed 3-channel architecture is beyond the uni-channel framework, and thus its performance cannot be directly measured by the proposed metric. But based on the post-aggregation node similarity matrix, we can show and prove the effectiveness of the high-pass filter on addressing heterophily, which is one of the main reasons that we design the 3-channel architecture.

---

> > ### Author Response · Authors · 2022-08-02
> > **Author Response to Reviewer vQE7 (Part 2)**
> >
> > ### Q3.
> > The proposed metric seems to be correlated with model performances. However, the correlation between the performances of real-world datasets and the proposed metric hasn’t been shown.
> >
> > ### R3.
> >
> > The results of proposed metrics on real-world datasets are reported in Table 8 in Appendix H and we also provide an explanation to the results. We elaborate the explanation in the following paragraphs for you.
> >
> > There are three key factors that influence the performance of GNNs in real-world tasks: labels, features and graph structure. The (modified) aggregation homophily tries to investigate the consistency of graph structure and labels from post-aggregation node similarity perspective with given features.The advantage is verified through the synthetic experiments.
> >
> > In real-world datasets, besides graph-label consistency, we need to consider feature-label consistency and aggregated-feature-label consistency as well to fully investigate the performance of NNs and GNNs. With aggregation similarity score of the features $S_\text{agg}\left(S(I,X)\right)$ and aggregated features $S_\text{agg}\left(S(\hat{A},X)\right)$ listed in Table 8, our methods open up a new way on analyzing and comparing the performance of graph-agnostic models and graph-aware models in real-world tasks. Here are two concrete explanations to the results from Table 8.
> >
> > Explanation 1: It is observed that GCN (graph-aware model) underperforms MLP-2 (graph-agnostic model) on $\textit{Cornell, Wisconsin, Texas, Film}$ and people commonly thinks that the bad graph structure is the reason for performance degradation. But based on the proposed aggregation homophily, the graph-label inconsistency is not the main cause of it. Furthermore, from Table 8 we can see that the $S_\text{agg}\left(S(\hat{A},X)\right)$ for the above 4 datasets are lower than their corresponding $S_\text{agg}\left(S(I,X)\right)$, which implies that it is the aggregated-feature-label inconsistency that causes the performance degradation, i.e. the aggregation step actually decrease the quality of node features rather than making them more distinguishable.
> >
> > Explanation 2: For the rest 5 datasets $\textit{Chameleon, Squirrel, Cora, Citeseer, PubMed}$, we all have $S_\text{agg}\left(S(\hat{A},X)\right)$ larger than $S_\text{agg}\left(S(I,X)\right)$ except $\textit{PubMed}$. We can see that the proposed metrics are much more instructive than the existing ones.
> >
> > We also need to point out that (modified) aggregation similarity score, $S_\text{agg}\left(S(\hat{A},X)\right)$ and $S_\text{agg}\left(S(I,X)\right)$ are not deciding values because they only capture linear relations and a low score does not mean the GNN models will definitely perform worse than NNs. In practice, we also need to consider the non-linear relation among labels, features and graph structure, which is lacking in the existing metrics and our metrics (this can explain the failure of our metrics on $\textit{PubMed}$). But our proposed metrics can be a good starting point for future research.
> >
> >
> > ### Q4.
> > Two variants of the adaptive channel mixing mechanism are proposed. From a practical point of view, which model should be used?
> >
> > ### R4.
> > We design those two different options to allow our model to have the flexibility to extract linear or non-linear information from features before feeding them into each channel. It depends on the nonlinearity structure in the features and the relation between feature and graph. Although in most applications, we did not find big differences between those two options, we encourage the users to try both in their own tasks. We do not have a concrete instruction for now.
> >
> >
> > ### Q5.
> > The new metric can be extended to the multi-hop aggregation setting. Was there any further finding when the authors consider the multi-hop aggregation matrix?
> >
> > ### R5.
> >
> > We haven't got interesting findings on multi-hop aggregation setting for now.
> >
> > Although it is easy to extend our metric to higher order neighborhood, it is found that higher order homophily is not just a simple extension of 1-order homophily [1]. So we just keep the discussion within 1-hop neighborhood in this paper. In the future, we might need to find out a new principle beyond post-aggregation node similarity for multi-hop aggregation.

---

> > > ### Author Response · Authors · 2022-08-02
> > > **Author Response to Reviewer vQE7 (Part 3)**
> > >
> > > ### Q6.
> > > The intuition behind the mixing matrix is unclear. Could authors elaborate more on the necessity of the mixing matrix W_Mix? Why the attention is insufficient to mix the channel outputs?
> > >
> > > ### R6.
> > >
> > > We want to grant our model the flexibility to learn more diverse weight values for each channel. We compare ACM with and without $W_\text{mix}$ and here are the results.
> > >
> > > Models       |          With $W_\text{mix}$        ||  Without $W_\text{mix}$     ||
> > >  ------------ | :-----------: | :-----------: | :-----------: | :-----------: |
> > > Datasets       |   ACM    |  ACMII |       ACM |      ACMII |
> > >  Cornell | 94.75 $\pm$ 3.8 | **95.9 $\pm$ 1.83** | 93.61 $\pm$ 2.37 | 90.49 $\pm$ 2.72
> > > Wisconsin | 95.75 $\pm$ 2.03 | 96.62 $\pm$ 2.44 | 95 $\pm$ 2.5	 | **97.50 $\pm$ 1.25**
> > > Texas | 94.92 $\pm$ 2.88 | **95.08 $\pm$ 2.07** | 94.92 $\pm$ 2.79 | 94.92 $\pm$ 2.79
> > > Film  | 41.62 $\pm$ 1.15 | **41.84 $\pm$ 1.15** | 40.79 $\pm$ 1.01	| 40.86 $\pm$ 1.48
> > > Chameleon | **69.04 $\pm$ 1.74** | 68.38 $\pm$ 1.36 | 68.16 $\pm$ 1.79	 | 66.78 $\pm$ 2.79
> > > Squirrel | **58.02 $\pm$ 1.86** | 54.53 $\pm$ 2.09 | 55.35 $\pm$ 1.72 | 52.98 $\pm$ 1.66
> > > Cora  | 88.62 $\pm$ 1.22 | **89.00 $\pm$ 0.72** | 88.41 $\pm$ 1.63 | 88.72 $\pm$ 1.5
> > > Citeseer | 81.68 $\pm$ 0.97 | **81.79 $\pm$ 0.95** | 81.65 $\pm$ 1.48 | 81.72 $\pm$ 1.58
> > > PubMed | 90.66 $\pm$ 0.47 | **90.74 $\pm$ 0.5** | 90.46 $\pm$ 0.69	 | 90.39 $\pm$ 1.33
> > >
> > > We can see that ACM with $W_\text{mix}$ shows superiority in most datasets, although it is not statistically significant on some of them.
> > >
> > > One possible explanation of the advantage is that $W_\text{mix}$ could help alleviate the dominance and bias to majority: Suppose in a dataset, most of the nodes need more information from LP channel than HP and identity channels, then $W_L, W_H, W_I$ tend to learn larger $\alpha_L$ than $\alpha_H$ and $\alpha_I$. For the minority nodes that need more information from HP or identity channels, they are hard to get large $\alpha_H$ or $\alpha_I$ values because $W_L, W_H, W_I$ are biased to the majority. And $W_\text{mix}$ can help us to learn more diverse alpha values when $W_L, W_H, W_I$ are biased.
> > >
> > > Attention with more complicated design can be found for the node-wise adaptive channel mixing mechanism, but we do not explore this direction deeper in this paper because investigating attention function is not the main contribution of our paper.
> > >
> > > [1] Evtushenko, Anna, and Jon Kleinberg. "The paradox of second-order homophily in networks." Scientific Reports 11.1 (2021): 1-10.

---

### Official Review · Reviewer_Y2Du · 2022-07-12

**Rating:** 4
**Confidence:** 4
**Soundness:** 2 fair
**Presentation:** 3 good
**Contribution:** 2 fair

**Summary:**

This paper investigates the relationship between heterophily and the performance of current GNNs. First, the paper proposes a novel homophily metric that specifics harmful heterophily. The metric is shown to be more correlated with the GNNs’ performances than traditional metrics. To handle with harmful heterophily, based on the metric, the paper proposes Adaptive Channel Mixing (ACM) Framework. Extensive experiments are conducted on real-world datasets that verify the superiority of ACM framework.

**Questions:**

refer to "limitations"

**Limitations:**

1.	The proposed diversification operation is not novel since many previous works have utilized high- frequency components directly or indirectly. The authors have emphasized the difference between ACM and GPRGNN/FAGCN in node-wise channel mixing, but GPRGNN also contains node- wise feature transformation before the propagation step, which could cause γ_k to be different for each node. And the attention mechanism in FAGCN could be seen as a node-wise mixing as well. While the superiority of ACM over other mixing mechanisms is revealed in the experiment, I would be glad to see a more intuitive explanation or example of why ACM is better.
2.	The homophily metric on real-world datasets are indistinguishable. In table 8, homophily metrics on most datasets are above 0.8, and seem not correlated with GNNs’ performances. (e.g. chameleon, film, squirrel) I wonder if the proposed metric is still instructive in real-world datasets.
3.	Why are there some bumps in figure 2(c), 2(d) when h(G)=1.0 and h(g)=0.0 ?
4.	There seems to be too many overfull lines.
5.	In Fig.2, please explain the performance drop in the interval [0.0, 0.2] and why (d) does not include this interval.


**Strengths And Weaknesses:**

1.	The proposed homophily metric is interesting and makes sense. The given example in Fig. 1 and comparison with other metrics in Fig.2 nicely illustrate the advantages of this new metric.
2.	The experiments including detailed ablation study are clear and comprehensive.

---

> ### Author Response · Authors · 2022-08-02
> **Author Response to Reviewer Y2Du (Part 1)**
>
> ### Q1.
> The proposed diversification operation is not novel since many previous works have utilized high-frequency components directly or indirectly. The authors have emphasized the difference between ACM and GPRGNN/FAGCN in node-wise channel mixing, but GPRGNN also contains node-wise feature transformation before the propagation step, which could cause γ_k to be different for each node. And the attention mechanism in FAGCN could be seen as a node-wise mixing as well. While the superiority of ACM over other mixing mechanisms is revealed in the experiment, I would be glad to see a more intuitive explanation or example of why ACM is better.
>
> ### R1.
> We have discussed the differences between our proposed model and GPRGNN and FAGCN in Appendix J. We will elaborate them for you here.
> 1) Difference with GPRGNN:
>
> - GPRGNN does not feed distinct $\textbf{node-wise feature transformation}$  to different "multi-scale channels"
>
> We first rewrite GPRGNN as
> $$\mathbf{Z} = \sum\limits\_{k=0}^{K} \gamma\_{k} \mathbf{H}^{(k)} = \sum\limits\_{k=0}^{K} \gamma\_{k} I \mathbf{H}^{(k)} = \sum\limits\_{k=0}^{K} diag(\gamma\_{k}, \gamma\_{k},\dots,\gamma\_{k}) \mathbf{H}^{(k)}, \text{ where } \mathbf{H}^{(k)} = \hat{A}\_{\text{sym}}\mathbf{H}^{(k-1)}, \mathbf{H}^{(0)}\_{i:} = f\_\theta(X\_{i:}).$$
> From the above equation we can see that $\mathbf{Z} = \sum\limits\_{k=0}^{K} \gamma\_{k} \hat{A}\_{\text{sym}}^k f\_\theta(X\_{i:})$, i.e. the $\textbf{node-wise feature transformation}$ in GPRGNN is only learned by the same $\theta$ for all the "multi-scale channels". But in the ACM framework, different channels extract distinct information with different parameters separately.
>
> - GPRGNN does not have node-wise mixing mechanism.
>
> There is no node-wise mixing in GPRGNN. The mixing mechanism in GPRGNN is $\mathbf{Z} = \sum\limits\_{k=0}^{K} diag(\gamma\_{k}, \gamma\_{k},\dots,\gamma\_{k}) \mathbf{H}^{(k)}$, i.e. for each "multi-scale channel $k$", all nodes share the same mixing parameter $\gamma_{k}$. But in the ACM framework, the node-wise channel mixing can be written as $\mathbf{Z} = \sum\limits_{k=0}^{K} diag(\gamma_{k}^1,\gamma_{k}^2,\dots,\gamma_{k}^N) \mathbf{H}^{(k)}$, where $K$ is the number of channels, $N$ is the number of nodes and $\gamma_{k}^i, i=1,\dots,N$ are the mixing weights that are learned by node $i$ to mix channel $k$. ACM and ACMII allow GNNs to learn more diverse mixing parameters in diagonal than GPRGNN and thus, have stronger expressive power than GPRGNN.
>
> 2) FAGCN: This question is similar to the Q1 from Reviewer 9hm2, we will elaborate the answer here for you.
>
> - The targets of node-wise operations in ACM (channel mixing) and FAGCN (negative message passing) are different.
>
> Instead of using a fixed low-pass filter $\hat{A}$, FAGCN tries to learn a more powerful aggregator $\hat{A}'$ based on $\hat{A}$ by allowing negative message passing. The node-wise operation in FAGCN is similar to GAT [3] which is trying to modify the $\textbf{node-wise filtering (message passing) process}$, i.e. for each node $i$, it assigns different weights $\alpha_{ij} \in [-1,1]$ to different neighborhood nodes (equation 7 in FAGCN paper).  The goal of this node-wise operation in FAGCN is $\textbf{to learn a new filter during the filtering process node-wisely}$. But in ACM, the node-wise operation is to $\textbf{mix the filtered information}$ from each channel which is processed by different fixed filters. The targets of two the node-wise operations are actually different things.
>
> - FAGCN does not learn distinct information from different "channels". FAGCN uses simple addition to mix information instead of node-wise channel mixing mechanism
>
> In addition, the learned filter $\hat{A}'$ can be decomposed as follows: $\hat{A}'=\hat{A}_1' + (-\hat{A}_2')$, where $\hat{A}_1'$ and $-\hat{A}_2'$ represent  positive and negative edge (propagation) information, respectively. But FAGCN does not feed distinct information to $\hat{A}_1'$ and $-\hat{A}_2'$. Moreover, the aggregated $\hat{A}_1' X$ and "diversified" information $(-\hat{A}_2') X$ are simply added together instead of using any node-wise channel mixing. In ACM, we learn distinct information separately in each channel with different parameters and add them adaptively and node-wisely instead of just adding them together. In section 6.1, the ablation study empirically shows that node-wise adaptive channel mixing is better than simple addition.
>
> Also, as we mentioned in the contribution part, we do not try to facilitate learning filters with high expressive power, e.g. FAGCN, GPRGNN, BernNet. The goal of ACM is that, when given a filter with certain expressive power, we can extract richer information from additional channels in a certain way to address heterophily. This makes ACM more flexible and easier to implement.
>
> From the above arguments we can see that ACM is different from GPRGNN and FAGCN.

---

> > ### Author Response · Authors · 2022-08-02
> > **Author Response to Reviewer Y2Du (Part 2)**
> >
> > ### Q2.
> > The homophily metric on real-world datasets are indistinguishable. In Table 8, homophily metrics on most datasets are above 0.8, and seem not correlated with GNNs’ performances. (e.g. chameleon, film, squirrel) I wonder if the proposed metric is still instructive in real-world datasets.
> >
> > ### R2.
> > Thanks for going through Table 8 carefully. The proposed metrics are still instructive in real-world datasets and the explanations are given in Appendix H. For your interest, we will collect them here:
> >
> > Firstly, we need to clarify that, for each curve in the synthetic experiments, the node features are fixed and we only generate graphs with different homophily levels, i.e. only change graph structures. But in real-world tasks, different datasets have different features and aggregated features. Thus, to get more instructive information for different datasets and compare them, we need to consider more metrics, e.g. feature-label consistency and aggregated-feature-label consistency. With the similarity score of the features $S_\text{agg}\left(S(I,X)\right)$ and aggregated features $S_\text{agg}\left(S(\hat{A},X)\right)$ listed in Table 8, our methods open up a new way of analyzing and comparing the performance of graph-agnostic models and graph-aware models in real-world tasks. Here are two concrete explanations to the results displayed in Table 8.
> >
> > - Explanation 1: It is observed that GCN (graph-aware model) underperforms MLP-2 (graph-agnostic model) on $\textit{Cornell, Wisconsin, Texas, Film}$ and people commonly think that the bad graph structure (low $H_\text{edge},H_\text{node},H_\text{class}$ values in Table 4) is the reason for performance degradation. But based on the high aggregation homophily values, the graph-label inconsistency is not the main cause of it. Furthermore, from Table 8 we can see that the $S_\text{agg}\left(S(\hat{A},X)\right)$ for the above 4 datasets are lower than their corresponding $S_\text{agg}\left(S(I,X)\right)$, which implies that it is the aggregated-feature-label inconsistency that causes the performance degradation, i.e. the aggregation step actually decrease the quality of node features rather than making them more distinguishable.
> >
> > - Explanation 2: For the rest 5 datasets $\textit{Chameleon, Squirrel, Cora, Citeseer, PubMed}$, we all have $S_\text{agg}\left(S(\hat{A},X)\right)$ larger than $S_\text{agg}\left(S(I,X)\right)$ except $\textit{PubMed}$, which means the aggregated features have higher quality than raw features. We can see that the proposed metrics are much more instructive than the existing ones.
> >
> > We also need to point out that (modified) aggregation similarity score, $S_\text{agg}\left(S(\hat{A},X)\right)$ and $S_\text{agg}\left(S(I,X)\right)$ are not deciding values because they only capture linear relations and a low score does not mean that the GNN models will definitely perform worse than NNs. As we mentioned in limitation part, in practice, we also need to consider the non-linear relation among labels, features and graph structure, which is lacking in the existing metrics and our metrics (this might explain the failure of our metrics on $\textit{PubMed}$). But our proposed metrics can be a good starting point for future research.
> >
> > ### Q3.
> > Why are there some bumps in figure 2(c), 2(d) when h(G)=1.0 and h(g)=0.0 ?
> >
> > ### R3.
> > The bumps in figure 2(c) are because of numerical perturbation and lack of ability to capture correct relation between graph structure and GNN performance.
> >
> > The bumps in figure 2(d) are because of small numerical perturbation.
> >
> > ### Q4.
> > There seems to be too many overfull lines.
> >
> > ### R4.
> > We will make changes in the revised version.
> >
> > ### Q5.
> > In Fig.2, please explain the performance drop in the interval [0.0, 0.2] and why (d) does not include this interval.
> >
> > ### R5.
> > 1). For Figure 2(a,b), just as what we show in the example in Figure 1, when the homophily value is extremely low (near 0), the node features are actually distinguishable after aggregation step. When the homophily value starts to get higher (0 --->0.2), node features from different classes will actually be mixed and become indistinguishable. Figure 2(a,b) verify this phenomenon and in Appendix B, we theoretically prove this and calculate the homophily value to reach the lowest point for regular graphs.
> >
> > 2). This is because, for all the generated graphs, their modified aggregation homophily values are larger than 0.2. Here is a simplified example to help you understand how we plot Figure 2.
> >
> > As the graph generation process mentioned in our paper, suppose we generate 3 graphs with $H_\text{edge}=0.1,0.5,0.9$, the test accuracy of GCN on these 3 synthetic graphs are $0.8,0.5,0.9$. For those graphs, we calculate their $H_\text{agg}^M$ and suppose we get $H_\text{agg}^M=0.7,0.4,0.8$. Then we will draw the performance of GCN under $H_\text{agg}^M$ with ascend x-axis order $[0.4,0.7,0.8]$ and the corresponding reordered y-axis $[0.5,0.8,0.9]$. Other figures are drawn in the same way.

---

> > > ### Comment · Reviewer_Y2Du · 2022-08-08
> > > **Some general questions**
> > >
> > > Thanks for these detailed explanations!
> > >
> > > To my knowledge, it has been a consensus that diverse local assortativity or say homophilic levels are the pain for node classification, no matter whether the GNN is a low/high-pass filter. Meanwhile, there are some recent works, e.g., GPRGNN and BernNet, that can fit arbitrary graph filters. Thus, I believe the proposed mixing mechanism does work in some circumstances, particularly a graph with diverse local assortativity levels. However, I still want to discuss about the following questions:
> > >
> > > - Does the high-pass + low-pass + identity (full-pass?) channels together improve the expressiveness of GPRGNN/BernNet?
> > > - What is the ideal information (or say local patterns) should be the sufficient information for determining the coefficients?
> > > - Have you compare the corresponding spectral filters learned against that learned by GPRGNN?

---

> > > > ### Author Response · Authors · 2022-08-09
> > > > **Author Response to the General Questions from Reviewer Y2Du**
> > > >
> > > >
> > > >
> > > > Thanks so much for your recognition of our proposed node-wise channel mixing mechanism. And we find that those general questions are quite interesting and valuable for the whole GNN community and we are glad to share our opinions here.
> > > >
> > > >
> > > > ### Q1.
> > > > Does the high-pass + low-pass + identity (full-pass?) channels together improve the expressiveness of GPRGNN/BernNet?
> > > >
> > > > ### R1.
> > > > We do not have a rigorous theoretically proved answer at this moment, but we believe the answer is yes and to prove it, we need to find a different perspective beyond the expressiveness of graph filters. GPRGNN/BernNet and many other GNN models share the common belief that "expressiveness of GNN == expressiveness of graph filters". This might be true in many cases but not always because we have other parts besides the filters in GNN that can be improved. So at the first glance, the 3-channel architecture might look ordinary from the perspective of graph filter. But when we feed distinct information to each "channel" and use node-wise channel mixing mechanism to combine the filtered information, this simple architecture becomes powerful.
> > > >
> > > > For the proof, one angle might be that ACM can make baseline GNNs fit more complex node label patterns (distribution), but we might need to take the parameter matrix and non-linearity into consideration.
> > > >
> > > >
> > > > ### Q2.
> > > > What is the ideal information (or say local patterns) should be the sufficient information for determining the coefficients?
> > > >
> > > > ### R2.
> > > >
> > > > In our opinion, the coefficients $\alpha\_L,\alpha\_H,\alpha\_I$ do not only depend on the local patterns of each nodes, but also depend on the relation between its local patterns and its neighbors' local patterns and label distribution. It will be interesting to formulate it into an optimization problem and solve it directly in the future, but the way we have on hand is to optimize it end-to-end by gradient descent with backpropagation.
> > > >
> > > > ### Q3.
> > > > Have you compare the corresponding spectral filters learned against that learned by GPRGNN?
> > > >
> > > > ### R3
> > > >
> > > > We did not compare the spectral filters against GPRGNN, because our analysis and derivation of ACM are from node level instead of spectral domain. Also,
> > > > ACM-GCN is not learning a spectral filter because each channel has different input information rather than the same information. But we will consider visualizing the output layer on spectral domain and compare it with GPRGNN as you suggest. What we have for now is the t-SNE visualization of the output layer (Figure 4).
> > > >
> > > > In addition, we would like to share an opinion that are against the mainstream opinion about the spectral analysis of the filters in GNNs: The spectral analysis is based on the eigensystem of graph Laplacian and those eigenvectors are functions with variant smoothness defined on the given graph structure, e.g. connected nodes share similar values. So when the given graph structure is "bad" or "trivial", the smoothness of those eigenvectors becomes trivial as well. The traditional spectral analysis is valid for "good" graph structures, e.g. Cora, Citeseer, PubMed, but for "bad" graphs, we might need to consider more, e.g. label distribution, post-aggregation node similarity. This is just one of our opinions and not conclusive. We share it here for your interest.
> > > >
> > > >
> > > >
> > > >
> > > > #### We are very glad to discuss these general questions with you and we find them pretty interesting. If your concerns on our paper are addressed, we politely request a raise of your rating. We will appreciate it. If you still have any question left, please let us know. Thanks.
> > > >
> > > > Authors

---

### Official Review · Reviewer_9hm2 · 2022-07-12

**Rating:** 3
**Confidence:** 4
**Soundness:** 2 fair
**Presentation:** 3 good
**Contribution:** 2 fair

**Summary:**

This paper presents an analysis of existing homophily metrics, and proposes a new metric more informative for the performance. Based on the analysis, they design a 3-way filterbank, enabling adaptive filtering (high-pass, low-pass or identity) at different nodes. Experiments validate the effectiveness of the proposed method.

**Questions:**

1. More comparisons with FAGCN.
2. There is a gap between the proposed metric and method.
3. The improvement in Table 4 does not seem statistically significant because of high variance.


**Limitations:**

In addition to the limitations mentioned in the paper, the intrinsic relationship between the proposed metric and method should be taken into consideration. No potential negative societal impact.

**Strengths And Weaknesses:**

Strength:
1.	This paper is the first to analyze heterophily from post-aggregation node similarity.
2.	The proposed filterbank is plug-and-play for backbone GNNs, which extracts richer information from additional channels.
Weakness:
1.	Contribution is not convincing. They argue that the traditional adaptive filterbank uses a scalar weight shared by all nodes, and their proposed method learns different weights for different nodes. However, in my opinion, FAGCN can do the same thing.
2.	There is a gap between the proposed metric and method. Based on post-aggregation node similarity, they propose an aggregation similarity metric. However, the final 3-channel filterbank has nothing to do with the above metric.
3.	The novelty of the idea is not enough. In addition to the limitations pointed out above, both new metric and method are relatively straightforward.
4.	The improvement in Table 4 does not seem statistically significant because of high variance.
5.	There is a problem with the typesetting of the paper.

---

> ### Author Response · Authors · 2022-08-02
> **Author Response to Reviewer 9hm2 (Part 1)**
>
> ### Q1.
> Contribution is not convincing. They argue that the traditional adaptive filterbank uses a scalar weight shared by all nodes, and their proposed method learns different weights for different nodes. However, in my opinion, FAGCN can do the same thing.
>
> ### R1.
>
> We have discussed the differences between ACM and FAGCN in Appendix J. We would like to elaborate them here for you:
>
> - The targets of node-wise operations in ACM (channel mixing) and FAGCN (negative message passing) are different.
>
> Instead of using a fixed low-pass filter $\hat{A}$, FAGCN tries to learn an aggregator $\hat{A}'$ based on $\hat{A}$ by allowing negative message passing. The node-wise operation in FAGCN is similar to GAT [3] which tries to modify the $\textbf{node-wise filtering (message passing) process}$, i.e. for each node $i$, it assigns different weights $\alpha_{ij} \in [-1,1]$ to different neighborhood nodes (equation 7 in FAGCN paper). The goal of this node-wise operation in FAGCN is $\textbf{to learn a new filter during the filtering process node-wisely}$. But in ACM, the node-wise operation is to mix the $\textbf{filtered information}$ from each channel which is processed by different fixed filters. The targets of two the node-wise operations are actually different things.
>
> - FAGCN does not learn distinct information for different "channels". FAGCN uses simple addition to mix information instead of node-wise channel mixing mechanism
>
> The learned filter $\hat{A}'$ can be decomposed as follows: $\hat{A}'=\hat{A}_1' + (-\hat{A}_2')$, where $\hat{A}_1'$ and $-\hat{A}_2'$ represent positive and negative edge (propagation) information respectively. But FAGCN does not feed distinct information to $\hat{A}_1'$ and $-\hat{A}_2'$. Moreover, the aggregated $\hat{A}_1' X$ and "diversified" information $(-\hat{A}_2') X$ are simply added together instead of using any node-wise mixing mechanism. In ACM, we learn distinct information separately in each channel with different parameters and add them adaptively and node-wisely  instead of just adding them together, because different nodes need information from different channels. In section 6.1, the ablation study empirically shows that node-wise adaptive channel mixing is better than simple addition.
>
> Also, as mentioned in the contribution highlights, we are NOT facilitating learning filters with high expressive power, e.g. FAGCN, GPRGNN, and BernNet. Given a filter with certain expressive power, ACM can extract richer information from additional channels in a certain way to address the heterophily issue.
>
> From the above argument we can see that ACM and FAGCN are different.
>
> ### Q2.
> There is a gap between the proposed metric and method. Based on post-aggregation node similarity, they propose an aggregation similarity metric. However, the final 3-channel filterbank has nothing to do with the above metric.
>
> ### R2.
>
> Like the existing homophily metrics, the purpose of designing aggregation homophily is just to measure whether the aggregation (message passing) step would help $\textbf{uni-channel}$ graph-aware model outperform graph-agnostic model. Our proposed 3-channel architecture is beyond the uni-channel framework, and thus its performance cannot be directly measured by the proposed metric.
>
> The key part to connect the investigation part (heterophily; new metrics) and the methodology part (new model) is the post-aggregation node similarity matrix, not the metrics. The metrics are the by-products of the similarity matrix.
>
> In methodology part, based on the post-aggregation node similarity matrix, we can show  the effectiveness of the high-pass filter on addressing heterophily, which is one of the main reasons that we design the 3-channel architecture.

---

> > ### Author Response · Authors · 2022-08-02
> > **Author Response to Reviewer 9hm2 (Part 2)**
> >
> > ### Q3.
> > The novelty of the idea is not enough.
> >
> > ### R3.
> >
> > We have summarized the main contribution of this paper in the contribution part and we will elaborate it here for you.
> >
> > 1. To our knowledge, we are the first to analyze heterophily from the post-aggregation node similarity perspective. Based on the proposed similarity matrix, we derive novel homophily metric which is verified to be superior to the existing metrics. The effectiveness of high-pass filter is also proved based on the similarity matrix, which is novel as well.
> >
> > 2. The proposed ACM framework is highly different from adaptive filterbank and existing GNNs for heterophily: 1) the traditional adaptive filterbank uses a scalar weight for each filter and this weight is shared by all nodes. In contrast, in our method different nodes can learn different weights to utilizes the $\textbf{filtered information from different channels}$ adaptively to account for heterophily; 2) Unlike existing methods that leverage the high-order filters and global property of high-frequency signals, ACM successfully addresses heterophily by $\textbf{considering only the nodewise local information adaptively}$.
> >
> > 3. Unlike existing methods that try to facilitate learning filters with high expressive power, e.g. FAGCN, GPRGNN and BernNet etc., the goal of ACM is that, when given a filter with certain expressive power, we can extract richer information from additional channels in a certain way to address heterophily. This makes ACM more flexible and easier to implement.
> >
> > ### Q4.
> >
> > The improvement in Table 4 does not seem statistically significant because of high variance.
> >
> > ### R4.
> >
> >
> > Compared to the SOTA models, the variance of our model is not large. This is consistent with some recently published papers, e.g. [1,2]. To reduce the possibility that the high variance on certain dataset would affect the model evaluation and comparison, we calculate the average rank of the performance over all datasets. From the average rank we can see that the proposed model outperforms the SOTA model.
> >
> >
> >
> > ### Q5.
> > There is a problem with the typesetting of the paper.
> >
> > ### R5.
> >
> > We will modify it in the revised version.
> >
> >
> >
> >
> > [1] He, Mingguo, Zhewei Wei, and Hongteng Xu. "Bernnet: Learning arbitrary graph spectral filters via bernstein approximation." Advances in Neural Information Processing Systems 34 (2021): 14239-14251.
> >
> > [2] Li, Xiang, et al. "Finding Global Homophily in Graph Neural Networks When Meeting Heterophily." arXiv preprint arXiv:2205.07308 (2022).
> >
> > [3] Veličković, Petar, et al. "Graph attention networks." arXiv preprint arXiv:1710.10903 (2017).

---

> > > ### Author Response · Authors · 2022-08-09
> > > **Discussion**
> > >
> > > Dear Reviewer 9hm2,
> > >
> > > Thanks for spending your time evaluating our paper. Since you have negative rating on our paper, we would like to know if you still have any question left to discuss. If your concerns are addressed, we respectfully request a raise of your rating. We will appreciate that.
> > >
> > > Authors

---

### Official Review · Reviewer_icHY · 2022-07-17

**Rating:** 6
**Confidence:** 4
**Soundness:** 3 good
**Presentation:** 2 fair
**Contribution:** 3 good

**Summary:**

This paper first points out that existing homophily metrics cannot precisely reflect the performance of GNN in some cases, and develops a new one based on the similarity comparison between the local neighbors in the same and different class. Next, based on the proposed metric, the paper shows the case that a high-pass filter can address the harmful heterophily, and further propose a node-wise mixing filter that combines low-pass and high-pass filters.

**Questions:**

What's the correlation between the high-pass filter and diversification? It would be better to define diversification and describe how a high-pass filter can extract such information. A synthetic example is not clear enough.

**Limitations:**

1. Some experiments are missing: (1) The comparison between the existing homophily metrics and the proposed one on real-world datasets. (2) The case study of \alpha score on different nodes. These analysis could better help readers understand the effect of architecture and hyper-parameters of the architecture.

2. The theoretical analysis is based on the random walk Laplacian. It would be better if the authors can extend it to the widely used symmetric Laplacian.

**Strengths And Weaknesses:**

Pros
1. The motivation for the proposed homophily metric is clear and theoretically guaranteed, although the analysis of one-layer SGC is relatively simplistic. The observations from Table 8 show that the metric can accurately determine whether the additional graph information is harmful.
2. Node-wise aggregation is reasonable since the local structure differs between different nodes.
3. The experimental results on many different GNNs shown in the Appendix are complete and convincing.
​
Cons

---

> ### Author Response · Authors · 2022-08-02
> **Author Response to Reviewer icHY (Part 1)**
>
> Thanks for your constructive comments and suggestions. Here are our answers to your questions.
>
> ### Q1.
> What's the correlation between the high-pass filter and diversification? It would be better to define diversification and describe how a high-pass filter can extract such information. A synthetic example is not clear enough.
>
> ### R1.
>
> High-pass filter and diversification are essentially the same operation described in matrix and node form. To be more specific
> $$\text{HP filter: } (I-\hat{A})X, \  \ \  \text{Diversification on node $i$: } [(I-\hat{A})X]\_{i,:}  = X\_{i:} - \sum\limits\_{k\in \mathcal{N}(i)} \frac{1}{d\_i} X\_{k:}$$
>
> As we mentioned in our paper, diversification operation extract neighborhood dissimilarity by a subtraction of node features $X\_{i:}$ and aggregated features from neighbors $\sum\limits\_{k\in \mathcal{N}(i)} \frac{1}{d\_i} X\_{k:}$.
>
>
> ### Q2.
> Some experiments are missing: (1) The comparison between the existing homophily metrics and the proposed one on real-world datasets. (2) The case study of $\alpha$ score on different nodes. These analysis could better help readers understand the effect of architecture and hyper-parameters of the architecture.
>
> ### R2.
>
> (1)
> The results of existing metrics are reported in Table 4 in Appendix A. The results of proposed metrics on real-world datasets are reported in Table 8 in Appendix H and we also provide an explanation to those results and some comparisons with the existing metrics. We elaborate the explanations about the results in Table 4 and 8 in the following paragraphs for you.
>
> There are three key factors that influence the performance of GNNs in real-world tasks: labels, features and graph structure. The existing metrics and (modified) aggregation homophily tries to investigate the consistency of graph structure and labels from different principles with $\textbf{given features}$. And the post-aggregation node similarity principle is verified to be advantageous over homophily principle through the synthetic experiments.
>
> In real-world tasks, different datasets have features with variant quality. Thus, besides graph-label consistency, we need to consider feature-label consistency and aggregated-feature-label consistency as well to fully investigate the performance of NNs and GNNs. With aggregation similarity score of the features $S_\text{agg}\left(S(I,X)\right)$ and aggregated features $S_\text{agg}\left(S(\hat{A},X)\right)$ listed in Table 8, our methods open up a new way of analyzing and comparing the performance of graph-agnostic models and graph-aware models in real-world tasks. Here are two cases of result comparison.
>
> Case 1: It is observed that GCN (graph-aware model) underperforms MLP-2 (graph-agnostic model) on $\textit{Cornell, Wisconsin, Texas, Film}$ and people commonly thinks that the bad graph structure (low $H_\text{edge},H_\text{node},H_\text{class}$ values in Table 4) is the reason for performance degradation. But based on the proposed aggregation homophily, the graph-label inconsistency is not the main cause of it. Furthermore, from Table 8 we can see that the $S_\text{agg}\left(S(\hat{A},X)\right)$ for the above 4 datasets are lower than their corresponding $S_\text{agg}\left(S(I,X)\right)$, which implies that it is the aggregated-feature-label inconsistency that causes the performance degradation, i.e. the aggregation step actually decrease the quality of node features rather than making them more distinguishable.
>
> Case 2: For the rest 5 datasets $\textit{Chameleon, Squirrel, Cora, Citeseer, PubMed}$, we all have $S_\text{agg}\left(S(\hat{A},X)\right)$ larger than $S_\text{agg}\left(S(I,X)\right)$ except for $\textit{PubMed}$, which has two close values. But the existing metrics give extremely low values to $\textit{Chameleon, Squirrel}$, which implies that graph-aware models would outperform graph-agnostic models. This is far from the observations in experiments.
>
>
> (2)
> In Figure 4, we have plotted the learned $\alpha$ values in output layer of ACM-GCN trained on Squirrel. The $\alpha$ values show that the additional channels play a nontrivial role for most of the nodes. We have put the pictures for other datasets in Appendix I2 in the revised version.

---

> > ### Author Response · Authors · 2022-08-02
> > **Author Response to Reviewer icHY (Part 2)**
> >
> > ### Q3.
> > The theoretical analysis is based on the random walk Laplacian. It would be better if the authors can extend it to the widely used symmetric Laplacian.
> >
> > ### R3.
> >
> > The definitions of the similarity matrix, (modified) aggregation similarity score and diversification distinguishability value can be extended to symmetric normalized Laplacian or other aggregation operations. But we cannot extend Theorem 1, because we need a condition that the row sum of $\hat{A}$ is not greater than 1 in the proof (see Appendix E). This condition is guaranteed for random walk normalized Laplacian but not for symmetric normalized Laplacian.
> >
> > In practice, the answer is yes. We evaluate our models with symmetric filters and compare them with random walk filters. From the following table we can see that, there is no big differences between these two filters.
> >
> > Models       |          *Random Walk*        ||  *Symmetric*        ||
> >  ------------ | :-----------: | :-----------: | :-----------: | :-----------: |
> > Datasets       |   ACM    |  ACMII |       ACM |      ACMII |
> >  Cornell | 94.75 $\pm$ 3.8 | **95.9 $\pm$ 1.83** | 94.92 $\pm$ 2.48 | 94.1 $\pm$ 2.56
> > Wisconsin | 95.75 $\pm$ 2.03 | **96.62 $\pm$ 2.44** | 95.63 $\pm$ 2.81 | 96.25 $\pm$ 2.5
> > Texas | 94.92 $\pm$ 2.88 | **95.08 $\pm$ 2.07** | 94.75 $\pm$ 2.01 | 94.59 $\pm$ 2.65
> > Film  | 41.62 $\pm$ 1.15 | **41.84 $\pm$ 1.15** | 41.58 $\pm$ 1.3 | 41.65 $\pm$ 0.6
> > Chameleon | **69.04 $\pm$ 1.74** | 68.38 $\pm$ 1.36 | 67.9 $\pm$ 2.76 | 68.03 $\pm$ 1.68
> > Squirrel | **58.02 $\pm$ 1.86** | 54.53 $\pm$ 2.09 | 54.18 $\pm$ 1.35 | 53.68 $\pm$ 1.74
> > Cora  | 88.62 $\pm$ 1.22 | **89.00 $\pm$ 0.72** | 88.65 $\pm$ 1.26 | 88.19 $\pm$ 1.38
> > Citeseer | 81.68 $\pm$ 0.97 | 81.79 $\pm$ 0.95 | **81.84 $\pm$ 1.15** | 81.81 $\pm$ 0.86
> > PubMed | 90.66 $\pm$ 0.47 | **90.74 $\pm$ 0.5** | 90.59 $\pm$ 0.81 | 90.54 $\pm$ 0.59

---

### Meta-Review · Area_Chair_jSY4 · 2022-08-26

**Recommendation:** Accept
**Confidence:** Less certain

**Metareview:**

In this submission, the authors revisit the existing homophily metrics and point out the limitations of existing metrics in analyzing the performance of GNN. Then the authors propose a novel homophily metric that specifics harmful heterophily, and further propose Adaptive Channel Mixing (ACM) framework to handle the harmful heterophily.

Although there exist some concerns about the novelty of the idea (as pointed out by 9hm2 and Y2Du), overall, the proposed metric and framework are well-motivated, interesting, and effective (as pointed out by icHY, Y2Du, and S2KT), and the experiments are comprehensive and convincing (as pointed out by icHY, Y2Du, and vQE7). Due to these, here, I recommend accepting this submission.

This submission also can be improved based on the suggestions by reviewers (such as writing and typesetting), and hope they find the discussion useful and make this submission a better one.


**Award:**

No

---

### Decision · Program_Chairs · 2022-09-14

Accept